# Epigenetic alterations facilitate transcriptional and translational programs in hypoxia

Kathleen Watt[1,2,9], Bianca Dauber[2,9], Krzysztof J. Szkop[1,9], Laura Lee[3], Predrag Jovanovic[4,5], Shan Chen[1], Ranveer Palia[4,5], Julia A. Vassalakis[2], Tyler T. Cooper[2,6], David Papadopoli[4,7], Laìa Masvidal[1], Michael Jewer[3], Kristofferson Tandoc[4,5], Hannah Plummer[2], Gilles A. Lajoie[6], Ivan Topisirovic[4,5,7,8], Ola Larsson[1,10] ✉ & Lynne-Marie Postovit[2,3,10] ✉

Adaptation to cellular stresses entails an incompletely understood coordination of transcriptional and post-transcriptional gene expression programs. Here, by quantifying hypoxia-dependent transcriptomes, epigenomes and translatomes in T47D breast cancer cells and H9 human embryonic stem cells, we show pervasive changes in transcription start site (TSS) selection associated with nucleosome repositioning and alterations in H3K4me3 distribution. Notably, hypoxia-associated TSS switching was induced or reversed via pharmacological modulation of H3K4me3 in the absence of hypoxia, defining a role for H3K4me3 in TSS selection independent of HIF1-transcriptional programs. By remodelling 5′UTRs, TSS switching selectively alters protein synthesis, including enhanced translation of messenger RNAs encoding pyruvate dehydrogenase kinase 1, which is essential for metabolic adaptation to hypoxia. These results demonstrate a previously unappreciated mechanism of translational regulation during hypoxia driven by epigenetic reprogramming of the 5′UTRome.

Cellular plasticity enables adaptation to microenvironmental changes, and underlies cancer cell survival and metastasis[1]. Plasticity requires coordinated reprogramming of gene expression at transcriptional and post-transcriptional levels to reshape the proteome[2]. Cancer cells commonly adapt to hypoxia, which enhances plasticity and cancer stem-cell-like phenotypes[1,3]. Hypoxia imposes metabolic restrictions impacting gene expression[4,5] and the epigenome[6–8]. This includes accumulation of H3K4me3 (trimethylation of histone H3 lysine 4) around transcription start sites (TSSs), correlating with the degree and consistency of transcriptional activation[9–12]. H3K4 methylation is deposited by complex proteins associated with set1 (COMPASS) methyltransferase complexes and erased by Jumonji C histone demethylases, including KDM5 (ref. 13). Although hypoxia stabilizes HIF1α and thereby induces transcription of several Jumonji C histone demethylases[14–16], their enzymatic activity declines when $O_2$ is limiting[17], leading to increased H3K4 methylation, chromatin remodelling and enhanced cellular plasticity[6,8]. However, the impact of this oxygen-sensing capacity on processes such as TSS selection remains unclear.

[1]Department of Oncology-Pathology, Science for Life Laboratory, Karolinska Institutet, Stockholm, Sweden. [2]Department of Biomedical and Molecular Sciences, Queen's University, Kingston, Ontario, Canada. [3]Department of Oncology, University of Alberta, Edmonton, Alberta, Canada. [4]Lady Davis Institute, Sir Mortimer B. Davis Jewish General Hospital, Montréal, Québec, Canada. [5]Division of Experimental Medicine, McGill University, Montréal, Québec, Canada. [6]Department of Biochemistry, Western University, London, Ontario, Canada. [7]Gerald Bronfman Department of Oncology, McGill University, Montréal, Québec, Canada. [8]Department of Biochemistry, McGill University, Montréal, Québec, Canada. [9]These authors contributed equally: Kathleen Watt, Bianca Dauber, Krzysztof J. Szkop. [10]These authors jointly supervised this work: Ola Larsson, Lynne-Marie Postovit. ✉e-mail: ola.larsson@ki.se; l.postovit@queensu.ca

Hypoxia suppresses cap-dependent messenger RNA translation[2,5] via mTOR inhibition[18] and subsequent reduction in eIF4F levels[19]. Hypoxia also induces the integrated stress response (ISR), wherein eIF2α phosphorylation-dependent suppression of eIF2B attenuates ternary complex recycling and initiator transfer RNA delivery[5,20]. This reprogramming of the translational apparatus reduces global protein synthesis while allowing selective translation of messenger RNAs encoding central regulators of stress responses[5]. Such transcripts (for example ATF4) often contain distinct 5′ untranslated region (5′UTR) features, including upstream open reading frames (uORFs)[21] that facilitate translation under hypoxia[22]. Other 5′UTR features, including length[23] and 5′ terminal oligopyrimidine (TOP) motifs, render translation selectively mTOR-sensitive[23–25]. Notably, mRNAs encoding stemness factors (NODAL, SNAIL and NANOG) express 5′UTR isoforms with features that enhance translation under hypoxia[26], suggesting that in addition to reprogramming of the translational machinery, changes in 5′UTR composition may also drive adaptive protein synthesis.

Here, we profiled epigenomes, transcriptomes, 5′UTRomes and translatomes under normoxia and hypoxia. We thereby uncovered that perturbation of H3K4me3 facilitates hypoxia-induced changes in mRNA translation via abundant TSS switching, remodelling 5′UTRs, which, alongside hypoxia-induced alterations of the translational machinery, helps shape an adaptive translatome. This mechanism of TSS switching regulates synthesis of key factors enabling metabolic adaptations to hypoxia, including pyruvate dehydrogenase kinase 1 (PDK1). Many 5′UTR isoform changes occur independently of altered transcript abundance and seem to depend on reduced KDM5 activity, leading to H3K4me3 expansion and redistribution. Pharmacological KDM5 inhibition mimics hypoxia-induced TSS switching and modulates the proteome in the absence of HIF1 stabilization or altered mRNA abundance. Conversely, inhibiting mixed lineage leukaemia (MLL)-containing COMPASS methyltransferases blocks a subset of TSS switching and reduces cellular proliferation under hypoxia. Collectively, hypoxia-induced H3K4me3 remodelling alters TSS selection to establish an adaptive translatome.

## Results

### Hypoxia-induced TSS switching results in extensive remodelling of 5′UTRs

We previously observed that several stem cell-associated factors express multiple 5′UTR isoforms, with some preferentially translated under hypoxia[26]. Divergent cell types respond to hypoxia with differing kinetics, modifying gene expression and mRNA translation regulators at different time points and oxygen concentrations[26,27]. Accordingly, to investigate hypoxia-associated 5′UTR isoform dynamics, we performed nanoCAGE sequencing on total mRNA from T47D breast cancer cells and H9 human embryonic stem (hES) cells cultured for 48 or 24 h, respectively, in hypoxia (0.5% $O_2$) or normoxia (20% $O_2$) (Fig. 1a). These conditions induce *LOX1* mRNA and HIF1α protein levels, suppress mTOR signalling and induce ISR[26–28]. We detected ~20,000 RefSeq transcripts at a near-saturation sequencing depth (Extended Data Fig. 1a,b). Expression of hypoxia-associated transcripts was increased and samples were separated by condition in principal-component analyses (PCA) for both cell types (Extended Data Fig. 1c,d). As previously observed[23], the weighted mean 5′UTR length was often shorter than RefSeq annotations (Extended Data Fig. 1e). We defined TSS clusters for each transcript, representing distinct 5′UTR isoforms. Over 70,000 TSS clusters were identified in both cell types, with >80% of protein-coding transcripts expressing multiple 5′UTR isoforms (Fig. 1b). More than 20% of transcripts showed hypoxia-induced changes in 5′UTR availability, excluding TSS-switching events that alter open reading frames (ORFs) (Fig. 1c, Extended Data Fig. 1f and Supplementary Data 1). Many of these changes occurred independently of changes in mRNA abundance (Fig. 1d and Supplementary Data 2). For example, *SH3BP2* mRNA expression increased under hypoxia alongside a switch in 5′UTR isoform expression (Fig. 1e), whereas *PELP1* underwent TSS switching under hypoxia despite unchanged overall mRNA levels (Fig. 1f).

As TSS switching was often complex, involving multiple 5′UTR isoforms (Fig. 1g), we employed change-point analysis[29] to identify 5′UTR sequence segments (the regions before, between and after change points (short, middle and long)) enriched or depleted under hypoxia, and assigned a score describing the extent of isoform switching (TSS-switch score; Methods and Supplementary Data 3). For instance, *SLC25A20* (Fig. 1g) exhibited change points 58 and 91 nt upstream of the start codon (Fig. 1h), with an ~20% enrichment in relative expression of shorter isoforms and depletion of isoforms longer than 91 nt under hypoxia (Fig. 1i). The resulting TSS-switch score of ~50 (Fig. 1i) indicates substantial 5′UTR remodelling, occurring without changes in overall mRNA levels (Fig. 1j). Applying this method to all transcripts with significant changes in TSS usage under hypoxia revealed that TSS scores tended to be larger in T47D than in H9 cells (Extended Data Fig. 1g). Furthermore, in T47D cells,

**Fig. 1 | Hypoxia-induced TSS switching results in extensive remodelling of 5′UTRs. a**, T47D and H9 cells were cultured under hypoxia (0.5% $O_2$) or normoxia (20% $O_2$) and 5′UTR isoforms were quantified transcriptome-wide using nanoCAGE sequencing. T47D were treated for 48 h (*n* = 3 hypoxia, *n* = 2 normoxia) and H9 for 24 h (*n* = 4). *n* denotes independent experiments. **b**, Histograms showing the number of TSS clusters per protein-coding transcript detected by nanoCAGE in hypoxia and normoxia. Percentage of transcripts with more than one isoform and mean isoforms per condition are indicated. **c**, Bar plot indicating the proportions of protein-coding transcripts with altered TSS usage between hypoxia and normoxia (statistical approach described in Methods; FDR < 0.15) in T47D (*n* = 2,552) and H9 cells (*n* = 3,423). **d**, Pie charts indicating the proportions of transcripts with TSS switching from **c** that have significantly (anota2seq[36] analysis of RNA-seq data; FDR < 0.15) increased (up), decreased (down), or unchanged overall mRNA expression between hypoxia and normoxia in T47D and H9 cells. **e**, An example of a transcript (NM_001122681; *SH3BP2* mRNA) that undergoes quantitative (Methods) hypoxia-induced TSS switching in T47D cells. Top panel: *x* axis represents distance to the AUG start codon; black bar denotes the RefSeq-annotated 5′UTR. Bar plot of total transcript expression (all 5′UTR isoforms) (bottom left). Mean ± s.d.; *P* = 0.0085; two-sided *t*-test. Bar plots summarizing 5′UTR isoform expression within each TSS cluster (positions relative to the start codon indicated below) (bottom right). The percentage of expression of each TSS cluster relative to the total transcript expression is indicated for each condition above. Mean ± s.d.; NS, not significant; *P* = 0.013 for 75–114 nt; *P* = 0.94 for 69 nt; *P* = 0.14 for 57–62 nt; *P* = 0.14 for 26–28 nt; two-sided *t*-test (*n* = 3 hypoxia; *n* = 2 normoxia; independent experiments). **f**, The same as in **e**, but an example of a transcript (NM_014389; *PELP1* mRNA) with a qualitative hypoxia-induced TSS change in T47D cells. Bar plots indicate mean ± s.d.; *P* = 0.3264 for overall expression; *P* = 3.2 × 10$^{-16}$ for 27–37 nt; *P* = 0.15 for 15–18 nt; *P* = 0.74 for 2–8 nt; two-sided *t*-test. **g**–**j**, An example of change-point analysis used to identify 5′UTR sequence regions in NM_000387 (*SLC25A20* mRNA) enriched or depleted by hypoxia-induced TSS switching seen in **g**. Change points in the difference in cumulative expression between hypoxia and normoxia (at 58 and 91 nt, dotted orange lines) define sequence segments with differential enrichment or depletion (**h**). The degree of isoform switching is scored by the maximum difference in isoform expression across identified 5′UTR regions (TSS-switch score) (**i**), demonstrating a large difference in isoform expression, without changes in overall transcript levels (*P* = 0.3391) (**j**). Bar plots show mean ± s.d.; two-sided *t*-test (*n* = 3 hypoxia; *n* = 2 normoxia; independent experiments). **k**, Bar plots displaying categories of 5′UTR sequences enriched by significant hypoxia-induced TSS switching, identified by change-point analysis. **l**, Venn diagram showing the overlap of transcripts with significant hypoxia-induced TSS switching in T47D and H9 cells (top). Of these (*n* = 586), 219 transcripts had the same pattern of 5′UTR isoform switches in both cell types. Heatmap shows *z*-scores of the relative enrichment or depletion in each sequence segment defined by change points for these 219 transcripts (bottom). **m**, Venn diagram showing the overlap of genes (some have several transcript isoforms) that undergo the same pattern of hypoxia-induced TSS switching in both T47D and H9 cells, and known transcriptional targets of HIF1α (from Schödel et al.[30] and Sugimoto et al.[31]).

enrichment of 5′UTR variants favoured the shortest or longest isoforms, whereas in H9s, 5′UTRs were predominantly lengthened (Fig. 1k). Hypoxia-enriched 5′UTR sequences were more GC-rich in T47D cells with increasing TSS-switch scores (Extended Data Fig. 1h) and a similar pattern was observed in the H9 cells at lower (>15) but not higher (>50) TSS-switch scores (Extended Data Fig. 1i), possibly reflecting the cell-type-specific patterns of shifts in isoform length (Fig. 1k).

Among transcripts with TSS switching under hypoxia, 586 (~11%) were shared across cell types. Of these, 219 (~37%) exhibited highly similar changes in 5′UTR isoform expression (Fig. 1l) and were enriched for Gene Ontologies, including hypoxia response, glucose metabolism, cell cycle and proliferation, and chromatin remodelling (Extended Data Fig. 1j), suggesting that shared TSS-switching events impact core hypoxia-related processes. Notably, <2% of genes

with shared TSS switching are known HIF1α transcriptional targets[30,31] (Fig. 1m). Together, these findings reveal pervasive hypoxia-induced 5′UTR remodelling occurring largely independently of HIF1.

## TSS switching is associated with translational reprogramming under hypoxia

Translational reprogramming is a critical component of cellular adaptations to hypoxia[2,5,18,32–34]. We therefore examined the impact of 48 h (T47D) or 24 h (H9) of hypoxia (1% $O_2$) on the translatome using polysome profiling[35]. These conditions stabilized HIF1α, suppressed mTOR signalling and activated the ISR, indicated by reduced 4E-BP1 phosphorylation and increased eIF2α phosphorylation (Fig. 2a,b). As expected[2], hypoxia suppressed global translation, reducing polysome:monosome ratios by 80–85% in both cell types (Fig. 2c,d).

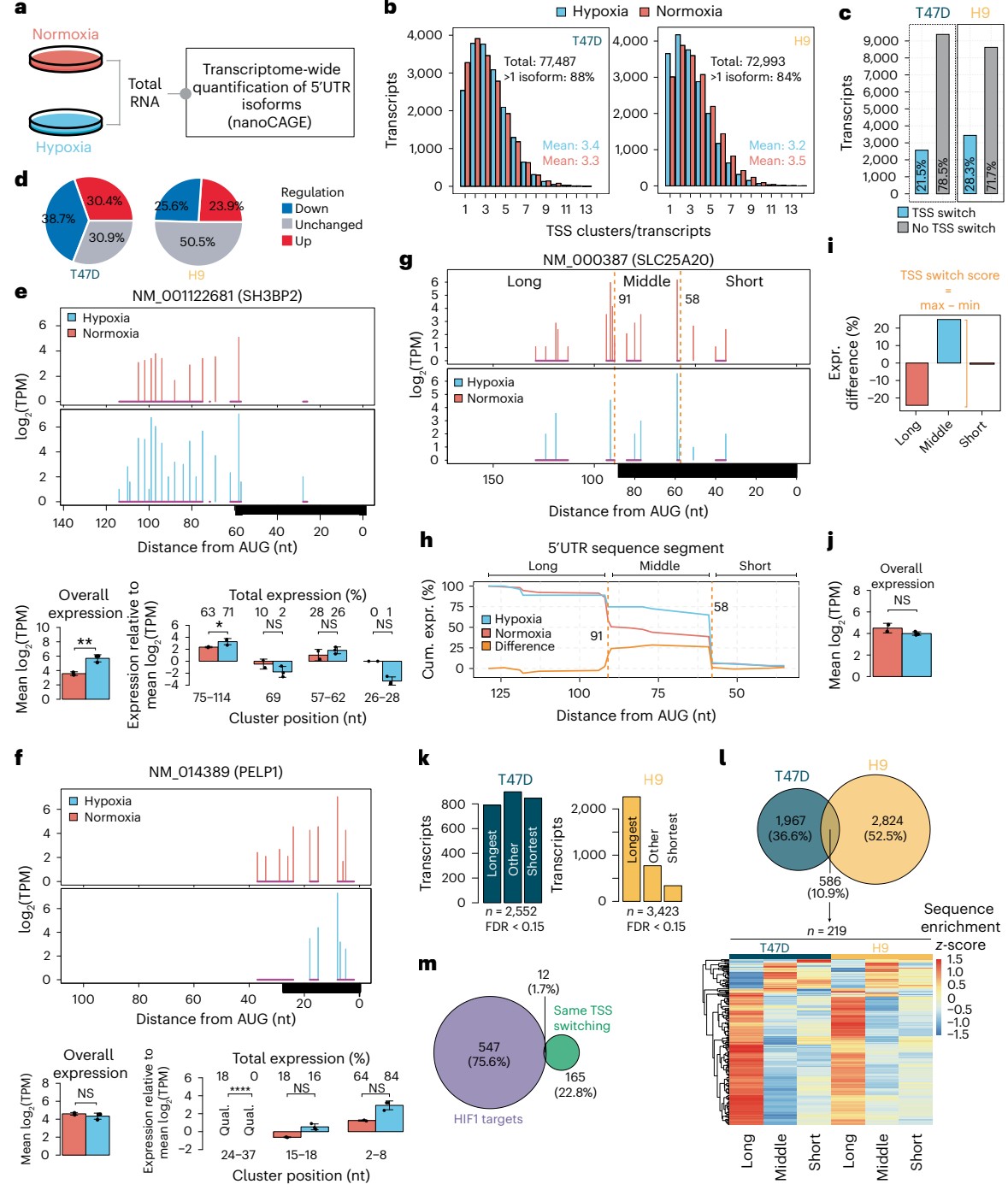

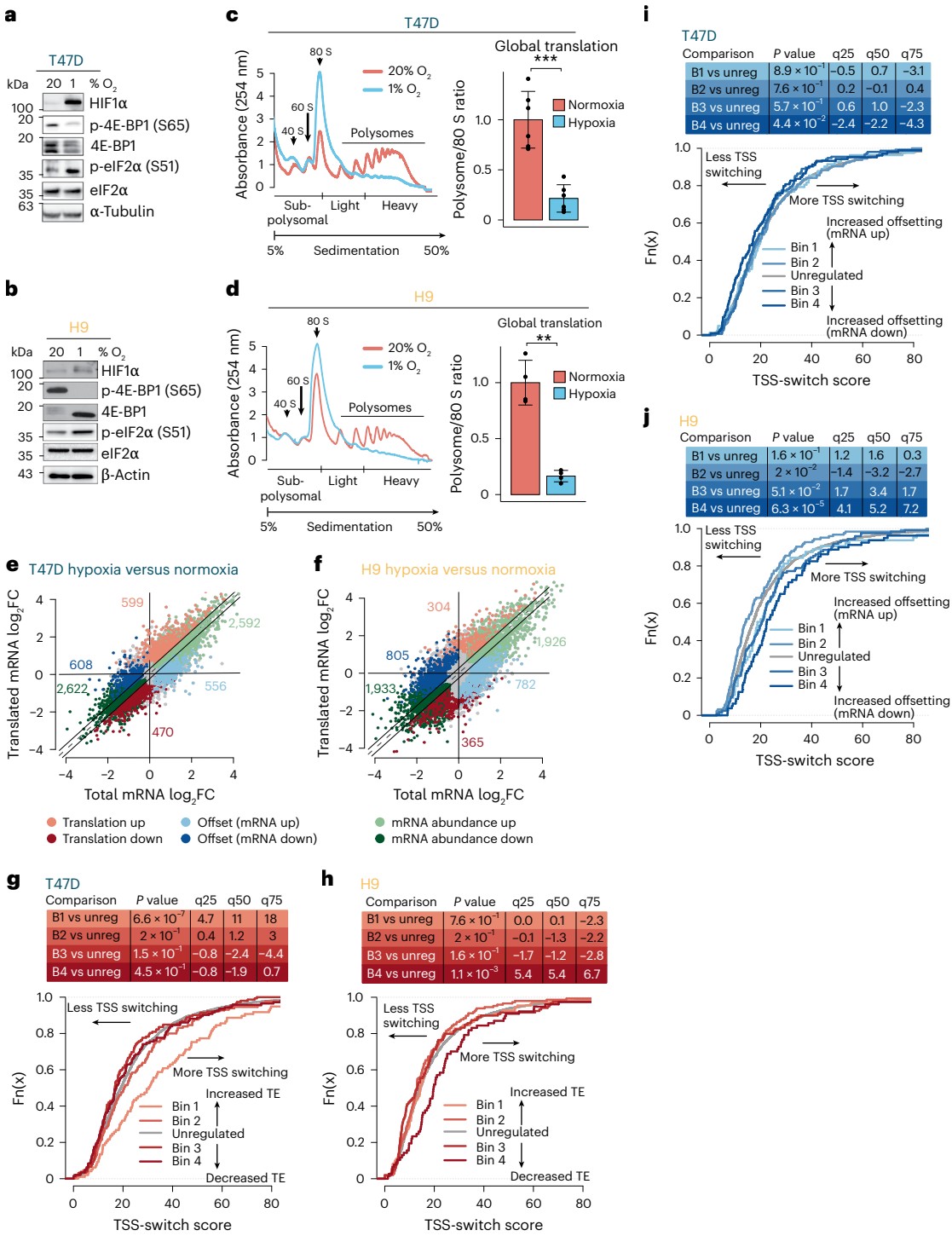

**Fig. 2 | TSS switching is associated with translational reprogramming under hypoxia. a,b**, Representative immunoblots of HIF1α, phosphorylated eIF2α (S51), and phosphorylated 4E-BP1 (S65) from T47D (**a**) and H9 (**b**) cells under normoxia and hypoxia (T47D $n = 3$, 48 h; H9 $n = 2$, 24 h; independent experiments). eIF2α, 4E-BP1, α-tubulin and β-actin were used as loading controls. **c,d**, Polysome tracing and global translation change in hypoxia-treated T47D (48 h) (**c**) and H9 (24 h) (**d**) cells compared to normoxia. Polysome-associated mRNAs are considered those associated with >3 ribosomes. Global change in translation was quantified as the ratio between the area under the curve for polysomes and 80S monosomes in each condition, after normalization. Mean polysome/80S ratios are displayed ± s.d. T47D $P = 0.0004$; H9 $P = 0.0025$; two-sided $t$-test (T47D $n = 6$; H9 $n = 4$; independent experiments). **e,f**, Scatter-plot of polysome-associated mRNA versus total mRNA log$_2$ fold changes in T47D (**e**) and H9 (**f**) cells (hypoxia versus normoxia). Genes are coloured according to the mode of regulation assigned by anota2seq[36] (FDR < 0.15). Number of regulated mRNA in each

category indicated in corresponding colours. **g,h**, The 400 most upregulated and downregulated genes were separated into four bins based on quartiles of the fold changes in translation efficiency (determined by anota2seq) for T47D (**g**) and H9 (**h**) cells. Empirical distribution functions compare the TSS-switch scores (determined by change-point analysis) across the four bins. The set of background genes (that is not in bins) is also indicated (grey line). Differences in TSS-switch scores between each bin compared to background were assessed using two-sided Wilcoxon rank-sum tests. $P$ values and the magnitude of shifts at quartiles (q25–75) are indicated. Right-shifted curves indicate the sets of translationally regulated genes with more extensive TSS switching under hypoxia compared to unregulated genes. **i,j**, The same as in **g** and **h**, but relating TSS-switching scores to translational offsetting in T47D (**i**) and H9 (**j**) cells. Significant shifts in TSS-switch scores between each bin compared to background were assessed using two-sided Wilcoxon rank-sum tests. $P$ values and the magnitude of shifts at q25–75 are indicated.

Beyond global suppression, transcript-selective changes in translation efficiency shape the newly synthesized hypoxic proteome. To identify these changes, we compared RNA sequencing of total and heavy polysome-associated mRNA (>3 ribosomes) using anota2seq[36]. In both cell types, ~2,000 genes showed non-congruent changes in total and polysome-associated mRNA (Fig. 2e,f and Supplementary Data 4). These included transcripts with altered polysome association without corresponding changes in total mRNA level ('translation') or those with changes in total mRNA levels that are offset by unaltered polysome association ('offsetting'). Notably, the 'translation' mode is expected to alter protein levels, whereas 'offsetting' opposes changes in protein levels despite altered mRNA abundance[37]. Notably, mRNAs encoding translational regulators were induced but translationally offset under hypoxia in both cell types (Extended Data Fig. 2a,b), including TOP mRNAs whose translation is mTOR-sensitive[23–25] (Extended Data Fig. 2c,d).

To assess whether 5′UTR remodelling associates with altered translation, we compared TSS-switch scores (Extended Data Fig. 1g) to translation changes (Fig. 2e,f). Higher TSS-switch scores associated with increased translation in T47D cells (Fig. 2g), and suppressed translation in H9 cells (Fig. 2h). While there was not a strong relationship between TSS switching and translational offsetting in T47D cells (Fig. 2i), increased switching was associated with offsetting for mRNAs with decreased levels in H9s (Fig. 2j). Together, these findings suggest that hypoxia-induced TSS switching impacts the translatome.

## Changes in the hypoxic translatome depend on multiple pathways and 5′UTR features

Translation is commonly regulated by interactions between 5′UTR features and the translational machinery during the rate-limiting initiation step[21,38]. In yeast, TSS switching can lead to gain or loss of uORFs in a subset of 5′UTRs, altering translation during meiosis or endoplasmic reticulum stress[39,40]. To examine how TSS switching contributes to translatome changes under hypoxia, we first identified 5′UTR features and pathways associated with hypoxia-induced translation changes (Fig. 3a; 'Model 1'), and then assessed the specific impact of TSS switching (Fig. 3a; 'Model 2'). We developed a method to identify variables explaining translation changes (post-transcriptional network modelling (postNet); Methods), generating networks of regulatory nodes (such as 5′UTR features, pathways regulating mRNA translation and TSS switching) with edges reflecting covariance between nodes (Fig. 3a–e).

For genes with altered translation efficiency under hypoxia, modelling included 22 established translational signatures (including pathways and factors such as mTOR, ISR, eIF4E2 (also known as 4EHP) and DAP5)[22,41–51], 5′UTR features derived from the most abundant isoform detected under hypoxia (length, GC content, folding energy, uORFs and known motifs) and de novo identified 5′UTR motifs (complete list of variables in Supplementary Data 4 and Fig. 3a). In T47D

cells, these variables explained 41.4% of hypoxia-induced translation changes (Fig. 3b) and 37.5% of offsetting (Fig. 3c). As expected[5,18,20], mTOR suppression and ISR activation were major contributors and were anti-correlated (some ISR-activated genes were translationally suppressed by mTOR[46] and vice versa) (Fig. 3b and Supplementary Data 4). Hypoxia-induced changes in translational offsetting were also partially mTOR- and ISR-dependent (Fig. 3c and Supplementary Data 4). Additionally, U34 tRNA modification signatures[49] explained changes in translation independently of ISR and mTOR, suggesting a previously unknown role in translational regulation under hypoxia. Of note, the eIF4E-dependent signature also explained hypoxia-dependent translation independently of mTOR (Fig. 3b).

Several 5′UTR features also associated with translatome changes under hypoxia (Fig. 3b,c and Supplementary Data 4). The 5′UTR GC content independently explained changes in translation and offsetting, with a higher GC being associated with suppressed translation under hypoxia (Fig. 3b,c and Extended Data Fig. 3a,b) and overlapping with mTOR translational signatures (Fig. 3b). Conversely, translation of longer, AU-rich, and uORF-containing 5′UTRs was enhanced upon mTOR inhibition and ISR induction, consistent with previous studies[23,46] (Fig. 3b). In agreement with the observed Gene Ontology enrichments (Extended Data Fig. 2a–c), the TOPscore[25] (summarizing TOP motifs across 5′UTR isoforms) best explained offsetting (Fig. 3c). Accordingly, this approach accurately captured known pathways and 5′UTR features mediating selective regulation of mRNA translation.

Beyond known regulatory features of 5′UTRs, we identified an AAGAAA motif associated with translational activation under hypoxia in T47D cells and correlated with mTOR-sensitive translation. Furthermore, an SGCSGCS (S = C/G) motif associated with translational offsetting. We also identified numerous additional 5′UTR motifs associated with altered translation and offsetting under hypoxia that did not co-vary (<10%) with known translational signatures (Fig. 3b,c and Extended Data Fig. 3a,b). Some of these motifs are predicted to interact with RNA binding proteins (RBPs) and are largely distinct between translation and offsetting modes of regulation (Supplementary Data 4). In addition, several were enriched in mRNAs encoding factors involved in WNT signalling, cell adhesion and angiogenesis (Supplementary Table 1).

In H9 cells, 50.1% and 41.8% of the variance in translation and offsetting, respectively, were explained using the abovementioned analysis (Extended Data Fig. 4a,b). As in T47D, mTOR suppression and ISR activation accounted for the greatest proportion of changes in translation efficiencies, alongside eIF4E phosphorylation, eIF4GI and DAP5 (Extended Data Fig. 4a,b). The 5′UTR motifs associated with modulated translation in T47D and H9 cells were largely distinct (Fig. 3b,c, Extended Data Figs. 4a,b and 5a,b and Supplementary Data 4), suggesting differing repertoires of *trans*-acting factors between cell types. Indeed, many RBPs with predicted binding to identified

**Fig. 3 | TSS switching alters regulatory 5′UTR features and shapes the hypoxia-induced translatome. a**, Schematic describing postNet translatome modelling. To identify features or pathways involved in hypoxic translatome remodelling, changes in translation efficiency or offsetting were modelled using known signatures of genes translationally regulated downstream of different pathways or factors (green), mRNA features of 5′UTRs characterized by nanoCAGE sequencing in hypoxia-treated T47D cells (purple) and de novo 5′UTR motifs (pink) (Model 1). The impact of TSS switching was then assessed by adding signatures describing 5′UTR alterations to modelling (Model 2). **b**, Network plot displaying the results of Model 1. Percentages of explained and unexplained variance in translation efficiency and contributions from each input category are indicated. Connections between features (nodes) indicate substantial correlations. Node colours indicate the mode of translational regulation under hypoxia the feature is associated with. **c**, The same as in **b**, but modelling changes in translational offsetting induced by hypoxia in T47D cells. **d**, Selection of the full network plot for Model 2, displaying the additional impact of adding

TSS switching signatures to Model 1 in explaining changes in translation efficiency under hypoxia in T47D cells. **e**, The same as in **d** explaining changes in translational offsetting under hypoxia in T47D cells. **f**, Scatter-plot comparing the TSS-switch score (indicative of altered 5′UTRs) versus the change in translation efficiency under hypoxia. Pearson's $r = 0.176$; $P = 4.64 \times 10^{-5}$, two-sided test. **g**, The same as in **f**, for translational offsetting. Pearson's $r = 0.108$; $P = 1.02 \times 10^{-2}$, two-sided test. **h**, Scatter-plot comparing changes in 5′UTR length (resulting from TSS switching) versus the change in translation efficiency under hypoxia. Pearson's $r = -0.143$; $P = 9.63 \times 10^{-4}$, two-sided test. **i**, Scatter-plot comparing 5′UTR shortening events versus the change in translational offsetting under hypoxia. Pearson's $r = 0.101$; $P = 1.69 \times 10^{-2}$, two-sided test. **j**, Bar plot indicating the number of transcripts that gain or lose 5′UTR elements identified in either Model 1 or Model 2 in T47D cells. Transcripts were considered if >10% of the expressed 5′UTR isoforms gained or lost the element. The translation mode associated with each element is indicated by coloured squares below.

5′UTR motifs were distinct between cell types, while others, such as SRSF1, were shared (Supplementary Data 4). In both cell types, identified 5′UTR motifs were enriched in mRNAs related to WNT signalling and cell migration, further suggesting that these processes may be translationally regulated under hypoxia (Supplementary Table 1).

Overall, we confirm that mTOR suppression and ISR activation play a pivotal role in hypoxic translatome remodelling, while implicating 5′UTRs features, including length, GC content, folding energy and uORFs. We also identified factors (for example, U34 tRNA modification and DAP5) and 5′UTR motifs (for example, AAGAAA and SGCSGCS)

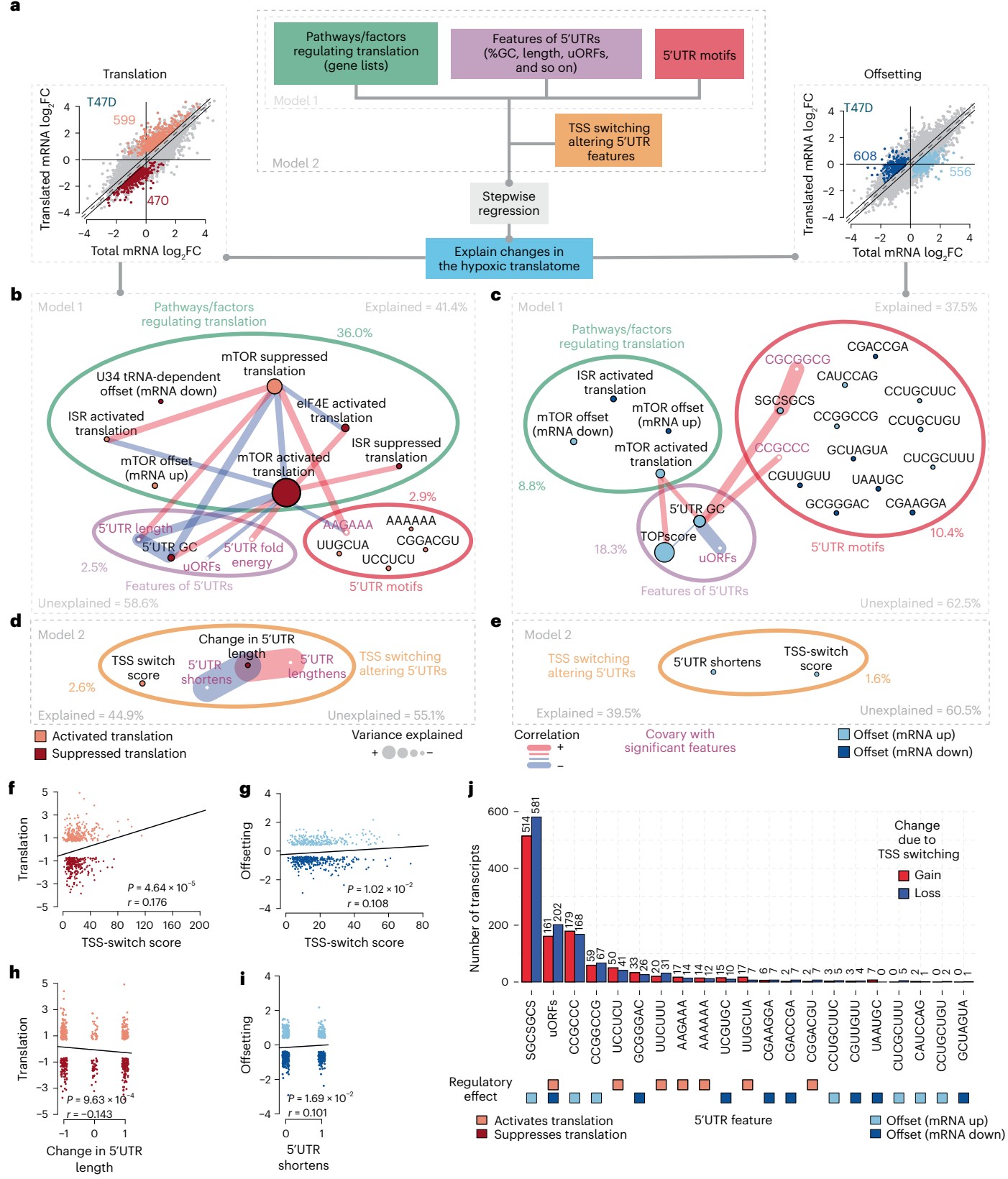

associated with changes in the hypoxic translatome that have not been previously reported, suggesting currently unexplored mechanisms of translational control in hypoxia.

## TSS switching alters regulatory 5′UTR features and shapes the hypoxia-induced translatome

We next examined whether TSS switching independently explains hypoxia-induced translatome changes ('Model 2'; Fig. 3a,d,e). In T47D cells, addition of 5′UTR remodelling increased the explained variance to 44.9% for translation and 39.5% for offsetting (Fig. 3d,e and full models in Extended Data Fig. 3c,d). Notably, alterations in 5′UTR length and the TSS-switch score independently explained changes in translation and offsetting (Fig. 3d,e). For translation, the independent contribution from TSS switching was comparable to that of the ISR and the combined independent effects of 5′UTR length, GC content, uORFs and folding energy (Fig. 3b–e and Supplementary Data 4). Higher TSS-switch scores (more extensive TSS switching) associated with enhanced translation efficiency (Figs. 2g and 3f), and translational offsetting (Fig. 3g). Furthermore, changes in 5′UTR length were also associated with alterations in both translation and offsetting (Fig. 3h,i). In H9 cells, including TSS switching increased explained variance to 51.4% for translation and 43.1% for offsetting (Extended Data Fig. 4c,d, full models in Extended Data Fig. 5c,d). Changes in TOPscore[25] between hypoxia and normoxia explained 2.1% of the variance in translation (Extended Data Figs. 4c,e and 5c), indicating that gain or loss of TOP motifs through TSS switching altered the H9 hypoxic translatome. Furthermore, 5′UTR lengthening was associated with translational offsetting of mRNAs with reduced levels under hypoxia (Extended Data Figs. 4d,f and 5d).

While specific 5′UTR features like TOP motifs or inhibitory uORFs impact translation efficiency with an expected directionality, TSS switching may both enhance or suppress translation through the loss of specific features in some transcripts and the gain in others. Our approach favours identification of features associated with directional changes in translation efficiency, and therefore likely underestimates the extent to which TSS switching impacts the translatome by modifying 5′UTR features. Accordingly, we evaluated how the identified 5′UTR motifs (Fig. 3b,c and Extended Data Fig. 4c,d) are impacted by TSS switching. In T47D cells, 514 transcripts gained and 581 lost one or more SGCSGCS motif (>10% change in motif-containing 5′UTR isoforms) (Fig. 3j). Similarly, 161 transcripts gained and 202 lost uORFs, contributing to both translational activation and suppression. In H9 cells, where 5′UTR lengthening was prevalent under hypoxia (Fig. 1k), TSS switching more often led to gain of 5′UTR regulatory elements (Extended Data Fig. 4g). For example, uORFs were

gained in 278 and lost in 79 transcripts, whereas the CCCUGC motif associated with translational suppression was gained in 130 and lost in 38 (Extended Data Fig. 4g). Other identified motifs also showed widespread gain or loss in hundreds of transcripts, supporting both activation and suppression of translation and offsetting.

Collectively, these findings suggest that hypoxia-induced changes in translation efficiency are driven by a myriad of 5′UTR features that can be altered via TSS switching, in parallel with remodelling of the translational machinery (for example, mTOR inhibition and ISR induction). TSS switching is therefore a previously unappreciated mechanism contributing to hypoxic translatome remodelling.

## Hypoxia-induced TSS switching is associated with altered H3K4me3 and changes in nucleosome context

We next interrogated mechanisms driving hypoxia-induced TSS switching. Previous work in VHL-null RCC4 renal cell carcinoma cells showed that TSS switching of HIF1α-target genes is linked to translational changes[31]. While we observed TSS switching for some transcriptional targets of HIF1α, a large proportion of TSS switching in T47D and H9 cells occurred without changes in transcript levels (Fig. 1d). Furthermore, less than 2% of the equivalent TSS-switching events between cell types were known HIF1α targets (Fig. 1m), suggesting that most hypoxia-induced TSS switching may be HIF1-independent. Hypoxia reshapes the epigenome, partly via inactivation of histone demethylases including KDM5A[6,8]. As H3K4me3 marks TSSs[9–11] and accumulates under hypoxia[6] (Fig. 4a), we examined whether H3K4me3 alterations correlate with TSS switching. H3K4me3-ChIP-seq (Extended Data Fig. 6a,b) revealed hypoxia-induced changes in H3K4me3 distribution around TSSs of 94% (98% in H9 at false discovery rate (FDR) < 0.01) of protein-coding genes detected by nanoCAGE (Fig. 4b and Supplementary Table 2). These changes fell into three categories: downstream or upstream shifts, and other alterations without a dominant directionality (for example, genes in Fig. 4c with TSS switching under hypoxia and Supplementary Data 1). The proportions of these H3K4me3 changes differed dramatically between cell types (Fig. 4d) and partly mirrored observed patterns of hypoxia-induced changes in 5′UTR length identified by change-point analysis (Fig. 1k), although significant TSS-switching events were associated with all categories of H3K4me3 changes in both cell types (Supplementary Data 1). These findings are consistent with the deposition of H3K4me3 under hypoxia being dependent on the existing epigenetic landscapes and therefore diverging between cell types.

We next examined whether TSS switching was associated with altered nucleosome context around TSSs. ATAC-seq in T47D cells under normoxia and hypoxia (Extended Data Fig. 6c), followed by NucleoATAC[52] analysis of nucleosome positioning and

**Fig. 4 | Hypoxia-induced TSS switching is associated with altered H3K4me3 and changes in nucleosome context. a**, Representative immunoblots of HIF1α and H3K4me3 following 0 h and 24 h of hypoxia (0.5% O₂) in T47D cells. H3 and α-tubulin were used as loading controls (*n* = 3 independent experiments). **b**, Changes in H3K4me3 distribution at TSSs were detected using a two-sided Kolmogorov–Smirnov test. The proportion of TSSs for protein-coding genes with significant changes (FDR < 0.01) is shown. The mean of replicates was used in all comparisons (*n* = 3 independent experiments). **c**, Examples of downstream, upstream, or other (for example broadening) changes in the distribution of H3K4me3 around the TSS that are accompanied by significant TSS switching under hypoxia in T47D cells. The Wilcoxon and Kolmogorov–Smirnov *P* values comparing the distributions are indicated, as well as the directionality and magnitude of the shifts in distribution at the quartiles. Lines and shading indicate the mean ± s.d. (*n* = 3 independent experiments). **d**, Summary of directional shifts in H3K4me3 distributions detected by two-sided Wilcoxon rank-sum test. Directionality was assigned by comparing shifts in the distribution between conditions at the quartiles of the H3K4me3 distribution for each locus. The proportion of TSSs for protein-coding genes with changes (FDR < 0.01) is shown. The mean of replicates was used in all comparisons (*n* = 3 independent

experiments). **e**, Nucleosome occupancy determined by NucleoATAC[52] analysis of ATAC-seq performed on T47D cells under hypoxia or normoxia (0.5% or 20% O₂, *n* = 3 independent experiments) (top). Occupancy signal is anchored around the most abundant nanoCAGE-determined TSSs under normoxia. Lines indicate the mean and shaded areas show the bootstrapped 95% CI (1,000 iterations). Adjusted two-sided Wilcoxon rank-sum *P* values between nucleosome occupancy signals across bins (bottom). Grey-shaded areas represent an FDR threshold of 0.05. **f**, Representative immunoblots of HIF1α and H3K4me3 following 24 h of 10 μM C48 or DMSO treatment in T47D cells. H3 and α-tubulin were used as loading controls (*n* = 4 independent experiments). **g**, Changes in the distribution of H3K4me3 marks at TSSs in C48-treated T47D cells, the same as in **b**. The mean of replicates was used in all comparisons (*n* = 4 independent experiments). **h**, Significant directional shifts in H3K4me3 distributions in C48-treated T47D cells, the same as in **d**. The mean of replicates was used in all comparisons (*n* = 4 independent experiments). **i**, Comparison of the directionality of significant changes in H3K4me3 distribution around TSSs between T47D cells treated with hypoxia versus normoxia and C48 versus DMSO. Changes were considered equivalent if the directionality of shifts in the distribution at quartiles were the same between treatment conditions.

occupancy showed that hypoxia-induced TSS switching coincided with decreased +1 and increased −2 nucleosome occupancy relative to normoxia (Fig. 4e). Although TSS-switching events were associated with changes in H3K4me3 distribution and distinct nucleosome remodelling, the hypoxic response is complex and impacts many cellular processes, including additional epigenetic alterations[6–8]. To isolate the role of H3K4me3, we treated T47D cells with the selective KDM5 inhibitor compound 48 (refs. 53–55) (C48; 10 μM; 24 h) (Extended Data Fig. 6d), which increased H3K4me3

similarly to 24 h of 0.5% $O_2$, but without stabilizing HIF1α (Fig. 4a,f). ChIP-seq (Extended Data Fig. 6e) revealed that C48 treatment significantly altered H3K4me3 distributions around the TSS of 52.9% of detected protein-coding genes (Fig. 4g), including downstream (17.6%) and upstream shifts (9.4%), and other alterations (73.0%) (Fig. 4h). Comparing C48 treatment to hypoxia, >59% of H3K4me3 alterations had the same directionality (downstream, upstream or other) (Fig. 4i), indicating that KDM5 inhibition partially reproduced hypoxia-induced H3K4me3 remodelling.

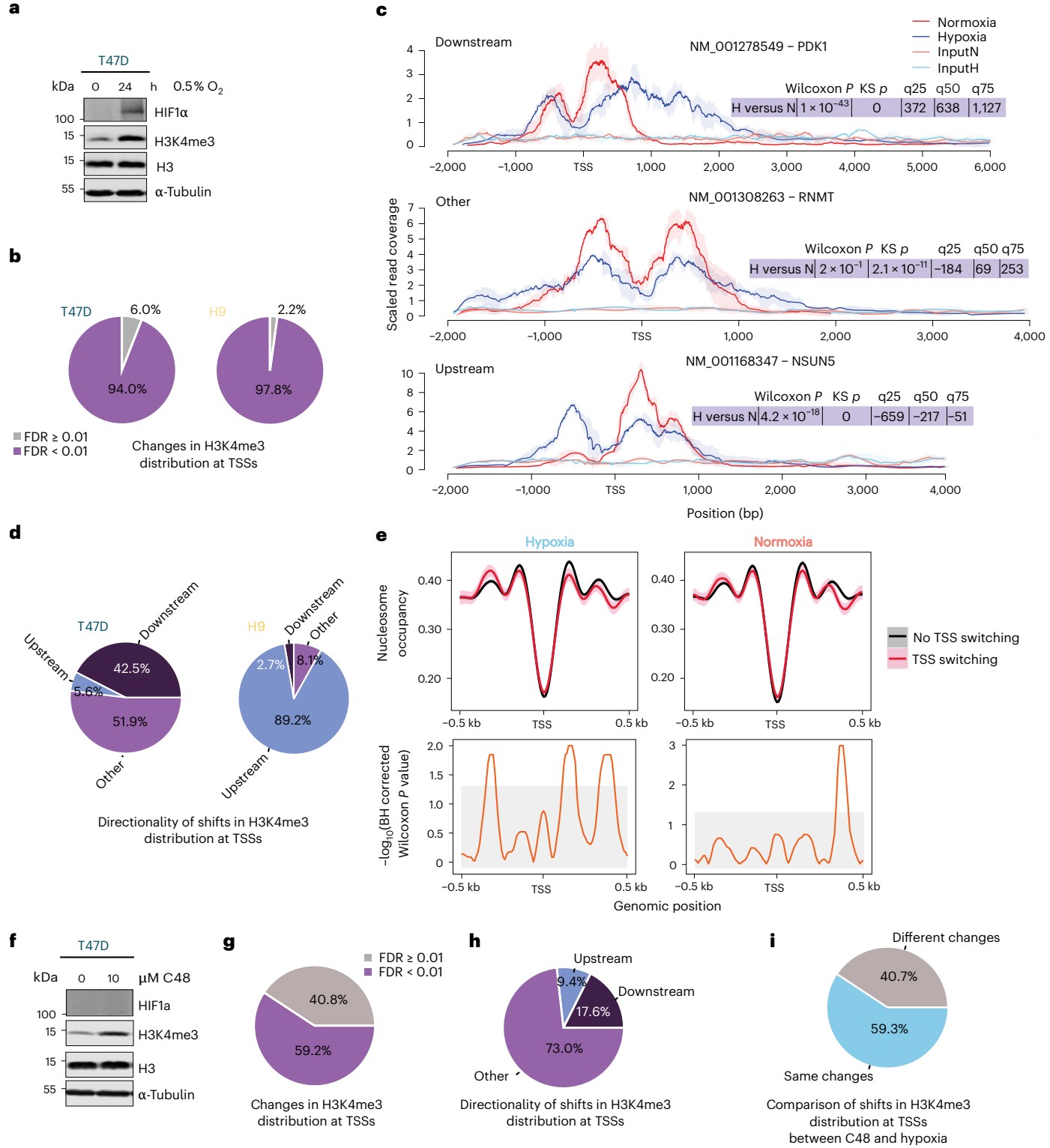

## Inhibition of KDM5 induces TSS switching that remodels 5'UTRs and is associated with proteome changes

To clarify the role of H3K4me3 in hypoxia-induced TSS switching, we performed nanoCAGE sequencing (Extended Data Fig. 7a) in C48 or vehicle (dimethylsulfoxide; DMSO)-treated T47D cells (Fig. 5a,b), which identified TSS switching for >3,000 transcripts (Fig. 5c,d and Supplementary Data 1). Notably, C48 treatment did not alter mTOR or ISR signalling (Fig. 5a). TSS-switch scores were lower with C48 than hypoxia, suggesting that additional mechanisms may tune the magnitude of isoform switching under hypoxia (Extended Data Fig. 7b). A comparison of 5'UTR sequence segment enrichments from change-point analysis showed that 682 transcripts had the same enrichments between C48 treatment and hypoxia (28% of hypoxia-associated TSS switching; Supplementary Data 3). A Monte Carlo simulation showed this was a greater proportion than expected by chance (Fig. 5e). Of these, 32% (hypoxia) and 79% (C48) showed no change in mRNA levels (Fig. 5f). Therefore, modulation of H3K4me3 caused by KDM5 inhibition is sufficient to alter TSS selection independent of hypoxic transcriptional programmes.

As requiring that all sequence segment enrichments are equivalent between hypoxia and C48-induced TSS switching is stringent and may underestimate similarity, we also examined categorical changes. C48-induced 5'UTR isoform enrichments (Fig. 5g) resembled those observed under hypoxia (Fig. 1k). Among transcripts enriched for the longest or shortest 5'UTR isoforms under hypoxia, 42.4% and 28.4% showed the same enrichment with C48 treatment (Fig. 5h). Overall, 30.1% of hypoxia-induced TSS switching was partially or fully mirrored with C48 treatment. For example, change-point analysis of *SFXN3* mRNA identified a relative loss of longer (>436 nt) and increase of shorter 5'UTR isoforms (<436 nt) under both conditions (Fig. 5i). Similarly, TSS switching increased levels of longer 5'UTR isoforms (>246 nt) for *RCAN3* in both treatment conditions (Fig. 5j). Accordingly, KDM5 inhibition induces extensive TSS switching, recapitulating a significant proportion of TSS alterations observed under hypoxia.

To assess whether this altered pool of 5'UTR isoforms impacts the proteome under C48 treatment, we performed GPF-DIA proteomics in parallel with RNA-seq on the same C48 and DMSO-treated cells used to profile TSSs (Extended Data Fig. 7c,d). Unlike hypoxia, C48 did not alter mTOR or ISR signalling, as shown by unchanged phosphorylation

of S6 and eIF2α (Fig. 5a). Using anota2seq[36], we identified 352 proteins (FDR < 0.15) altered without corresponding changes in mRNA levels following C48 treatment (Fig. 5k,l). These proteins were enriched for Gene Ontologies including extracellular vesicles, protein modifications and response to hypoxia, among others (Extended Data Fig. 7e). As with hypoxia (Fig. 2g), higher TSS-switch scores following C48 treatment were associated with increased protein levels, independent of mRNA levels (Fig. 5m).

Together, these findings demonstrate that KDM5 inhibition alone is sufficient to induce TSS switching that recapitulates a subset of hypoxia-induced switching and contributes to proteome alterations. Therefore, TSS switching-dependent remodelling of the 5'UTRome affects the proteome independently of changes in mRNA levels, and in the absence of alterations in mTOR or ISR signalling.

## Inhibiting H3K4me3 accumulation under hypoxia blocks TSS switching and decreases cellular fitness

We next examined whether reducing H3K4me3 accumulation affects hypoxia-induced TSS switching. T47D cells were pretreated with DMSO or OICR-9429 (25 μM, 48 h), an inhibitor of the interaction between WDR5 and MLL-associated COMPASS H3K4 methyltransferases[56], followed by 48 h of hypoxia (0.5% O₂). OICR-9429 attenuated hypoxia-induced H3K4me3 accumulation (Fig. 6a,b). NanoCAGE mapping of TSSs (Extended Data Fig. 7f) revealed >5,000 transcripts with TSS switching (FDR < 0.15) when comparing hypoxia plus OICR-9429 versus hypoxia alone. Of these, 618 showed sequence segment enrichments opposite to those observed under hypoxia, exceeding what is expected by chance (Fig. 6c–e). Unlike C48, treatment with OICR-9429 under hypoxia predominantly enriched expression of shorter 5'UTR isoforms (Fig. 6f), reversing hypoxia-associated enrichment in 57.6% of the longest and 26.0% of the shortest isoform-expressing transcripts (Fig. 6g). For example, hypoxia suppressed expression of shorter 5'UTR mRNA isoforms of *TOP3A*. However, this expression was restored upon addition of OICR-9429 (Fig. 6h). Comparing the subsets of hypoxia-induced TSS-switching events recapitulated by KDM5 inhibition and blocked by OICR-9429 under hypoxia, we identified 268 transcripts for which TSS selection seems H3K4me3-dependent (Fig. 6i). This subset was enriched for Gene Ontologies such as

---

**Fig. 5 | Inhibition of KDM5 induces TSS switching that remodels 5'UTRs and is associated with proteome changes. a**, Representative immunoblots of H3K4me3, and phosphorylated eIF2α (S51) and S6 (S240/244) from T47D cells treated with 10 μM C48 (24 h) or DMSO (0 h). H3, eIF2α, S6 and α-tubulin were used as loading controls (*n* = 4 independent experiments). **b**, Densitometry of H3K4me3 normalized to α-tubulin and H3 loading controls. Bars indicate mean ± s.d. *P* = 0.00018; two-sided *t*-test (*n* = 4 independent experiments). **c**, Kernel density estimation *P* value and FDR distributions for differential TSS usage between C48 and DMSO-treated T47D cells (*n* = 3,287). Dotted grey line indicates an FDR threshold of 0.15. **d**, Bar plot of protein-coding transcripts with significantly altered TSS usage between C48 and DMSO treatments (FDR < 0.15) in T47D cells. **e**, TSS-switching events under hypoxia with lower FDRs are more likely to be recapitulated by C48 treatment. The FDR range for TSS switching under hypoxia was divided into ventiles and the proportion of equivalent changes in 5'UTR isoforms between hypoxia versus normoxia and C48 versus DMSO comparisons was determined. Dashed line marks the proportion of changes expected to be the same by chance (21.8%, estimated by Monte Carlo simulation). The red line represents smoothed linear regression, with 95% CI shaded. **f**, Pie charts indicating the proportions of transcripts with the same changes in 5'UTR isoform expression under hypoxia versus C48 treatment (*n* = 682) with significantly (FDR < 0.15) increased (up), decreased (down) or unchanged overall expression between hypoxia and normoxia (left), and between C48 and DMSO (right). **g**, Bar plot of categories of 5'UTR sequences enriched in transcripts with significant TSS switching after C48 treatment, identified by change-point analysis (*n* = 3,287). **h**, Bar plot of categories of 5'UTR sequences enriched for transcripts with significant TSS switching between hypoxia and normoxia, identified by change-point analysis. The percentage

recapitulated with C48 treatment is indicated. **i**, 5'UTR isoform expression for NM_001388028 (*SFXN3* mRNA) in hypoxia and normoxia-treated (top), and C48 and DMSO-treated T47D cells (bottom). Change-point analysis (dotted orange lines) identified enriched and depleted 5'UTR segments in both comparisons (right). Both hypoxia and C48 treatment enrich shorter 5'UTR isoforms. **j**, The same as in **i**, but for NM_001251977 (*RCAN3* mRNA). Both hypoxia and C48 treatment enrich longer 5'UTR isoforms. **k**, Scatter-plot of protein (from GPF-DIA proteomics analysis) versus total mRNA log₂ fold changes in T47D cells (C48 versus DMSO, *n* = 4 independent experiments). Genes are coloured according to the mode of regulation assigned by anota2seq (FDR < 0.15). Protein up and down represent changes in protein level occurring independently of changes in mRNA level. Abundance up and down represent congruent changes in protein and mRNA levels. The number of regulated genes in each category is indicated in corresponding colours. **l**, Kernel density estimation *P* value and FDR distributions for anota2seq analysis (in **k**) of changes in protein, total mRNA and protein adjusted for mRNA between C48 and DMSO-treated T47D cells. **m**, The 400 most up- and downregulated genes were separated into 4 bins based on the quartiles of the fold changes in protein levels adjusted for mRNA levels (determined by anota2seq) in T47D cells treated with C48. Empirical distribution functions compare the TSS-switch scores (determined by change-point analysis) across the four bins. The set of background genes (that is not in bins) is also indicated (grey line). Differences in TSS scores between each bin compared to background were assessed using two-sided Wilcoxon rank-sum tests. *P* values and the magnitude of shifts at quartiles (q25–75) are indicated. Right-shifted curves indicate the sets of genes with more extensive TSS switching after C48 treatment compared to unregulated genes.

protein modifications, cell adhesion and cellular ion homeostasis (Extended Data Fig. 7g), and included *TNPO3*, where hypoxia and C48 treatment depleted longer 5′UTR mRNA isoforms, whereas addition of OICR-9429 blocked this effect (Fig. 6j).

Finally, to test whether epigenetically mediated TSS switching affects cellular fitness under hypoxia, we measured T47D proliferation under hypoxia, co-treated with vehicle (DMSO) or OICR-9429. Addition

of OICR-9429, which blocked ~30% of hypoxia-induced TSS switching, significantly decreased cell proliferation under hypoxia, relative to the same treatment under normoxia (Fig. 6k). Together, these findings indicate that hypoxia-induced H3K4me3 alterations drive a substantial proportion of TSS-switching events, identifying a previously unknown role for this epigenetic modification in 5′UTR-determining TSS selection that contributes to cellular adaptations under hypoxia.

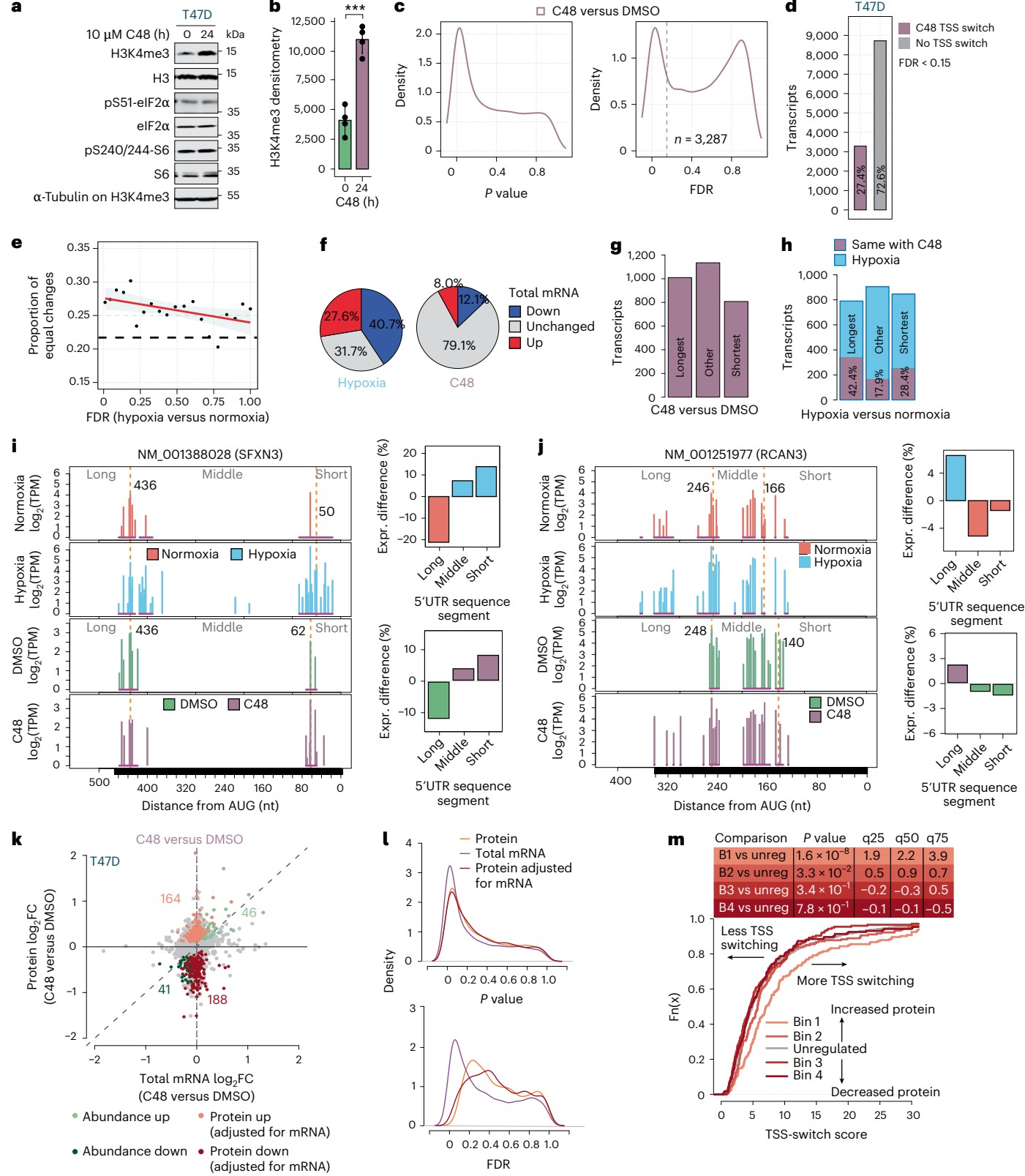

## TSS switching orchestrates adaptation to hypoxia by regulating availability of differentially translated mRNA isoforms

We next examined whether TSS switching regulates specific biological processes. Gene Ontology analysis of shared (Extended Data Fig. 1j) and cell-type-specific hypoxia-induced TSS-switching events revealed enrichment of metabolism-related terms (Extended Data Fig. 8a). To survive hypoxia, cells switch from oxidative phosphorylation to glycolysis[57]. Of note, many glycolytic enzymes, including PDK1, underwent significant TSS switching in hypoxic T47D cells (Extended Data Fig. 8b). PDK1 was among the most significant TSS-switching events in both cell types (Fig. 7a, Extended Data Fig. 8c and Supplementary Data 1). Under hypoxia, PDK1 phosphorylates pyruvate dehydrogenase (PDH) preventing pyruvate entry into the citric acid cycle, conserving it for other metabolic processes, including its $NAD^+$-regenerating reduction to lactate for subsequent rounds of glycolysis[57]. *PDK1* transcription is induced by HIF1α, and it is post-translationally activated by ATP, NADH and CoA and inactivated by ADP, $NAD^+$, pyruvate and CoA-SH[58]. As expected, hypoxia increased PDK1 protein and transcript levels, and PDH1 S232 phosphorylation (Fig. 7b,c). Change-point analysis revealed loss of longer (>87 nt) *PDK1* 5′UTR mRNA isoform expression and enrichment of shorter isoforms in hypoxic T47D cells (Fig. 7d). We confirmed expression of the hypoxia-inducible short isoforms using 5′RACE (Fig. 7a) and isoform-selective quantitative PCR with reverse transcription (RT–qPCR) (Fig. 7e). In the more glycolytic H9 cells[2,59], more *PDK1* 5′UTR mRNA isoforms were detected at baseline (Extended Data Fig. 8c). Nevertheless, change points and 5′UTR sequence segment enrichments were highly similar (Extended Data Fig. 8d), suggesting that hypoxia-induced TSS switching may be an important regulator of PDK1 expression.

To assess whether translation of *PDK1* 5′UTR mRNA isoforms depends on oxygen availability, we monitored their distribution across a sucrose gradient under normoxia or hypoxia (Fig. 7f and Extended Data Fig. 8e). In both conditions, shorter (<45 nt) *PDK1* 5′UTR isoforms were enriched in heavier polysomes relative to longer (>78 nt) isoforms (Fig. 7e), indicating higher translation efficiency, regardless of oxygen tension. As RT–qPCR primers could not allow full separation of *PDK1* 5′UTR isoforms, we further confirmed these findings using dual-luciferase reporter assays in HEK293T cells (Fig. 7g and Extended Data Fig. 8f,g). As expected[2,5,22], hypoxia globally reduced translation driven by the *ACTB* 5′UTR and all 5′UTR isoforms of *PDK1*, while sustaining translation of the *ATF4* 5′UTR reporter.

However, translation of the shorter (57 and 36 nt) *PDK1* 5′UTR isoforms was higher than the longer (134 nt) isoform in both conditions (Fig. 7g). Similar results were observed with mTOR inhibition (INK128) and ISR activation (thapsigargin; TG) (Fig. 7h and Extended Data Fig. 8h,i), reinforcing that the hypoxia-enriched shorter 5′UTR isoforms of *PDK1* mRNA are more efficiently translated both under normal and stress conditions. To further establish the impact of TSS switching on *PDK1* mRNA translation, we expressed the 134, 57 and 36 nt 5′UTR isoforms followed by CRISPR-mediated knockout of the endogenous *PDK1* gene (Extended Data Fig. 9a). Cells expressing the 36-nt 5′UTR isoform produced more PDK1 protein relative to mRNA levels, both under hypoxia and normoxia (Fig. 7i,j and Extended Data Fig. 9b,c), consistent with higher translation efficiency. Furthermore, actinomycin D chase showed no notable differences in stability between 134-nt and 36-nt 5′UTR isoforms under hypoxia (Extended Data Fig. 9d–f), confirming that differential TSS usage, and not altered mRNA stability, drives enrichment of the shorter *PDK1* 5′UTR mRNA isoform.

To examine the impact of *PDK1* TSS switching under hypoxia, T47D cells expressing individual *PDK1* 5′UTRs were subjected to [$^{13}C$] pyruvate labelling followed by stable isotope tracing analysis by gas chromatography–mass spectroscopy (GC–MS) after 24 h in hypoxia (0.5% $O_2$) or normoxia (20% $O_2$). In cells expressing the short but not the long 5′UTR isoforms, $^{13}C$-pyruvate tracing into lactate (lactate m + 3) was increased under hypoxia (Fig. 7k and Extended Data Fig. 9g,h). As expected, $^{13}C$-pyruvate tracing into citrate (citrate m + 2) was diminished across all lines under hypoxia relative to normoxia (Extended Data Fig. 9h–j), with a modest reduction in conversion to alanine (alanine m + 3) observed in hypoxia upon expression of the longer isoform (134 nt) (Extended Data Fig. 9h,k,l). These results suggest that cells expressing the shorter, more efficiently translated, 5′UTR mRNA isoform of *PDK1* display enhanced lactate production under hypoxia.

Finally, we examined TSS switching for *PDK1* after C48 treatment. Although TSS switching differed somewhat from hypoxia, possibly due to lack of HIF1 transcriptional activation, there was enrichment of shorter, more efficiently translated 5′UTR isoforms (Extended Data Fig. 9m,n). PDK1 protein levels increased after C48 treatment without changes in total mRNA levels (Extended Data Fig. 9o), consistent with more efficient translation of the shorter 5′UTR isoforms. Under hypoxia, shorter *PDK1* 5′UTR isoform expression was accompanied by extension of H3K4me3 around the TSS, a 25-base upstream shift of the +1 nucleosome dyad, and increased −2 nucleosome occupancy

---

**Fig. 6 | Inhibiting H3K4me3 accumulation under hypoxia blocks TSS switching and decreases cellular fitness. a**, Representative immunoblots of H3K4me3 from T47D cells pre-treated with 25 µM OICR-9429 or DMSO for 48 h and treated with hypoxia or normoxia, with the addition of 25 µM OICR-9429 or DMSO for an additional 48 h. H3 and α-tubulin were used as loading controls, and HIF1α as a positive control for hypoxia (*n* = 3 independent experiments). **b**, Densitometry of H3K4me3 normalized to α-tubulin and H3 loading controls. Bars indicate mean ± s.d. *P* = 0.01; two-sided *t*-test (*n* = 3 independent experiments). **c**, Kernel density estimation *P* value and FDR distributions for differential TSS usage in T47D cells between co-treatments of hypoxia and OICR-9429 or DMSO (*n* = 5,016). Dotted grey line indicates an FDR threshold of 0.15. **d**, Bar plot indicating the proportions of protein-coding transcripts with significantly altered TSS usage between T47D cells co-treated with hypoxia and OICR-9429 versus DMSO (FDR < 0.15). **e**, TSS-switching events under hypoxia with lower FDRs are more likely to be reversed by OICR-9429 treatment. The FDR range for TSS switching under hypoxia was divided into ventiles and the proportion of opposite changes in 5′UTR isoforms in hypoxia + OICR-9429 versus hypoxia + DMSO comparisons was determined. Dashed line marks the proportion of changes expected to be the same by chance (22.9%, estimated by Monte Carlo simulation). Red line represents smoothed linear regression, with 95% CI shaded. **f**, Bar plot of categories of 5′UTR sequences enriched in transcripts with significant TSS switching after co-treatment with hypoxia and OICR-9429, identified by change-point analysis (*n* = 5,016). **g**, Bar plot of

categories of 5′UTR sequences enriched for transcripts with significant TSS switching between hypoxia and normoxia. The percentage reversed with OICR-9429 is indicated. In total, 786 (31.1%) of change-point-defined 5′UTR categorical changes under hypoxia were reversed by OICR-9429. **h**, Quantification of 5′UTR isoforms of NM_004618 (*TOP3A* mRNA) in hypoxia and normoxia-treated (top) and hypoxia + DMSO or OICR-9429-treated T47D cells (bottom). Change-point analysis (dotted orange lines) identified enriched and depleted 5′UTR segments in both comparisons (right). OICR-9429 restores expression of shorter 5′UTR isoforms that were lost under hypoxia. **i**, Venn diagram showing the overlap of transcripts where hypoxia-induced TSS switching was recapitulated by C48 treatment and reversed by OICR-9429 treatment in T47D cells. **j**, Quantification of 5′UTR isoforms of NM_001191028 (*TNPO3* mRNA) in hypoxia and normoxia-treated (top), C48 and DMSO-treated (middle) and hypoxia-treated T47D cells co-treated with either DMSO or OICR-9429 (bottom). Change-point analysis (dotted orange lines) identified enriched and depleted 5′UTR segments in all three comparisons (right). C48 treatment recapitulates the enrichment in expression of shorter 5′UTR isoforms that occurs under hypoxia, whereas OICR-9429 reverses this effect. **k**, Trypan blue exclusion assays to quantify viable cells under hypoxia. T47D cells treated the same as in **a** were counted at the end point. Displayed is the mean of the ratio of cell count in hypoxia versus normoxia in OICR-9429 or DMSO-treated cells ± s.d. *P* = 0.012; two-sided *t*-test (*n* = 6 independent experiments).

(Fig. 8a). These changes were mirrored by C48 treatment, where H3K4me3 modestly extended around the TSS. NucleoATAC[52] analysis of ATAC-seq (Extended Data Fig. 9p) on the same C48-treated cells revealed a similar 36-base upstream shift of the +1 nucleosome dyad, and increased −2 nucleosome occupancy (Fig. 8a) in the absence of HIF1 induction (Fig. 4f) or altered *PDK1* mRNA levels (Extended Data Fig. 9o). These findings suggest that modulation of H3K4me3 results in changes to nucleosome conformations and altered selection of TSSs.

Overall, this proposes a model where adaptive translational responses driving cellular phenotypes can be coordinated from the level of chromatin modifications, altering TSS selection and remodelling the 5′UTRome.

## Discussion

Our findings reveal that hypoxia reprogrammes gene expression through coordination of epigenetic, transcriptional and translational

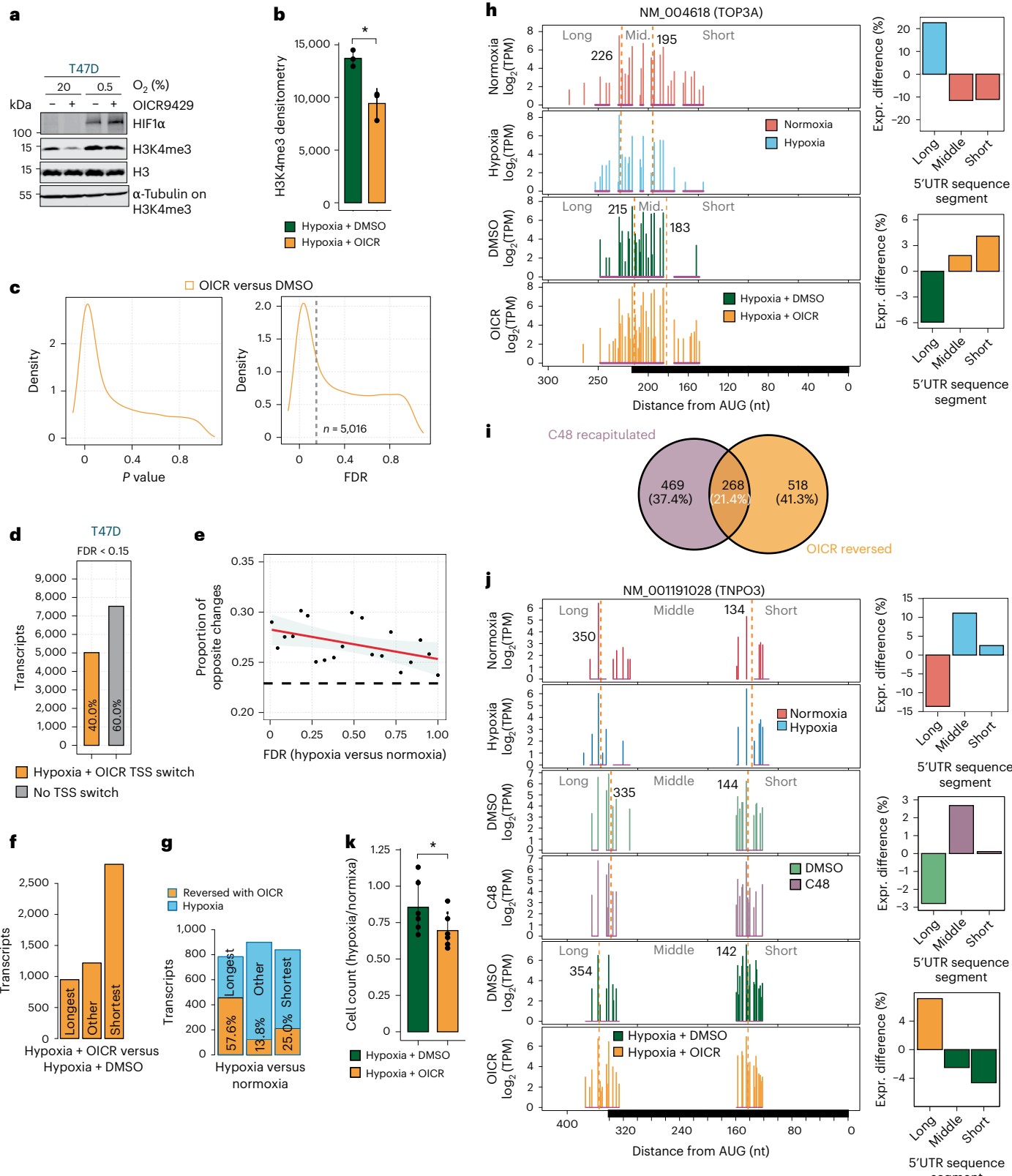

programs. Central to this process is widespread TSS switching, which altered 5′UTRs of thousands of transcripts, modulating potential for interactions with the translation initiation machinery. These isoform-level changes were recapitulated by inhibiting the H3K4me3 eraser KDM5, and blocked by impeding MLL-associated COMPASS methyltransferase complexes, linking H3K4me3 to TSS selection. This hypoxia-induced 5′UTR landscape occurs in parallel with translational apparatus reprogramming, together promoting a survival-enhancing adaptive translational programme. This included a shift towards shorter, more efficiently translated *PDK1* 5′UTR mRNA isoforms under hypoxia that facilitated the switch towards glycolytic metabolism. Disrupting TSS switching reduced cellular fitness under hypoxia, underscoring its functional importance. Our results support a mechanism of gene expression control whereby chromatin modifications can direct alternate TSS selection, generating 5′UTR isoforms with distinct translation efficiencies (Fig. 8b).

Over 50 years ago, it was proposed that mRNA translation may be selectively modulated by altering the availability of translation initiation machinery components[60]. This can largely be explained by 5′UTR features that distinguish translationally 'strong' mRNAs that outcompete 'weak' mRNAs for recruitment to the translation initiation apparatus. In this context, our results suggest that H3K4me3-dependent TSS selection under hypoxia adjusts the abundance of 'strong' versus 'weak' 5′UTR isoforms to remodel the proteome.

Most genes have multiple TSSs, with differential use linked to tissue-specificity and disease states[61–63]. Our previous work identified several stem cell factors with distinct TSSs giving rise to 5′UTR isoforms with different translation efficiencies under hypoxia[26]. Hypoxia also altered promoter usage in colon cancer cells for 191 genes[64]. Our analysis focused on TSS switching often within single promoters, without altering the resulting proteoform. This revealed that hypoxia-induced TSS switching is far more prevalent than previously appreciated, impacting over 20% of protein-coding transcripts across

highly divergent cell types. This high-resolution approach captures positional 5′UTR regulatory elements, such as TOP motifs, where small shifts can drastically alter translation of corresponding mRNAs[25]. Indeed, we found that TSS switching-driven changes in TOP motifs significantly impacted translational reprogramming in hypoxic H9 cells. Even modest changes in 5′UTR length were significantly associated with altered translation under hypoxia, demonstrating the surprising impact of this regulatory mechanism.

Dynamic TSS switching generating differentially translated 5′UTRs has been observed during meiosis and endoplasmic reticulum stress in yeast[39,40]. This mechanism is driven by factors that initiate transcription from upstream TSSs, producing 5′UTRs containing inhibitory uORFs. Similarly, we also observed the gain or loss of uORFs and altered translation efficiency, suggesting this mechanism may also operate in mammalian cells responding to hypoxia. However, only around 14% (T47D) and 10% (H9) of hypoxia-induced TSS switching altered the presence of uORFs in 5′UTRs, indicating that the vast majority of the TSS switching reported here impacts translation through distinct mechanisms.

While hypoxia suppresses global protein synthesis via mTOR inhibition and ISR activation[5], mechanisms governing selective translation of survival-promoting transcripts remain less understood. We propose that epigenetically mediated remodelling of the 5′UTRome contributes to this selectivity, potentially acting in concert with additional upstream regulators, including U34 tRNA modifications[49], DAP5[42] and alterations in eIF4E phosphorylation[45]. We further identified previously uncharacterized 5′UTR motifs impacted by TSS switching and associated with hypoxia-induced activation and suppression of translation, warranting future studies.

Previous work showed HIF-dependent TSS switching in RCC4 cells, impacting translation[31]. Here, although known HIF1 targets like *PDK1* exhibited TSS switching, most TSS switching occurred for genes not regulated by HIF1. Many of these changes could be reproduced solely by modulating H3K4me3, in the absence of HIF1 stabilization or changes in

---

**Fig. 7 | TSS switching orchestrates adaptation to hypoxia by regulating availability of differentially translated mRNA isoforms. a**, Quantification of 5′UTR isoforms for NM_001278549 (*PDK1* mRNA) in hypoxia and normoxia-treated T47D cells. Same outline as Fig. 1e,f but also indicating change-point-identified sequence segments (dotted orange lines) and isoforms detected by 5′RACE in hypoxia-treated T47D cells (black arrows). **b**, Representative immunoblot showing PDK1, PDHE1α and phosphorylated PDH (S323/293) with increasing time under hypoxia treatment in T47D cells. HIF1α is used as a positive control for hypoxia (*n* = 3 independent experiments). **c**, Difference in total *PDK1* transcript expression. Mean ± s.d.; *P* = 0.0015; two-sided *t*-test (*n* = 3 hypoxia, *n* = 2 normoxia independent experiments). **d**, Change-point analysis of NM_001278549 identifies change points at 36 and 87 nt upstream of the start codon (left). Shorter sequence segment isoforms are enriched under hypoxia and the longer segment is depleted (right). **e**, Quantification of *PDK1* 5′UTR isoform expression relative to total transcript in hypoxia- and normoxia-treated T47D cells using RT–qPCR. *P* value reflects the isoform–treatment interaction from a linear model (analysis of variance; ANOVA) with the design % isoform expression - replicate + isoform + treatment + isoform–treatment, evaluating whether the pattern of isoform expression differs between conditions. Residual degrees of freedom = 20; mean ± s.d.; *P* = 1.7 × 10⁻⁹ (*n* = 5 independent experiments). **f**, Polysome occupancy of *PDK1* 5′UTR isoforms under hypoxia (left) and normoxia (right). RNA was isolated from sucrose fractions separated by polysome fractionation and subjected to RT–qPCR. The proportion of PDK1 5′UTR isoforms measured by RT–qPCR in sub-polysomal, light (2–4) and heavy (>4) polysome fractions is shown, where the sum of all fractions for each mRNA is set to 100%. *P* values reflect the isoform–treatment interaction from a linear model (ANOVA) with the design % expression - isoform + fraction + isoform–fraction, evaluating whether the pattern of isoform abundance differs between fractions. Residual degrees of freedom = 6; mean ± s.d.; hypoxia *P* = 0.0015; normoxia *P* = 0.0081 (*n* = 2 independent experiments). **g**, Schematic of m⁷G-capped bicistronic reporter mRNA harbouring the 5′UTR of *PDK1* (134, 57 or 36 nt) or the 5′UTR of *ACTB* or *ATF4*, upstream of the firefly luciferase

(FLuc) ORF and HCV IRES upstream of *Renilla* luciferase (Rluc) ORF (top). Firefly relative to *Renilla* luciferase signal in normoxia- and hypoxia-treated 293T cells transfected with the reporter mRNA (bottom). *P* values result from a linear model (ANOVA) with the design log₂(normalized luminescence) - replicate + isoform + treatment + isoform–treatment. Residual degrees of freedom = 8; mean ± s.d.; isoform, *P* = 2.55 × 10⁻⁵, for differences between translation of *PDK1* 5′UTR isoforms; treatment, *P* = 2.18 × 10⁻³, for differences in translation depending on treatment; isoform–treatment, *P* = 0.23, testing whether isoforms are differentially translated between treatments (*n* = 3 for 134 and 57 nt, *n* = 2 for 36 nt; independent experiments). **h**, Firefly luciferase values relative to *Renilla* luciferase in DMSO, TG (400 nM) and INK128 (INK; 50 nM) treated 293T transfected with the reporter mRNA containing *PDK1* 5′UTR isoforms. *P* values are provided for the linear model (ANOVA) with the same design as in **g**. Residual degrees of freedom = 21; mean ± s.d.; isoform, *P* = 5.86 × 10⁻⁶; treatment, *P* = 5.81 × 10⁻⁸; isoform–treatment, *P* = 0.15 (*n* = 4 for 57 and 36 nt, *n* = 3 for 134 nt; independent experiments). **i**, Representative immunoblot showing PDK1 and phosphorylated PDH (S323 and 293) under hypoxia treatment (24 h) in T47D cells with knockout (KO) of endogenous *PDK1* and re-expression of individual *PDK1* 5′UTR isoforms. Two independent clones are shown per isoform. PDHE1α is used as a loading control and HIF1α is used as a positive control for hypoxia (*n* = 2 independent experiments). **j**, The ratio of PDK1 protein and transcript levels in T47D cells with KO of endogenous *PDK1* and re-expression of individual 5′UTRs after 24 h of hypoxia treatment. *P* values result from a linear model (ANOVA) with the design protein - transcript + isoform + treatment + replicate + isoform–treatment. Residual degrees of freedom = 16; mean ± s.d.; isoform, *P* = 7.88 × 10⁻⁵; treatment, *P* = 0.09; isoform–treatment, *P* = 0.67 (*n* = 2 independent experiments). **k**, The ratio of labelled lactate m + 3 to pyruvate m + 3 from T47D cells with KO of endogenous *PDK1* and re-expression of individual *PDK1* 5′UTR isoforms grown in hypoxia or normoxia for 24 h. Metabolites were determined using stable isotope tracing by GC–MS. Mean ± s.d.; *P* = 0.6678 for 134 nt, *P* = 0.01877 for 36 nt; two-sided *t*-test (*n* = 3 independent experiments).

mTOR or ISR signalling. While the role of H3K4me3 in gene expression remains debated[9,13], it has been shown to participate in anchoring transcriptional machinery to nucleosomes[65] and to activate transcription at silenced loci[66]. In contrast, other studies have shown that H3K4me3 is not required for transcription to occur, with global depletion having limited effect[9,13,67]. Here modulating H3K4me3 through both KDM5 inhibition and inhibition of MLL-containing COMPASS methyltransferases[68] extensively altered TSS usage, most often without changes in overall mRNA abundance. This suggests that H3K4me3 may play an important role in precise and dynamic TSS selection under different cellular conditions. These changes coincided with shifts in nucleosome

occupancy and positioning, suggesting H3K4me3 modifications may drive TSS selection concomitant with changes in nucleosome conformations. We observed associations between directional shifts in H3K4me3 distributions and the selection of TSSs resulting in longer or shorter 5'UTRs, and there is evidence that pre-existing chromatin states likely play an important role in determining cell-type-specific stress-responsive TSS switching. However, there is much left to learn about the specific mechanisms that govern these processes in hypoxia.

While our focus was on translation initiation under hypoxia, translation elongation may also be impacted, for example through modulation of eEF2K activity[69] and/or methylation of eEF1A[70], meriting

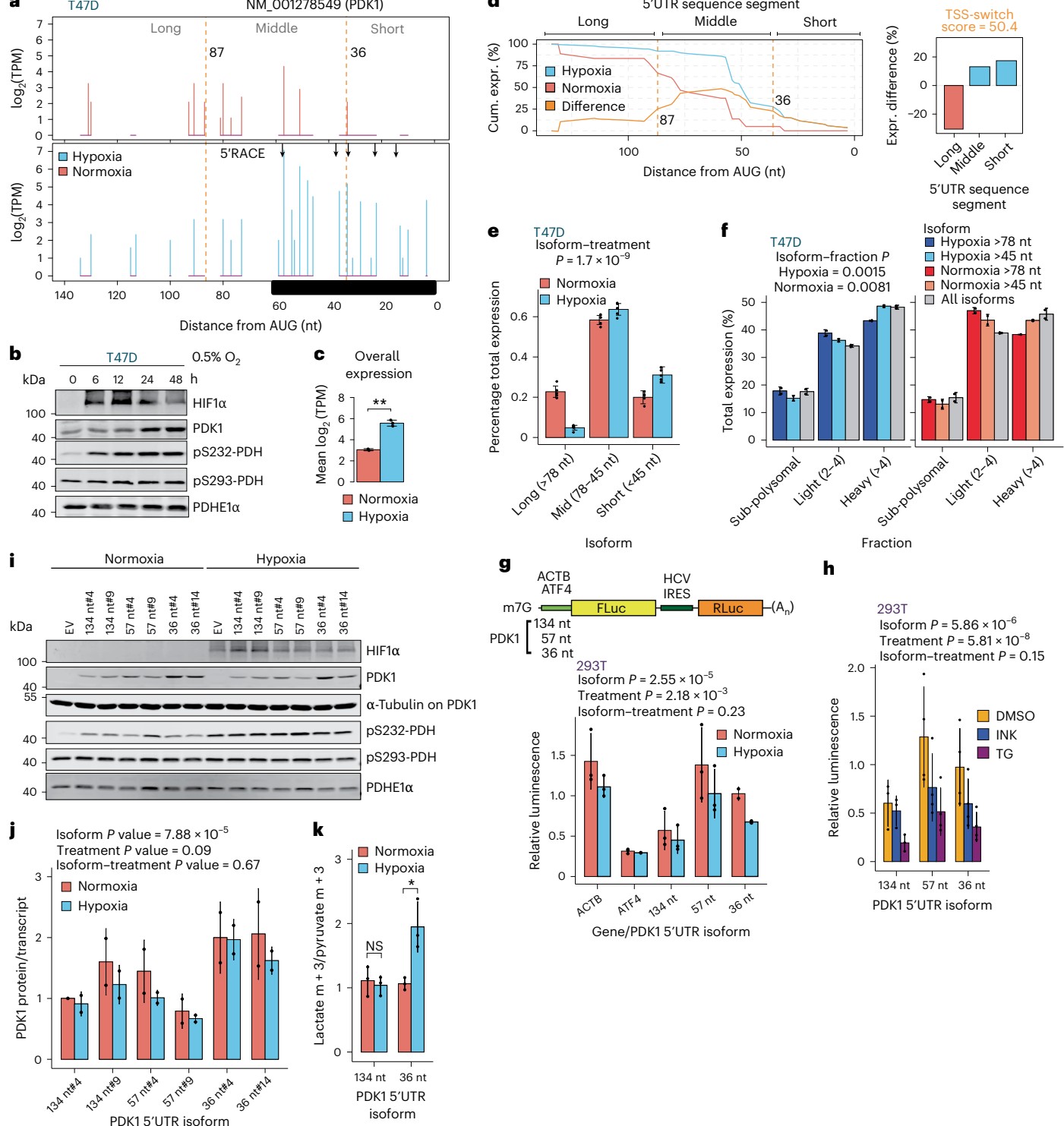

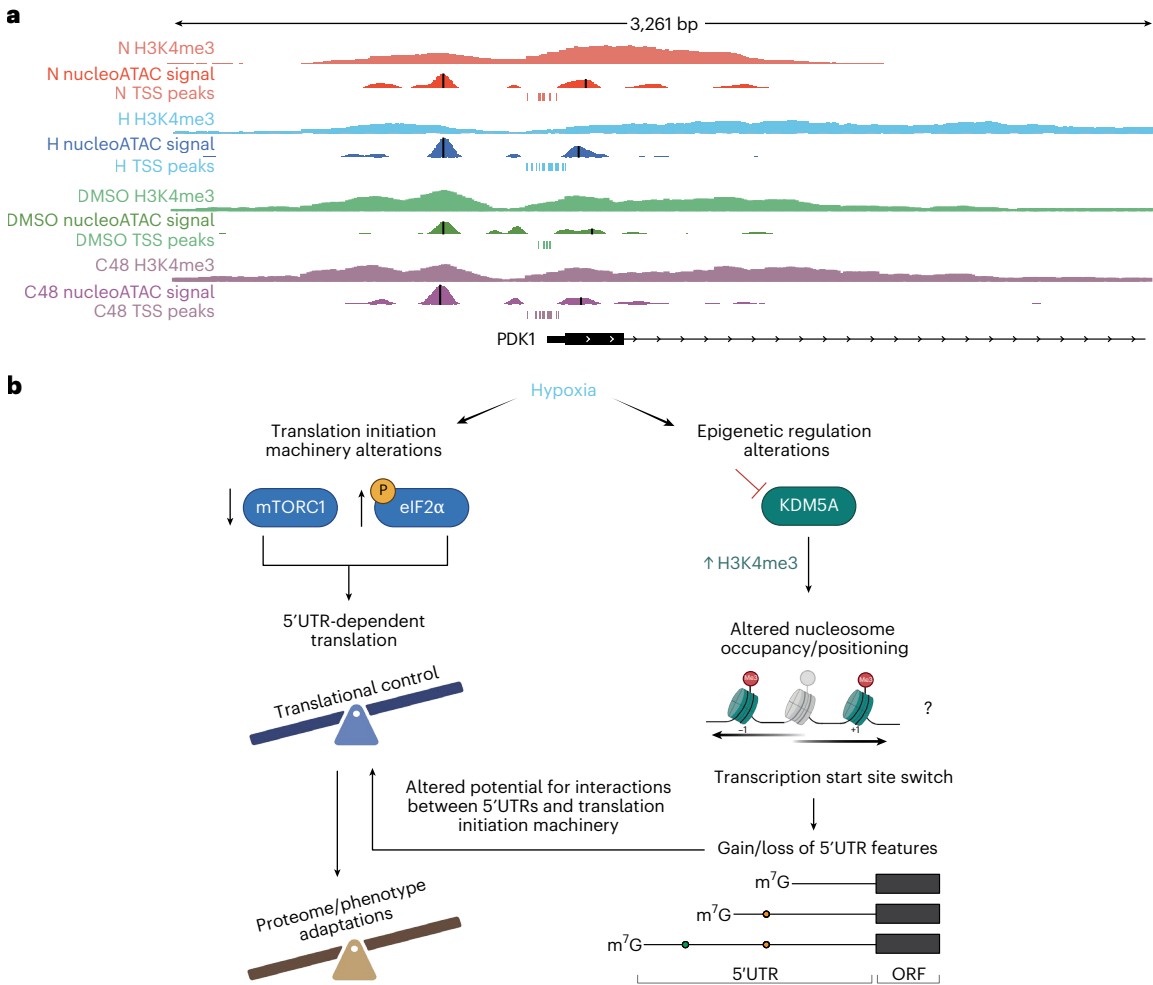

**Fig. 8 | Epigenetic alterations facilitate transcriptional and translational programmes in hypoxia. a**, H3K4me3, nucleosome occupancy, and TSS peaks for NM_001278549 mRNA isoform of *PDK1* under hypoxia and normoxia, and with DMSO and C48 treatments. H3K4me3 was measured by ChIP-seq, and reproducible TSS peaks measured by nanoCAGE are indicated. Nucleosome occupancy is displayed as smoothed NucleoATAC signal. Vertical black bars indicate the dyad positions of nucleosomes determined using NucleoATAC. **b**, Hypoxia induces reprogramming of the translational machinery largely through inhibition of mTORC1 and activation of the ISR via increased phosphorylation of eIF2α. These events together lead to global suppression of mRNA translation, and activation of transcript-selective translation that promotes proteome adaptations to hypoxia and associated phenotypes. Concurrently, hypoxia leads to remodelling of H3K4me3 due to loss of oxygen-dependent activity of KDM5A. This alteration of H3K4me3 around TSSs leads to changes in nucleosome occupancy and positioning, and TSS switching for a subset of genes. This extensive TSS switching alters the composition of regulatory mRNA features in 5′UTRs, changing their potential for interactions with the translation initiation machinery. The epigenetically mediated TSS switching produces a pool of 5′UTR isoforms that, in concert with the hypoxic translational machinery, help to drive changes in the proteome that are fundamental to cellular adaptations to hypoxia. Image in **b** created with BioRender.com.

further investigation. We also focused only on TSS switching linked to gene-level changes in translation efficiency or alterations in the proteome. However, extensive TSS switching altering 5′UTR isoforms occurred for genes that seemed to be unregulated under hypoxia, suggesting a mechanism of isoform-level offsetting that may help to maintain protein levels. This observation raises interesting questions regarding the role of TSS switching in proteome homeostasis in the context of cellular stress, and warrants future studies.

In conclusion, we described a mechanism where adaptive translational responses under hypoxia are facilitated by epigenetically mediated control of TSS usage, modulating the pool of 5′UTR isoforms. These findings further a paradigm in which cellular stress is sensed at the level of chromatin to direct adaptive remodelling of the translatome.

## Online content

Any methods, additional references, Nature Portfolio reporting summaries, source data, extended data, supplementary information,

acknowledgements, peer review information; details of author contributions and competing interests; and statements of data and code availability are available at https://doi.org/10.1038/s41556-025-01786-8.

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

## Methods

### Primer sequences and antibodies

Full details of primer sequences and antibodies are provided in Supplementary Table 3.

### Cell lines

T47D and HEK293T (ATCC HTB-133 and CRL-3216) cells were cultured in RPMI-1640 or DMEM (Gibco 11875093 and 10566-024), respectively, with 10% FBS (Corning), authenticated (SickKids Research Institute) and routinely tested for mycoplasma (ATCC 30-1012 K). H9 hES cells (WiCell WA09), approved for use by the Stem Cell Oversight Committee of Canada, were cultured on Corning Matrigel hES Cell-Qualified Matrix (Stemcell Technologies, 07181) in mTeSR1 feeder-free medium supplemented with mTeSR1 5X Supplement (Stemcell Technologies, 85850) and passaged using the StemPro EZPassage Disposable Stem Cell Passaging Tool (Thermo Scientific, 23181010). Colonies were inspected daily, and spontaneously differentiated colonies were manually removed. All cells were maintained at 37 °C and 5% $CO_2$ in a humidified environment.

### Cell culture experiments

T47D cells were plated 24 h before experiments. For hypoxia experiments, the medium was replaced with fresh RPMI with 10% fetal bovine serum (FBS) and cells were incubated under normoxia or hypoxia (0.5% $O_2$ unless otherwise stated, in Biospherix OxyCycler model C42 or ProOx model C21 chambers) for the indicated durations. For C48 (Axon Medchem, 2809/batch 1) treatments, the medium was replaced with fresh RPMI with 10% FBS containing 10 µM C48 or DMSO vehicle control. Dose–response experiments used a 1:10 dilution series starting at 10 µM C48. For C48 ChIP experiments, cells were treated with DMSO for 24 h, followed by 10 µM C48 for 24 h. For OICR-9429 (Selleckchem, S7833) co-treatments, cells were treated with 25 µM OICR-9429 or DMSO for 48 h. The medium containing small molecules was then refreshed and cells were incubated under hypoxia or normoxia for an additional 48 h.

### NanoCAGE library preparation and data preprocessing

NanoCAGE libraries were generated as described by Poulain et al.[71] with modifications. In brief, 100 ng RNA in 1 µl was mixed with 1 µl 2.5 mM dNTPs (TaKaRa, 4030) and 1 µl of a mastermix containing sorbitol/trehalose (2.64/0.53 M; Sigma, S1876 and T9531), 10 µM MS-RanN6 primer and 100 µM equimolar mixture of two template-switching oligonucleotides with 8-nt random unique molecular identifiers (UMIs) and one of six barcodes (ACAGAT, GTATGA, ATCGTG, GAGTGA, TATAGC and GCTGCA). This mixture was heat-denatured (65 °C for 10 min) and cooled (4 °C for 2 min) followed by reverse transcription with 2 µl 5× first-strand buffer, 0.25 µl RNaseOUT, 1 µl SuperScript IV, 1 µl 0.1 M dithiothreitol (DTT) (Life Technologies, 10777019 and 18090200) and 1.5 µl 5 M betaine (Sigma, B0300) (22 °C for 10 min, 55 °C for 30 min, 75 °C for 15 min and 4 °C hold). Duplicate reactions were pooled and cDNA was purified using a 1:1 ratio of AMPure XP beads (Beckman Coulter, A63881) and eluted in 30 µl $H_2O$.

Diagnostic qPCR was performed using 1.5 µl of cDNA and 100 nM MsDir1R and 1F primers per reaction, with TB Green Premix Ex Taq (TaKaRa RR420A). Cycle threshold (Ct) values were used as $n$ cycles in subsequent semi-suppressive PCR (ssPCR). For ssPCR, 20 µl cDNA was mixed with 25 µl KAPA HIFI HotStart Ready Mix (Roche KK2602), 0.5 µl MsDir1F and MSDir1R primers (10 µM) and 4 µl $H_2O$ (98 °C for 3 min, $n$ cycles of 98 °C for 20 s, 65 °C for 15 s, 72 °C for 2 min and a final 72 °C for 2 min before 4 °C hold). Products were purified using a 1:0.6 ratio of AMPure XP beads, and eluted in 25 µl $H_2O$.

Tagmentation was performed using 500 pg cDNA with 2× Tagment DNA Buffer, 3 µl Amplicon Tagment Mix (Illumina, FC-131-1096) and 1 µl PEG (40% w/w) at 55 °C for 10 min. Samples were immediately chilled and incubated with 2.5 µl of NT Buffer (Illumina, FC-131-1096)

at room temperature for 5 min. Fragments were PCR-amplified with 1 µl nanoCAGE S-series (10 µM) and Nextera XT N-Series Index primers (N7xx, 10 µM), 7.5 µl Nextera PCR Mastermix (Illumina, FC-131-1096) and 3 µl $H_2O$ (72 °C for 3 min, 95 °C for 30 s, 12 cycles of 95 °C for 10 s, 55 °C for 30 s, 72 °C for 1 min, before a final 72 °C for 5 min and 4 °C hold). Libraries were purified using a 1:0.6 ratio of AMPure XP beads, and eluted in 15 µl $H_2O$. Concentrations (Qubit, Thermo Scientific, Q32854) and size distributions were evaluated (BioAnalyzer, Agilent, 5067-4626) at each step. Final library concentrations were adjusted to 10 nM before pooling.

Sequencing was performed at the SciLifeLab NGI facility (Stockholm). Hypoxia and normoxia libraries were sequenced using the HiSeq2500 (Illumina) with a 100-bp single-end setup. Bcl-to-fastQ conversion was carried out using bcl2fastq (v.2.19). Libraries from C48 and OICR-9429 experiments were sequenced using the NovaSeq6000 (Illumina) with the same read setup and bcl2fastq (v.2.20.0.422).

NanoCAGE data preprocessing was performed as implemented previously[23,49]. In brief, barcoded reads were extracted using TagDust (v.2.33)[72], and 3′ adaptors were trimmed using Cutadapt (v.1.18)[73] (settings, -e 0.15 -O 1 -n 4 -m 25). PCR duplicates (identical UMIs and first 25 bp) were removed. Ribosomal RNA was removed using BBDuk (BBTools; v.36.59)[74] with default settings. Reads were aligned to GRCh38/hg38 (RefSeq release 109) using Bowtie (v.1.2.2)[75] (settings, -a -m 1 –best –strata -n 2 -l 28), and reads that uniquely aligned or failed to align were collected and aligned using Bowtie (settings, -a –best –strata -n 3 -l 25) to a custom index of 5′UTR sequences created by extending RefSeq 5′UTR sequences by 78 bases upstream using genomic sequence, and 78 bases downstream using mRNA sequences (reflecting the maximum nanoCAGE read length after adaptor trimming). Reads with 5′ alignment mismatches at the first or second positions were trimmed by 1–2 bases, and strand invasion artifacts were removed using the Tang et al.[76] Perl script (settings, -e 2). See Extended Data Fig. 1a for a preprocessing summary.

### Analysis of differential TSS usage

TSS peaks were defined as the 5′ end mapping positions of reads. Peaks present in fewer than $n$ – 1 replicates per condition were excluded. For C48 and OICR-9429 datasets where more peaks were detected, only those observed in all samples per condition were retained. Transcripts with fewer than ten reads were removed, and library complexity was assessed by sampling increasing numbers of reads (100,000 read increments) and counting unique peaks (>1 read) and transcripts detected (Extended Data Fig. 1b).

TSS peak counts were normalized for library size (tags per million; TPMs), and low-expression peaks (<25% of the mean TPMs per transcript) were removed. TSS clusters were defined using a dynamic sliding window approach with a window size of 5 nt, where clusters began at the position of the first TSS peak and extended until there were no peaks within the window. For 5′UTRs >200 nt, the sliding window scaled to increase by 2.5% of the longest isoform length, as differences in 5′UTR length may be more impactful for short versus long isoforms. Overlapping clusters were unified across conditions, and reproducible, expression-filtered TSS peaks within cluster regions were quantified per sample.

For 5′UTR isoforms detected in all replicates of both conditions ('quantitative' comparisons), TSS cluster counts were normalized using voom (limma[77] v.3.48.3). Significant changes in the relative expression of 5′UTR isoforms between conditions were identified using per-gene linear models with the design:

$$Ex_{nano} \sim \gamma_i + \gamma_c + \gamma_r + \gamma_i \times \gamma_c + \varepsilon$$

where $Ex_{nano}$ is the expression for each 5′UTR isoform across all conditions and replicates, $\gamma_i$ denotes the relationship with 5′UTR isoform, $\gamma_c$ is the relationship to the experimental condition, $\gamma_r$ is the relationship

to replicate, $\gamma_i \times \gamma_c$ is the interaction between $\gamma_i$ and $\gamma_c$, and $\varepsilon$ is the residual error. Similar to anota2seq[36], a random variance model was then applied[78]. A low $P$ value for the $\gamma_i \times \gamma_c$ interaction term indicates a significant difference in 5′UTR isoform expression depending on condition, independent of changes in overall transcript expression or replicate effects. For 5′UTR isoforms not detected in all samples, regression was not appropriate and a 'qualitative' approach was used. For 5′UTR isoforms detected in at least two replicates per condition, a Fisher's exact test was used to determine significant differences in expression relative to all other isoforms between conditions (using the sums of reads per condition across all replicates). For 5′UTR isoforms detected in only one condition, significance was assessed using probabilities. In the condition expressing the unique isoform the probability of expressing the shared isoforms relative to all isoforms was first calculated. In the condition not expressing the unique isoform, the probability of expressing only the shared isoform was then calculated:

$$p_{sj} = \left( \frac{r_{si}}{r_{ui} + r_{si}} \right)^{r_{sj}}$$

where $r$ is the number of reads summed across all replicates, $s$ denotes shared isoforms, $u$ denotes the uniquely expressed isoform, $i$ the condition expressing the unique isoform and $j$ the condition only expressing shared isoforms ($r_{uj} = 0$).

All $P$ values from the different arms of the statistical analysis were collected and adjusted for multiple testing using the Benjamini–Hochberg method. TSS-switching events were defined as those with FDR < 0.15.

## Change-point analysis to identify enriched or depleted 5′UTR sequences from TSS switching

For transcripts with significant TSS switching, TPM-normalized read counts were averaged across replicates, per condition, to calculate cumulative expression at each TSS position. Control distributions (normoxia, DMSO and hypoxia + DMSO) were subtracted from treatment conditions (hypoxia, C48 and hypoxia + OICR-9429) to obtain the difference in cumulative expression at each TSS position. Change points in these difference distributions, indicating shifts in TSS usage, were identified using the 'cpt.mean' function from the changepoint R package[29] (v.2.2.3) with penalty, 'AIC', method, 'SegNeigh' and Q = 3. The longest 5′UTR isoform was segmented at the detected change points into three regions, and enrichment or depletion of segments as a result of TSS switching was quantified as the difference in the proportion of total TSS peak expression between conditions. TSS-switch scores for each transcript were calculated as the maximum difference in expression across 5′UTR sequence segments, with higher scores indicating greater 5′UTR remodelling resulting from a given treatment condition (Fig. 1g–i).

## Gene Ontology enrichment analysis

Gene Ontology (GO) enrichment analysis was performed using the ClueGO[79] (v.2.5.8) plug-in in Cytoscape[80] (v.3.8.2). The background was defined as all genes passing expression thresholds in the given dataset. Enrichments were selected based on the following criteria: FDR cutoff of 0.05; Correction Method Used = Benjamini–Hochberg; Statistical Test Used = Enrichment (right-sided hypergeometric test); κ = 0.4; Min. Percentage = 10; Min GO Level = 3; Max GO Level = 8; Number of Genes = 5; GO Fusion = false; GO Group = true; Over View Term = SmallestPValue; Group By κ Statistics = true; Initial Group Size = 1; Sharing Group Percentage = 50.0; Ontology Used = GO_BiologicalProcess-EBI-UniProt-GOA-ACAP; Evidence codes used = All; and Identifiers used = SymbolID.

## Immunoblot analysis

Cells were washed on ice with ice-cold PBS, lysed in RIPA buffer containing 1× Halt Protease and Phosphatase Inhibitor (Thermo Scientific, 78445), and cleared by centrifugation (17,000$g$). Lysates for histone immunoblots were sonicated (one pulse and 50% amplitude) before centrifugation. Protein concentration was measured (Micro BCA Protein Assay kit; Thermo Scientific, 23235) and equal amounts of protein were separated by PAGE and transferred onto 0.45-μm nitrocellulose membranes (Bio-Rad, 1620115). Membranes were blocked with blocking buffer (LI-COR, 927-70001), incubated with primary antibody in antibody dilution buffer (LI-COR, 927-65001) overnight at 4 °C, washed in TBST and incubated with secondary antibody (IRDye 800CW or IRDye 680RD) for 1 h at room temperature. Blots were imaged on a LI-COR Odyssey CLx imaging system and quantified with ImageStudio software (v.5.2.0). All antibodies and dilutions are listed in Supplementary Table 3.

## Polysome profiles

T47D or H9 cells ($8 \times 10^6$ cells per condition) were seeded in 15-cm plates 24 h before treatment. The medium was replaced and cells were incubated under normoxia or hypoxia (0.5% $O_2$) for 24 or 48 h. As described previously[81], cells were pretreated with 100 μg ml$^{-1}$ cycloheximide (Sigma, C7698) for 5 min, scraped in ice-cold, cycloheximide-supplemented (100 μg ml$^{-1}$) PBS and pelleted (240$g$ for 5 min at 4 °C). Lysis was performed on ice in 500 μl hypotonic lysis buffer (5 mM Tris-HCl, pH 7.5, 2.5 mM MgCl$_2$, 1.5 mM KCl, 100 μg ml$^{-1}$ cycloheximide, 1 mM DTT, 0.5% Triton and 0.5% sodium deoxycholate) for 15 min. Lysates were cleared (20,817$g$ for 15 min at 4 °C) and adjusted to OD$_{260}$ of 10–20 and 10% was reserved as input. Linear sucrose gradients (5–50%) were generated using a gradient maker (Biocomp Gradient Master 108) and lysates were layered over the top and centrifuged at 222,228$g$ for 2 h at 4 °C (Beckman Coulter, Optima XPN-80, rotor SW41Ti). Gradients were fractionated using a density gradient fractionation system (Brandel, BR-188-177).

## Smart-seq2 preparation and data preprocessing

RNA-seq libraries from total and polysome-associated mRNA were prepared using the Smart-seq2 protocol[82]. In brief, RNA quantity and quality were assessed using Qubit (Thermo Scientific, Q32855) and Bioanalyzer (Agilent, 5067-1511) and 10 ng of RNA per sample was used as input. Following pre-amplification, 70 pg of cDNA underwent tagmentation in a total volume of 20 μl using the Nextera XT kit (Illumina, FC-131-1096). Sequencing was performed at the SciLifeLab NGI facility (Stockholm) on a NovaSeq6000 with a 50-bp-end setup. Bcl-to-fastQ conversion used bcl2fastq (v.2.20.0.422). Adaptors were trimmed using BBDuk[74] with settings $k = 13$; ktrim = $n$; useshortkmers = $t$; mink = 5; qtrim = $t$; trimq = 10; and minlength = 25. Ribosomal RNA was also removed using BBDuk[74] with default settings. Reads were aligned to GRCh38/hg38 (RefSeq, release 109) using HISAT2 (ref. 83) (v.2.1.0; settings –no-mixed –no-discordant) and uniquely mapping reads were quantified using the featureCounts function (RSubread[84] v.2.6.4) with RefSeq gene definitions and default settings.

## Analysis of differential translation

Raw counts from genes with at least one read across all samples were normalized using the trimmed mean of M values method[85] and log$_2$ transformed, and data reproducibility was assessed using PCA with genes in the top quartile of the s.d. across samples.

Changes in polysome-associated and cytosolic mRNA under hypoxia were analysed using anota2seq[36] (v.1.14.0). Batch effects were accounted for by including sample replicates in the models. Default threshold settings were used (minSlopeTranslation = −1; maxSlopeTranslation = 2; minSlopeBuffering = −2; maxSlopeBuffering = 1; deltaPT = deltaP = deltaTP = deltaT = log$_2$(1.2)) with a significance threshold maxRvmPAdj = 0.15 (FDR < 0.15). Transcripts were classified into three modes of regulation (changes in mRNA abundance, translation efficiency and translational offsetting) using the 'anota2seqRegModes' function.

## Translatome analysis with post-transcriptional network modelling

Translatome modelling was performed using a custom R package, postNet. Input data included significant per-gene fold changes in translation efficiency or offsetting between hypoxia and normoxia (from anota2seq) and a list of signatures describing regulatory variables for each gene (Supplementary Data 4). The postNet 'featureIntegration' function was used with arguments regOnly = TRUE; regulationGen = 'translation' or 'buffering'; allFeat = TRUE; useCorel = TRUE; analysis_type = 'lm'; covarFilt = 8; and NetModelSel = 'Omnibus'. In brief, the software performs three steps. First, univariate linear models identify significant associations between translational regulation and individual variables. Second, stepwise regression is performed beginning with the variable best explaining changes in translational regulation from step one. Variables are added to the model iteratively, retaining those that improve the ability to explain variance in translational regulation, both ranking variables by importance and revealing covariance between them. Last, covariance is removed allowing calculation of the percentage of variance in translation or offsetting explained by a given variable, adjusted for all others.

For Fig. 3b,c and Extended Data Figs. 3a,b, 4a,b and 5a,b, for each gene, the dominant 5'UTR isoform under hypoxia (from nanoCAGE; taking the longest if multiple isoforms had equal expression) was used to define mRNA features that may explain translational regulation. These included GC content, length, folding energy (calculated using the Mfold (v.3.6) algorithm[86]) and de novo 5'UTR motifs over-represented in regulated transcripts under hypoxia (identified using runStreme from memes (v.1.0.4) implementing STREME[87]). G-quadruplexes were predicted using pqsfinder[88] (v.1.10.1) with min_score = 47 and TOP motifs were quantified using the TOPscore method[25]. Evidence-supported RBP motifs from the ATtRACT database[89] were also quantified in 5'UTRs. Motifs were selected for inclusion in modelling using the 'featureIntegration' function of postNet (contrastSel = 1; regOnly = TRUE; regulationGen = 'translation' or 'buffering'; allFeat = TRUE; and analysis_type = 'lm') to perform stepwise regression modelling only using RBP motifs as variables, selecting motifs that independently explained translational changes (Supplementary Data 4).

For Fig. 3d–i and Extended Data Figs. 3c,d, 4c–f and 5c,d variables describing how 5'UTR features change due to TSS switching were added to the models, including change in the weighted mean 5'UTR length (as both numeric and categorical values), difference in weighted mean GC content and folding energy, change in TOPscore between hypoxia and normoxia, and TSS-switch scores for each transcript from change-point analysis.

All translatome models also included known signatures of translational regulation by various genes and pathways obtained from published studies, including ISR-sensitive translation[22,51], translation activated upon high eIF4E expression[44], translation suppressed upon KD of eIF4A1 (ref. [48]), U34-modified tRNA-dependent offsetting[49], DHX9-dependent translation[41], SNAT2-dependent translation[47], high p-eIF4E-dependent translation[45] and eIF4E2-dependent translation[50]. Additional signatures of eIF4GI[43] (GSE11011), DAP5 (ref. [42]) (GSE115142) and mTOR-dependent[46] (GSE76766) translation and offsetting were obtained following anota2seq analysis.

## Native ChIP-seq preparation and data analysis

H3K4me3 chromatin immunoprecipitation (ChIP) was performed based on previously described protocols[90,91] with modifications. Cells were washed twice with PBS and pelleted (500 g for 5 min at 4 °C). Pellet volume was assessed before flash freezing. Antibodies were conjugated overnight at 4 °C in ChIP Buffer 1 (25 mM Tris, pH 7.5, 5 mM MgCl$_2$, 100 mM KCl, 10% glycerol, 0.1% NP-40, 200 µM phenylmethyl sulfonyl fluoride (PMSF) and 50 µg ml$^{-1}$ BSA) with a 50:50 mix of Dynabeads Protein A and Protein G (Thermo Scientific, 10001D and 10003D) at a ratio of 6 µg antibody to 25 µl beads per sample. Frozen cells were washed and resuspended in 2× cell pellet volume of Buffer N (15 mM Tris Base, 15 mM NaCl, 60 mM KCl, 8.5% sucrose, 5 mM MgCl$_2$, 1 mM CaCl$_2$, 1 mM DTT, 200 µM PMSF, 1× protease inhibitor and 50 µg ml$^{-1}$ BSA) and lysed for 10 min on ice in an equivalent volume of 2× Lysis Buffer (1× Buffer N and 0.6% NP-40). Nuclei were pelleted (500 g for 5 min at 4 °C) and resuspended in 6× cell pellet volume of Buffer N, then spun through a sucrose cushion (15 mM Tris Base, 15 mM NaCl, 60 mM KCl, 30% sucrose, 5 mM MgCl$_2$, 1 mM CaCl$_2$, 1 mM DTT, 200 µM PMSF, 1× protease inhibitor and 50 µg ml$^{-1}$ BSA; 500 g for 12 min at 4 °C). Nuclei pellets were resuspended in 2× pellet volume of Buffer N and 2 µl was diluted with 18 µl 2 M NaCl in triplicate, vortexed and sonicated (30× cycle 30 s on and 30 s off).

Nuclei suspensions (1 µg µl$^{-1}$) were digested with 46 U (2 µl) MNase (Worthington Biochemical Corporation, LS004797) on a ThermoMixer (Eppendorf, 900 rpm at 37 °C for 12 min) and stopped with 1/10 volume of MNase Stop Buffer (100 mM EDTA and 10 mM EGTA) on ice. Then, 5 M NaCl was slowly added to a concentration of 0.6 M with gentle mixing and insoluble nuclei were pelleted (18,000 g for 1 min at 4 °C). Soluble chromatin supernatant was added to 0.07 g of hydroxyapatite (HAP) resin (Bio-Rad 157-0021) rehydrated in HAP Buffer 1 (5 mM NaPO$_4$, pH 7.2, 600 mM NaCl, 1 mM EDTA and 200 µM PMSF) and rotated for 10 min at 4 °C. The HAP mixture was run through an Ultrafree-MC Centrifugal Filter (Millipore, UFC30HV25; 600 g for 30 s at 4 °C) before the column was washed 4× with each of 200 µl HAP Buffer 1, 200 µl HAP Buffer 2 (5 mM NaPO$_4$, pH 7.2, 100 mM NaCl, 1 mM EDTA and 200 µM PMSF) and eluted 3× with 100 µl of HAP Elution Buffer (500 mM NaPO$_4$ pH 7.2, 100 mM NaCl, 1 mM EDTA and 200 µM PMSF). Fragment sizes were confirmed by agarose gel electrophoresis and the concentration was adjusted to 20 ng µl$^{-1}$ using ChIP Buffer 1 before immunoprecipitation (IP).

After reserving 15 µl as an input control, chromatin was added to beads conjugated with H3K4me3 antibody (Epigentek, A4033), rotated for 10 min at 4 °C, washed twice with ChIP Buffer 2 (25 mM Tris, pH 7.5, 5 mM MgCl$_2$, 300 mM KCl, 10% glycerol, 0.1% NP-40, 200 µM PMSF and 50 µg ml$^{-1}$ BSA) and once with each ChIP Buffer 3 (10 mM Tris, pH 7.5, 250 mM LiCl, 1 mM EDTA, 0.5% sodium deoxycholate, 0.5% NP-40, 200 µM PMSF and 50 µg ml$^{-1}$ BSA) and TE buffer. DNA was eluted in 50 µl ChIP Elution Buffer (50 mM Tris, pH 7.5, 1 mM EDTA and 1% SDS) on a ThermoMixer (600 rpm for 10 min at 55 °C). Input controls were adjusted to 50 µl with ChIP Elution Buffer and 2 µl 5 M NaCl, 1 µl 500 mM EDTA and 1 µl 20 mg ml$^{-1}$ Proteinase K were added to all samples and incubated for 2 h at 55 °C. DNA was purified using the MinElute PCR Purification kit (QIAGEN, 28004).

Shotgun library preparation and sequencing was performed at the Genome Quebec Innovation Center or the Institute for Research in Immunology and Cancer (IRIC; Montreal) on an Illumina NovaSeq6000. Samples were sequenced to a depth of ~50–80 million reads (H3K4me3) and ~75–250 million reads (inputs) per sample.

Fastq files were processed using the nf-core/chipseq pipeline (v.2.0.0; https://github.com/nf-core/chipseq)[92]. Reads were aligned to GRCh38/hg38 (RefSeq, release 109) in paired-end mode with parameters –narrow_peak –macs_fdr 0.15 –min_reps_consensus (n replicates – 1). All other settings were default. BAM files were filtered to remove duplicates, multi-mapping, unpaired and ENCODE blacklist[93] reads.

To evaluate H3K4me3 shifts relative to TSSs of protein-coding genes, filtered aligned reads were extracted from BAM files and re-aligned to a custom index of genomic regions corresponding to RefSeq transcripts extended 2 kb upstream and 10 kb downstream of the annotated TSS using Bowtie[75] (v.1.2.2; settings: -v 2 -X 1,000 -a). Genome coverage of H3K4me3 was computed with BEDtools[94] genomeCoverageBed (v.2.29.1; settings: -dz -pc -scale). The cumulative sum of H3K4me3 coverage was calculated for each genomic region corresponding to transcripts, and regions were trimmed where 95% of the total signal was reached due to variable H3K4me3 extension around TSSs. To reduce file sizes and computation time, the position

(relative to TSS) and magnitude of H3K4me3 coverage was summarized at 0.1% increments of cumulative signal (1,000 data points per region). Regions with a maximum scaled coverage below 0.8, or background signal (from input controls) above 0.6 were excluded.

Directional shifts in the position of H3K4me3 around TSSs between hypoxia and normoxia were assessed using two-sided Wilcoxon rank-sum tests, comparing distributions of positions of cumulative H3K4me3 signal. Directionality was determined by the differences between positions of distributions at the 25th, 50th and 75th quantiles. Non-directional changes were identified using two-sided Kolmogorov–Smirnov tests. $P$ values were adjusted for multiple testing using the Benjamini–Hochberg method, with a significance threshold of FDR < 0.01. Given the reproducibility between replicates (Extended Data Fig. 6a,b), the mean of the distributions across replicates was used in statistical tests. Only transcript IDs also detected by nanoCAGE were analysed.

### ATAC-seq preparation and data analysis

ATAC-seq libraries were prepared using a modified Omni-ATAC-seq protocol[95]. In brief, 100,000 cells were pelleted (500$g$ for 5 min at 4 °C) and lysed on ice for 3 min in ATAC-Resuspension Buffer (10 mM Tris-HCl, pH 7.4, 10 mM NaCl and 3 mM MgCl$_2$) containing 0.1% Tween-20, 0.01% digitonin (Thermo Scientific, BN2006) and 0.1% IGEPAL CA-630 (Sigma, I8896). Nuclei were washed with ATAC-Resuspension Buffer containing 0.1% Tween-20 and pelleted (500$g$ for 10 min at 4 °C), then incubated with transposition mix containing Tagment DNA TDE1 Enzyme (Tn5) and Buffer (Illumina 20034197), 0.1% Tween-20 and 0.01% digitonin (1,000 rpm, ThermoMixer; 30 min at 37 °C). Fragments were purified (MinElute PCR Purification kit; QIAGEN, 28004), amplified (NEBNext High-Fidelity 2× PCR Master Mix; NEB, M0541) and indexed (Nextera XT Index Kit; Illumina, FC-131-1002). Final libraries were purified and size-selected with AMPure XP beads (Beckman Coulter, A63881) and the quality was assessed by Bioanalyzer (Agilent, 5067-4626). Samples were sequenced on an Illumina NovaSeq6000 with a 100-bp paired-end read setup at the Genome Quebec Innovation Center (Montreal).

Fastq files were processed using the nf-core/atacseq pipeline (v.2.1.2; https://github.com/nf-core/atacseq)[92]. Reads were aligned to GRCh38/hg38 (RefSeq, release 109) in paired-end mode with default parameters. Duplicates, multi-mapping, unpaired and ENCODE blacklist[93] reads were filtered. BAM files from independent replicates were merged, and consensus peak regions were extended ±1,000 bp using the BEDtools[94] slop function (v.2.29.1). Nucleosome position and occupancy were estimated using the NucleoATAC[52] run function (v.0.3.4) with default settings, and visualized using Integrative Genomics Viewer (IGV; v.2.8.3)[96]. For genes undergoing TSS switching, changes were assessed using DeepTools[97] computeMatrix (v.3.4.3) with –referencePoint 'TSS', anchoring around the dominant normoxic TSS (from nanoCAGE). Nucleosome occupancy and position around TSSs were visualized and significant differences detected using the dsCompareCurves function from deepStats (v.0.4; https://github.com/gtrichard/deepStats)[98].

### Standard RNA-seq preparation and data preprocessing

RNA-seq libraries for C48 (10 μM for 24 h) and DMSO-treated T47D cells were prepared from total RNA by the IRIC Genomics Platform (Montreal), and sequenced on the Illumina Nextseq500 with a 75-base single-end setup. Data preprocessing was performed as described above for Smart-seq2 libraries, accounting for the single-end read setup.

### GPF-DIA proteomics

Protein precipitation and proteolysis for C48 or DMSO-treated T47D cells was carried out as previously described[99]. Cells were lysed in 4 M urea, 2% SDS (w/v), 1% NP-40 (w/v) and 50 mM ammonium bicarbonate using tip-probe sonication (20 × 1 s). Protein concentrations were adjusted to 1 μg μl$^{-1}$ before reduction with 5 mM TEAB and alkylation with 30 mM iodoacetamide for 30 min at room temperature in the dark.

Protein lysates (20 μg) were precipitated at 50% ethanol on 100 μg magnetic beads and washed with 80% ethanol according to SP3 protocols. Proteins were digested with LysC (1:100 dilution; 2 h) followed by Trypsin (1:100 dilution; overnight). Residual peptides were collected with water followed by 25% acetonitrile. Peptides were dried by vacuum centrifugation and resuspended in 0.1% formic acid.

Ultraperformance liquid chromatography with tandem mass spectrometry (UPLC–MS/MS) was performed using the gas phase fractionation data-independent acquisition (GPF-DIA) method[100]. Lysate digests (1 μg) were sampled using a Neo Vanquish in trap-and-elute mode for UPLC. In brief, peptide trapping and washing were performed using a PepMap Neo Trap Cartridge (Thermo Scientific, 174500) before reverse-phase peptide separation (25 cm Aurora TS column; IonOpticks AUR3-25075C18-TS) at 50 °C and 1,800–2,000 V spray voltage for 90 min. Mobile A consisted of 0.1% formic acid in H$_2$0 and Mobile B consisted of 0.1% formic acid and 80% acetonitrile in H$_2$0. Peptides were loaded on the separation column at 5% Mobile B for 5 min and increased to 10% over 5 min, 35% over 50 min and 85% over 5 min, before holding at 95% for 2 min. For spectral library generation, 1 μg lysate digest was serially injected to produce 100 m/z fractions across 300–1,000 m/z using a staggered window scheme of 4-m/z wide windows producing 2-m/z bins after demultiplexing. Raw files were converted to mzML using ProteoWizard (v.3.0.22137) with PeakPicking = 1; Demultiplex = 10ppm; and ZeroSamples = −1. Library and sample mzML files were searched together using DIA-NN (v.1.8.1) to generate a spectral library by allowing two missed cleavages and one variable modification of oxidation. See Supplementary Table 3 for Eclipse data-acquisition settings.

### Analysis of GPF-DIA proteomics and RNA-seq

Paired proteomics and RNA-seq datasets were independently normalized (median and trimmed mean of M values, respectively) and filtered to remove missing or zero values. Changes in protein and mRNA after C48 treatment were analysed using anota2seq[36]. Batch effects were accounted for by including sample replicates as covariate in the model. The threshold settings used were minSlopeTranslation = −1; maxSlopeTranslation = 2; minSlopeBuffering = -2; maxSlopeBuffering = 1; and deltaPT = deltaP = deltaTP = deltaT = 0.1, with a significance threshold maxRvmPAdj = 0.15 (corresponding to FDR < 0.15). Genes were classified into two modes of regulation (changes in protein levels that are congruent with changes in mRNA abundance, and changes in protein levels independent of mRNA levels) using the 'anota2seqRegModes' function.

### Cell proliferation

T47D cells were plated 24 h before treatment. At 0 h, the medium was changed to fresh RPMI with 10% FBS and 25 μM OICR-9429 or an equivalent volume of DMSO. At 48 h, cells received fresh RPMI with 10% FBS and 25 μM OICR-9429 or DMSO and were placed in normoxia or hypoxia (0.5% O$_2$) for an additional 48 h before cell counting.

### RNA ligase-mediated rapid amplification of cDNA ends

RNA ligase-mediated rapid amplification of cDNA ends (RLM-RACE) was performed using the FirstChoice RLM-RACE kit (Invitrogen, AM1700). In brief, 10 μg RNA was treated with calf intestinal phosphatase, extracted with acidic phenol–chloroform and treated with tobacco acid pyrophosphatase. A 5′RACE RNA adaptor was ligated to 5′-phosphate-containing molecules and 2 μl ligated RNA was reverse transcribed (SuperScript IV VILO Master Mix; Invitrogen, 11766050). Nested PCR was performed using 1 μl cDNA with 5′RACE adaptor-specific forward primers (5′RACE_outer or 5′RACE_forw_inner), PDK1-specific reverse primers (PDK1_rev_outer, PDK1_rev_inner) and the AmpliTaq Gold 360 Master Mix (Applied Biosystems, 4398881). Two rounds of PCR were performed: first 95 °C for 10 min, 32 cycles of 95 °C for 30 s, 59 °C for 30 s, 72 °C for 1 min, followed by 72 °C for 7 min and 4 °C hold; second 95 °C for 10 min, 35 cycles of 95 °C for 30 s, 58 °C

for 30 s, 72 °C for 1 min, followed by 72 °C for 7 min and 4 °C hold. PCR products were resolved by agarose gel electrophoresis, excised and gel-extracted (GE Healthcare 28903470) and inserted into the vector pCR4-TOPO using the TOPO TA Cloning kit (Invitrogen, 450030) and One Shot MAX Efficiency DH5α-T1R Competent Cells (Invitrogen, 12297016). Plasmids were purified (Plasmid Mini kit; QIAGEN, 12125) and 5′ends were determined by Sanger sequencing.

### RNA extraction and RT–qPCR

RNA was extracted using 1 ml of TRIzol or TRIzol LS reagent (Invitrogen, 15596018 and 10296010). Reverse transcription was performed on 500 ng RNA using the SuperScript IV VILO Master Mix with ezDNase (Invitrogen, 11766050) at 55 °C. cDNA was diluted fivefold with $H_2O$ and 1 μl was used per qPCR reaction. TaqMan primer/probes for *LOX* (Hs00942480_m1), *PDK1* (Hs01561850_m1), enhanced green fluorescent protein (*eGFP*) (Mr04329676_mr) and *RPLP0* transcripts were used with TaqMan Fast Advanced Master Mix (Thermo Scientific, 4444558).

Isoform-specific *PDK1* 5′UTR qPCRs were performed using SYBR Green PCR Master Mix (Applied Biosystems 4364346) with primer sets amplifying long (>78 nt, PDK1_93/80nt), mid (>45 nt, PDK1_57/51nt) and all (PDK1_ex2) *PDK1* 5′UTR isoforms. Relative quantification used a standard curve with either a 1:4 serial dilution of cDNA from a hypoxia-treated sample or a 1:5 dilution series of a linearized plasmid containing the 134-nt 5′UTR and *PDK1* ORF (pGL4.13_134nt-PDK1, described below) starting with 1 pg μl$^{-1}$. To determine the relative contribution of the three main *PDK1* 5′UTR isoform clusters (>78 nt, 78–45 nt and <45 nt) to total transcript levels (Fig. 7e,f), qPCRs were performed and the relative amount of mid-length or short cluster was calculated: mid-length = qPCR PDK1_57/51nt − qPCR PDK1_93/80nt and short = qPCR PDK1_ex2 − qPCR PDK1_57/51nt. The relative contribution to total transcript level was calculated as the ratio of the relative amount of each cluster versus the relative amount of total transcript. All qPCR reactions were performed in triplicate on a QuantStudio 5 or Viia 7 cycler (Thermo Scientific) and analysed using the QuantStudio Design and Analysis software (v.1.5.2) or QuantStudio Real-time PCR software (v.1.6.1).

### Plasmids

The 5′UTRs of human *ACTB* (NM_001101.5), mouse *ATF4* (NM_009716.3) and 134- and 57-nt *PDK1* 5′UTR mRNA isoforms were synthesized as gBlocks (IDT) containing a T3 promoter and flanking *Nde*I and *Mlu*I restriction sites. gBlocks were subcloned into the bicistronic luciferase reporter pKS-FF-HCV-Ren[101] using standard restriction digest/T4 DNA ligation. The 36-nt *PDK1* 5′UTR was synthesized as an Ultramer DNA oligonucleotides and inserted into the same vector using the NEBuilder HiFi DNA Assembly Cloning kit (NEB, E5520S).

To construct the pGL4.13_134nt-PDK1 plasmid, the 134-nt *PDK1* 5′UTR was PCR-amplified (primer pair PDK1-UTRstd_forw and PDK1_UTRstd_rev) from pKS-134nt-PDK1-FF-HCV-Ren (described above). The *PDK1* ORF was amplified from pDONR223-PDK1 (Addgene, #23804) (primer pair PDK1_ORFstd_forw and PDK1_ORFstd_rev). A pGL4.13 plasmid (Promega, E6681), modified by introduction of a *BsiW*I restriction site downstream of the SV40 promoter, was digested with *BsiW*I (Pfl23II) and *Xba*I (Thermo Scientific, FD0854 and FD0684) to excise the 5′UTR and luciferase ORF, and PCR fragments containing the *PDK1* 5′UTR and ORF were inserted (NEBuilder HiFi DNA Assembly Cloning kit; NEB E5520S).

To generate the CRISPR-resistant pGL4.13mod-134nt-PDK1-PAM, two PAM motifs within the *PDK1* ORF were modified by site-directed mutagenesis (Platinum SuperFi PCR Master Mix; Thermo Scientific 12358010) with overlapping mutation-containing PCR primers (PDK1_PAMmut_forw and PDK1_PAMmut_rev). The lentiviral vector pUltra (Addgene, #24129) was used to create 5′UTR isoform *PDK1* rescue plasmids, and was modified by cloning a second *eGFP* ORF downstream of the P2A sequence to create pUltra_2xEGFP (primer pair EGFP_XbaI_forw and EGFP_XbaI_rev). The Ubc promoter and first *eGFP* ORF were

replaced by a fragment amplified from pGL4.13mod-134nt-PDK1-PAM containing SV40 promoter, *PDK1* 5′UTR and ORF, and flanking *Pac*I and *BsrG*I restriction sites (primer pair SV40-PDK1-pUltra_F and SV40-PDK1-pUltra_R), creating pUltra-SV40-134nt-PDK1-PAM. Plasmids containing 57 and 36-nt PDK1 5′UTRs were generated by PCR amplification of the corresponding regions (5′UTR, PDK1 ORF, P2A and eGFP) flanked by *BsiW*I and *EcoR*I sites from pUltra-SV40-134nt-PDK1-PAM and subcloning into digested pUltra-SV40-134nt-PDK1-PAM (primer pairs BsiWI_PDK1-57 or BsiWI_PDK1-36nt and pUltra_P2A-EcoRI_rev).

All cloning and propagation with pUltra backbone was performed in Stable Competent *Escherichia coli* (NEB, C3040). One Shot MAX Efficiency DH5α-T1R Competent Cells (Invitrogen 12297016) were used for all other plasmids. All plasmids were confirmed by Sanger sequencing.

### In vitro transcription

pKS-FF-HCV-Ren[101] plasmids containing *PDK1* 5′UTR mRNA isoforms were linearized with *BamH*I (Thermo Scientific, FD0055) and purified by phenol/chloroform extraction. In vitro transcription and capping followed a modified protocol from Steinberger et al.[101]. In brief, a 3-μg linearized plasmid was transcribed with 10 μl T3 RNA polymerase, 1× RNA Polymerase Buffer, 1 mM CTP, 1 mM ATP, 1 mM UTP, 0.2 mM GTP, 1 mM 3′-*O*-Me-$^{m7}$GpppG (anti-reverse cap analogue (ARCA)) and 100 U RNase Inhibitor (NEB) in a volume of 100 μl for 3 h at 37 °C. After transcription, 2.5 μl DNase I (5 U, NEB, M0303) was added for 30 min at 37 °C. Samples were purified using the MEGAclear kit (Invitrogen, AM1908).

### In cell translation experiments

The 293T cells ($2 × 10^4$ for hypoxia/normoxia; $1.6 × 10^5$ for thapsigargin/INK128) were seeded in 24-well plates. After 24 h, the medium was refreshed and cells were incubated under normoxia or hypoxia (0.5% $O_2$) for 24 h. Cells were washed with serum-free medium and transfected with 150 ng in vitro transcribed RNA and 1 μl DMRIE-C reagent (Invitrogen, 10459-014) in 200 μl Opti-MEM. After 1 h incubation in normoxia or hypoxia, 200 μl DMEM 10% FBS (for hypoxia or normoxia) or medium containing either 400 nM thapsigargin, 50 nM INK128 (Cell Signalling, 12758S and 30695) or DMSO was added. After an additional 7 h incubation, cells were washed with PBS and lysed using 100 μl Passive Lysis Buffer (Promega). Luciferase activity was measured from 20 μl lysate using the Dual-Luciferase Reporter Assay System (Promega, E1910) on a GloMax Navigator Luminometer (Promega, GM2010). Transfections and measurements were performed in duplicate.

### PDK1 CRISPR knockout and 5′UTR mRNA isoform rescue

The T47D-KO cell lines overexpressing the 134-, 57- and 36-nt *PDK1* 5′UTR mRNA isoforms were generated using lentiviral transduction of pUltra-SV40-nt-PDK1-PAM constructs described above, encoding *PDK1* 5′UTRs, *PDK1* ORF, P2A and *eGFP*. A negative control line used the empty pUltra expressing eGFP alone.

Lentivirus was produced by co-transfecting HEK293T cells ($2.5 × 10^5$ per 6-cm dish; 40–50% confluency) with 4 μg pUltra (±PDK1 inserts), 2.66 μg psPAX2 (Addgene, #35002) and 1.66 μg pMD2.G (Addgene, #12259) using jetPRIME (Polyplus, 101000046). The medium was replaced after 24 h and viral supernatants were collected at 48, 72 and 96 h, filtered (0.45 μm), diluted 1:1 with fresh medium with 8 μg ml$^{-1}$ Polybrene and added to T47D cells ($1.0 × 10^4$ per well, six-well plate). Cells were transduced on three consecutive days, followed by a 48-h recovery. Approximately $2.0 × 10^4$ eGFP-positive cells were sorted by FACS (BD FACSAria Fusion Flow Cytometer; BD Biosciences; Supplementary Fig. 1) into separate six-well dishes as a population and expanded. Genomic integration was confirmed by PCR (AmpliTaq Gold 360 Master Mix; Applied Biosystems 4398881) and primers spanning the SV40 promoter and start of the *PDK1* ORF following extraction of genomic DNA (PureLink genomic DNA mini kit; Invitrogen, K182001). PCR product size was determined by agarose gel electrophoresis.

To deplete endogenous PDK1, two guide RNAs (gRNAs) targeting exon 2 of *hPDK1* (targeting GCAAGAGTTGCCTGTCAGACTGG and TTGCCGCAGAAACATAAATGAGG) were designed using CHOPCHOP[102] and cloned into the lentiCRISPRv2 plasmid conferring G418 resistance (Addgene, #98292) according to the protocol by Zhang et al.[103,104]. Modifications included using BsmBI-v2 (NEB, R0739) for lentiCRISPRv2 digestion and an overnight ligation reaction at 4 °C. Transformation of ligated plasmid was carried out via 42 °C heat-shock into chemically competent Stbl3 cells (Thermo Scientific, C737303). Bacterial colonies were screened and plasmid DNA was extracted using the QIAprep Spin Miniprep kit (QIAGEN, 27104). Insertion of gRNA was confirmed by Sanger sequencing using the human U6 promoter primer.

Lentivirus was produced from the two gRNA-containing or empty lentiCRISPRv2 plasmids and used to transduce the *PDK1* 5′UTR mRNA isoform-expressing or empty vector T47D cells ($1.0 \times 10^4$ per well, six-well plates) on 3 consecutive days, followed by a 48 h recovery and selection with G418 (500 µg ml$^{-1}$; Bio Basic BS723) for an additional 4 days. Single cells were sorted by FACS into 96-well plates containing conditioned medium (45% filtered cultured medium, 45% fresh medium and 10% FBS). Clones were expanded after approximately 2 weeks to generate clonal cell lines, including 134 nt (clones #4 and #9), 57 nt (clones #4 and #9) and 36 nt (clones #4 and #14).

PDK1 protein depletion was confirmed by immunoblotting. Additionally, genomic DNA was extracted and two rounds of PCR were performed using sets of primers in intronic regions flanking the CRISPR target sites (35 PCR cycles with Set 1 followed by 45 PCR cycles with Set 2). PCR products were separated by agarose gel electrophoresis, and the smaller product for each clone was purified (Zymoclean Gel DNA Recovery kit; Zymo Research, D4001) before Sanger sequencing with Set 2 primers, confirming mutations causing a premature stop codon or frameshift in *PDK1*.

### Stability of transcripts with different *PDK1* 5′UTRs
T47D EV, T47D-PDK1-KO_134nt-PDK1-OE, 57nt-PDK1-OE and 36nt-PDK1-OE cells ($8 \times 10^5$) were plated and the medium was refreshed after 24 h, followed by incubation in normoxia or hypoxia (0.5% $O_2$). At 8 h, cells were either collected for RNA extraction or treated with 5 µg ml$^{-1}$ actinomycin D (Sigma, A9415) and incubated for an additional 16 h before collection. RT–qPCR was performed with TaqMan probes for *eGFP* and *RPLP0* mRNA, and transcript levels were quantified by relative standard curve using a 1:5 dilution series of normoxic, untreated T47D EV cDNA. Transcript stability was calculated as the treated-to-untreated ratio and normalized to *RPLP0* to control for clonal differences. Induction of mRNA decay by actinomycin D was confirmed by qPCR for unstable transcripts *RIPK2* (Hs01572684_m1) and *SOX2* (Hs04234836_s1).

### Stable isotope tracing by GC–MS
T47D cells were cultured under normoxia or hypoxia (0.5% $O_2$) for 24 h, then incubated with medium containing 1 mM unlabelled pyruvate for 2 h, followed by 1 mM labelled ([U-$^{13}$C])-pyruvate (Cambridge Isotope Laboratories) for 15 min. Steady-state and tracing samples were processed in parallel. Cells were washed with cold saline (9 g l$^{-1}$ NaCl) and quenched in 80% methanol. Lysates were sonicated at 4 °C for 10 min and centrifuged (18,440*g* at 4 °C for 10 min). Internal standard (750 ng myristic acid-D27) was added to supernatants and dried overnight in a SpeedVac (Labconco) at 4 °C. Dried pellets were resuspended in methoxyamine hydrochloride (10 mg ml$^{-1}$ in pyridine), sonicated, vortexed (3× for 30 s) and centrifuged (18,440*g* at 4 °C for 10 min). Samples were heated to 70 °C for 30 min. Derivatization with *N*-tert-butyldimethylsilyl-*N*-methyltrifluoroacetamide (MTBSTFA) was performed at 70 °C for 1 h, and 1 µl was injected into an Agilent 5975C GC–MS. Metabolites were analysed with GC–MS MassHunter (v.10.2) (Agilent) and areas were normalized by the internal standard and cell number.

### Statistics and reproducibility
Statistical methods are detailed in the relevant Methods sections and figure captions. No statistical methods were used to pre-determine sample sizes, but sizes are comparable to those previously published and standard in the field. Covariates were controlled for in differential expression analyses by including batch/replicate in linear regression models. One nanoCAGE sample from normoxia T47D cells and one H9 replicate were excluded due to sequencing failure, identified by low read counts and outlier characteristics upon PCA. Reproducibility of sequencing data was assessed by PCA, hierarchical clustering and sample correlation. Reproducible replications for representative micrographs are described in figure legends and Source data. No randomization was applied, and investigators were not blinded to experimental conditions during analysis.

### Reporting summary
Further information on research design is available in the Nature Portfolio Reporting Summary linked to this article.

### Data availability
Raw and processed RNA-seq, ChIP-seq, ATAC-seq and nanoCAGE data are available on the National Center for Biotechnology Information (NCBI) Gene Expression Omnibus under accession number GSE243418. All sequencing datasets were aligned to the NCBI RefSeq[105] GRCh38/hg38 genome assembly, using corresponding RefSeq transcript annotations (release 109, 2020-11-20; https://ftp.ncbi.nlm.nih.gov/refseq/H_sapiens/annotation/annotation_releases/109.20201120/GCF_000001405.39_GRCh38.p13/). Proteomics data are available on the PRIDE database under accession number PXD058655. Metabolite data are available on the MetaboLights repository under accession number MTBLS12086. Other published raw datasets used in this study can be accessed at GSE11011 (eIF4G-dependent translation signature), GSE115142 (DAP5-dependent translation signature) and GSE76766 (mTOR-dependent translation signature). All other experimental data are provided in the associated Supplementary and Source data files. Unique biological materials generated in this study are available from the corresponding authors upon reasonable request. Source data are provided with this paper.

### Code availability
Original software packages used in this study, along with datasets and code to reproduce these analyses, are freely available on a Code Ocean capsule (https://doi.org/10.24433/CO.4525673.v1).

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

## Acknowledgements

We thank W. Hahn, D. Root, S. Igelmann and the late J. Pelletier for generously sharing plasmid constructs used in this work. We also acknowledge technical assistance from S. McLaughlan and C. Young and advice from V. Pelechano. We received support from the National Genomics Infrastructure in Stockholm funded by Science for Life Laboratory, the Knut and Alice Wallenberg Foundation and the Swedish Research Council, and the Swedish National Infrastructure for Computing/Uppsala Multidisciplinary Center for Advanced Computational Science for assistance with massively parallel sequencing and access to the UPPMAX computational infrastructure. This project was supported by the Canadian Institutes of Health Research (CIHR) (PS-159541 and PS-148615 awarded to L.-M.P.). Work carried out in the laboratory of I.T. was funded by CIHR (PJT-451236). I.T. is supported by a Canada Research Chair in Regulation of mRNA Translation and Metabolism. Work carried out in the laboratory of O.L. was supported by grants from the Swedish Research Council (2020-01665), the Swedish Cancer Society (22 2186), the Cancer Research Funds of Radiumhemmet (211222) and the Wallenberg Academy Fellow programme. K.W., D.P. and T.T.C. are supported by Cancer Research Society Next Generation of Scientists Awards. P.J. is supported by a Fonds de recherche du Québec (FRQ-S) Doctoral Training award.

## Author contributions

L.-M.P., O.L. and K.W. conceived of and designed the study. B.D., L.L., P.J., S.C., K.W., J.A.V., T.T.C., D.P., L.M., M.J., K.T., H.P. and R.P. performed experimental work. K.W., B.D., P.J., R.P., D.P., T.T.C., G.A.L., L.-M.P., O.L. and I.T. performed analysis and/or interpreted the data. K.J.S., K.W. and O.L. developed and applied computational methods used for the analysis of sequencing data. K.W., B.D., L.-M.P., O.L. and I.T. wrote the paper, with input from all authors.

## Competing interests

The authors declare no competing interests.

## Additional information

**Extended data** is available for this paper at https://doi.org/10.1038/s41556-025-01786-8.

**Correspondence and requests for materials** should be addressed to Ola Larsson or Lynne-Marie Postovit.

**Reviewer recognition** *Nature Cell Biology* thanks Davide Ruggero and the other, anonymous, reviewer(s) for their contribution to the peer review of this work. Peer reviewer reports are available.

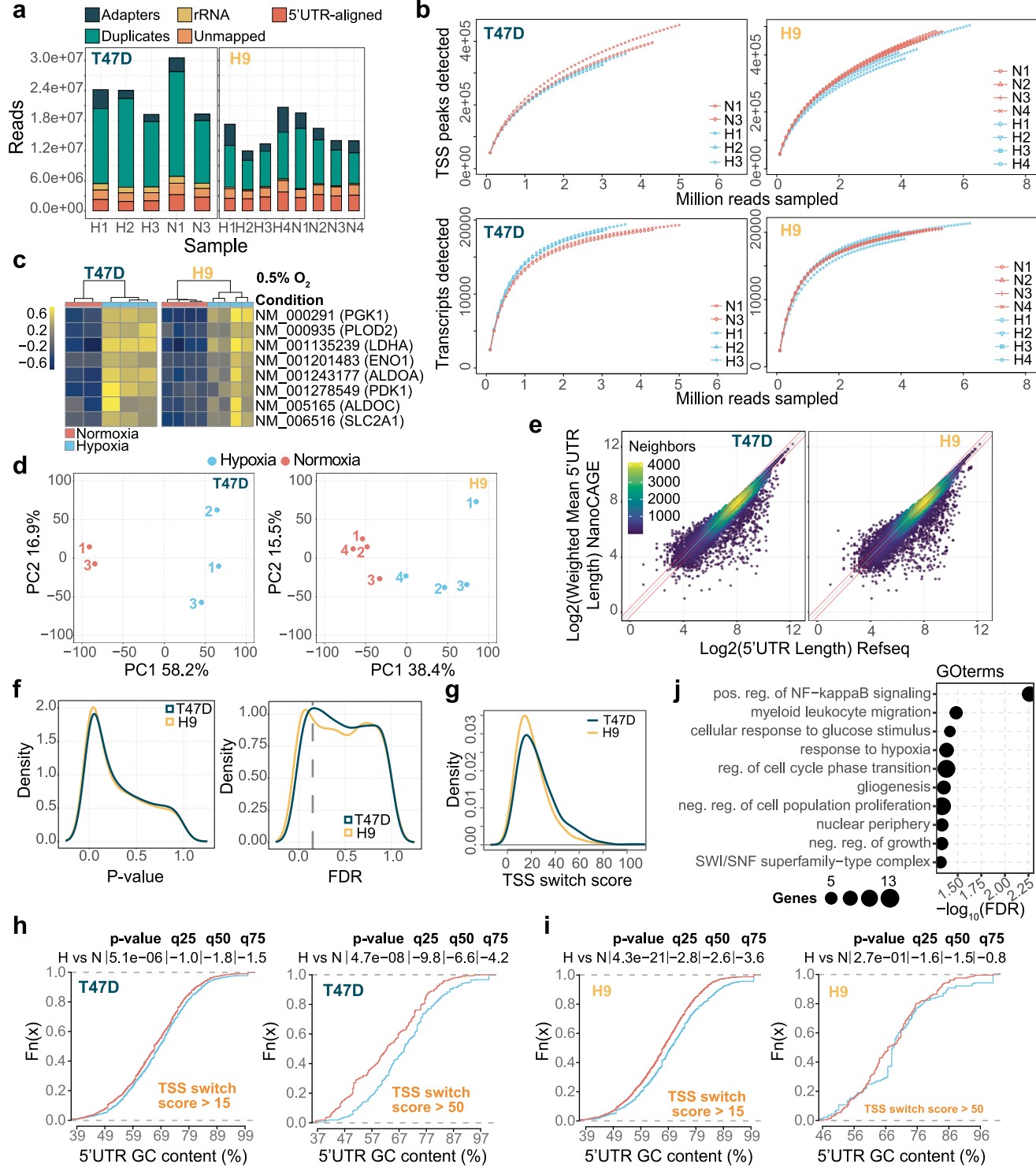

**Extended Data Fig. 1 | See next page for caption.**

**Extended Data Fig. 1 | Hypoxia-induced TSS switching results in extensive remodelling of 5´UTRs. a**, Summary of nanoCAGE preprocessing including filtering of reads for adaptors, duplicate reads, ribosomal RNA, and reads not mapping to 5´UTR regions. One replicate of T47D in normoxia was discarded due to outlier characteristics. **b**, NanoCAGE library complexity for T47D and H9 cells evaluated according to the number of distinct peaks (TSSs) and protein-coding transcripts (RefSeq) detected with increasing numbers of sampled sequencing reads. **c**, Heatmaps showing expression of hypoxia-responsive transcripts from nanoCAGE sequencing of T47D and H9 cells (all isoforms combined). **d**, PCA of 5´UTR isoform expression in T47D and H9 cells grown in hypoxia and normoxia (0.5% and 20% $O_2$; T47D 48 h; H9 24 h) quantified by nanoCAGE. **e**, Weighted mean 5´UTR length measured by nanoCAGE compared to RefSeq annotated 5´UTR length in T47D and H9 cells. Colour gradient indicates point density. **f**, Kernel density estimation *P* value and *FDR* distributions for differential TSS usage between hypoxia and normoxia in T47D and H9 cells. **g**, Kernel density estimation change-point TSS-switch score distributions for transcripts that underwent significant (FDR < 0.15) TSS switching under hypoxia in T47D (n = 2,552) and H9 (n = 3,423) cells. Higher values indicate a greater difference in relative 5´UTR isoform abundance between conditions. **h**, Empirical cumulative distribution function (eCDF) plots comparing %GC content between hypoxia-enriched (blue) and depleted (red) 5´UTR sequence segments identified by change-point analysis in T47D cells. Significant differences in %GC content were assessed using two-sided Wilcoxon rank-sum tests with the magnitude of shifts at each quartile (q25–75) indicated. Left panel: differences for transcripts with TSS switch score >15; *P* = 5.1e-06. Right panel: differences for transcripts with TSS switch score > 50; *P* = 4.7e-08. **i**, As in **h**, for H9 cells. Left panel: differences for transcripts with TSS switch score >15; *P* = 4.3e-21. Right panel: differences for transcripts with TSS switch score > 50; *P* = 2.7e-01. **j**, Gene ontology enrichments for the set of genes corresponding to transcripts that underwent the same patterns of hypoxia-induced TSS switching in both T47D and H9 cells. Source numerical data are available in.

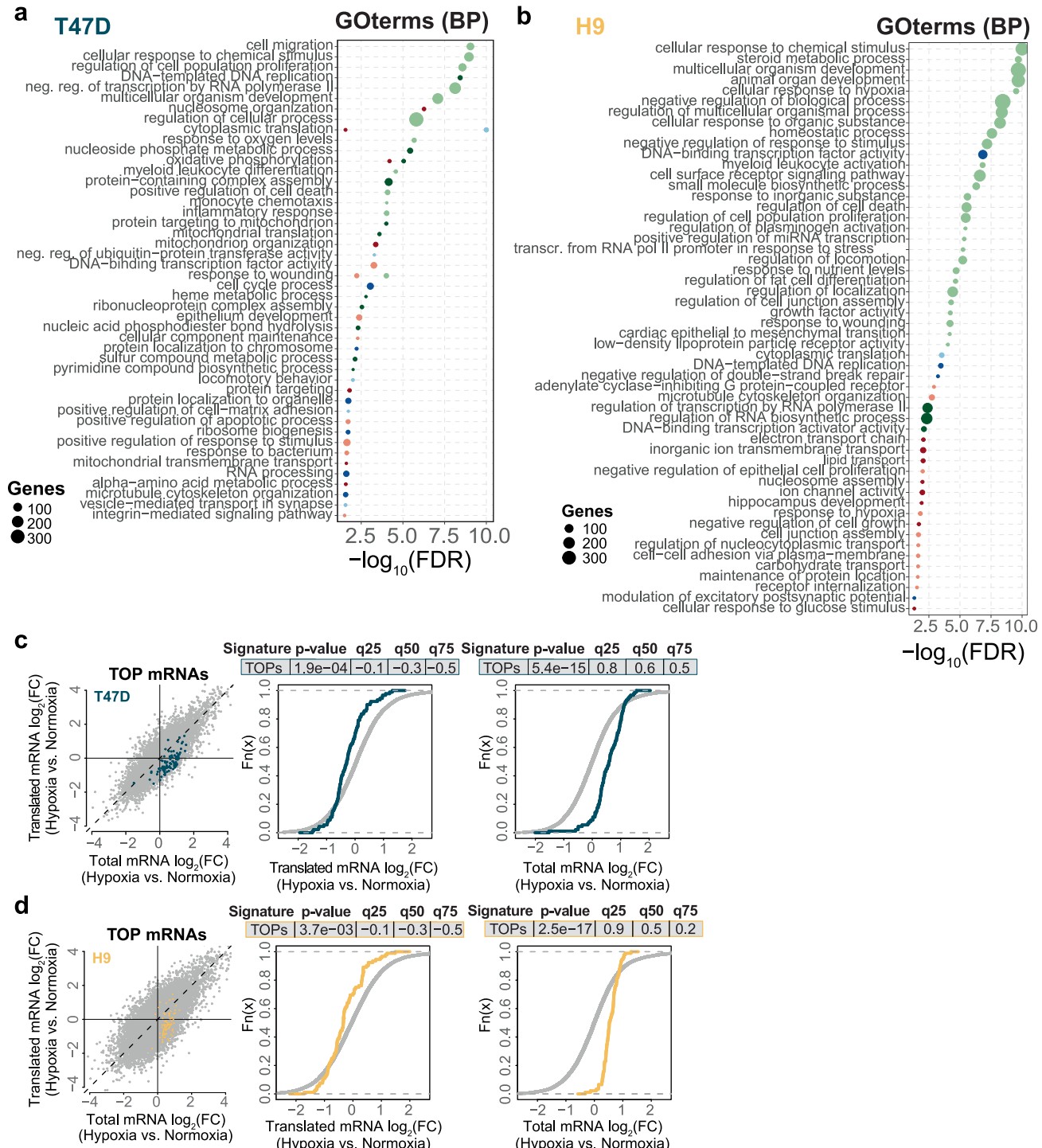

**Extended Data Fig. 2 | Hypoxia-induced translatome remodelling impacts many cellular processes. a,b**, Gene Ontology enrichments for biological processes among genes assigned to each regulatory mode in Fig. 2e for T47D cells (**a**) and Fig. 2f for H9 cells (**b**). **c,d**, Scatterplots from anota2seq translatome analysis of T47D (**c**) and H9 cells (**d**) with the location of canonical TOP mRNAs from Philippe et al.[25] coloured (left). eCDFs showing the log$_2$ fold changes for the

TOP mRNAs in polysome-associated (middle) (T47D $P$ = 1.9e-04; H9 $P$ = 3.7e-03), and total mRNA fractions (right) (T47D $P$ = 5.4e-15; H9 $P$ = 2.5e-17). Grey lines indicate the background (that is non-TOP genes). Significant differences between gene sets assessed using Wilcoxon rank-sum tests. Differences in gene expression at each quartile (q25-75) are indicated. Source numerical data are available in.

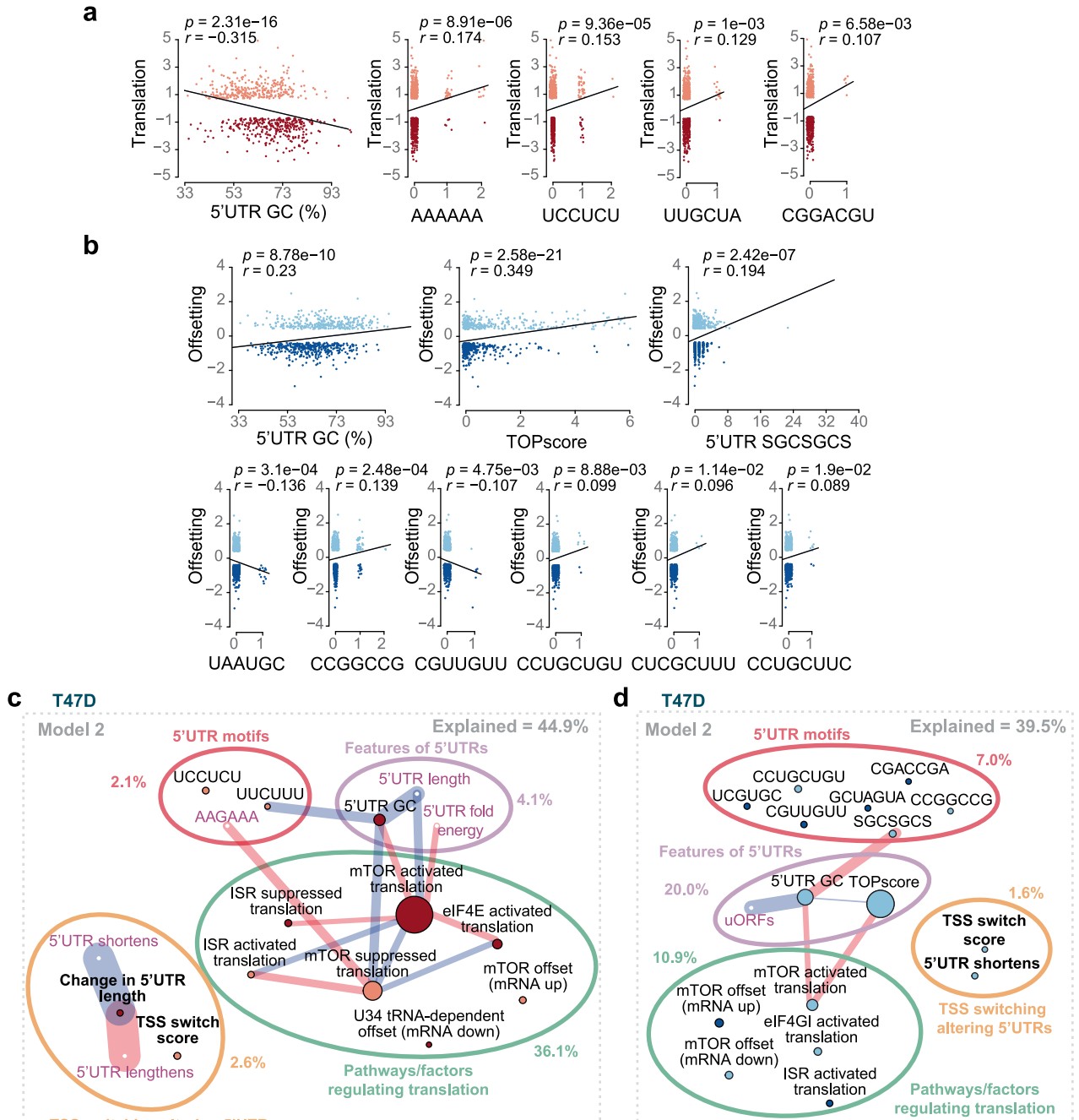

**Extended Data Fig. 3 | TSS switching alters regulatory 5´UTR features and shapes the hypoxia-induced translatome. a**, Scatterplot comparing the composition of each 5´UTR feature identified in Model 1 versus the change in translation efficiency that occurs under hypoxia. Pearson's correlation coefficient and $P$ values from two-sided tests indicated. %GC $r$ = -0.315, $P$ = 2.31e-16; AAAAAA $r$ = 0.174, $P$ = 8.91e-06; UCCUCU $r$ = 0.153, $P$ = 9.36e-05; UUGCUA $r$ = 0.129, $P$ = 1.0e-03; CGGACGU $r$ = 0.107, $P$ = 6.58e-03. **b**, As in **a**, for translational offsetting. %GC $r$ = 0.230, $P$ = 8.78e-10; TOPscore $r$ = 0.349, $P$ = 2.58e-21; SGCSGCS $r$ = 0.194, $P$ = 2.42e-07; UAAUGC $r$ = -0.136, $P$ = 3.1e-04; CCGGCCG $r$ = 0.139,

$P$ = 2.48e-04; CGUUGUU $r$ = -0.107, $P$ = 4.75e-03; CCUGCUGU $r$ = 0.099, $P$ = 8.88e-03; CUCGCUUU $r$ = 0.096, $P$ = 1.14e-02; CCUGCUUC $r$ = 0.089, $P$ = 1.9e-02. **c**, Networks displaying the complete results of Model 2. Percentages of explained and unexplained variance in translation efficiency, and contributions from each input category, are indicated. Connections between features (nodes) indicate substantial correlations. Node colours indicate the mode of translational regulation under hypoxia the feature is associated with. **d**, As in **c**, but modelling changes in translational offsetting induced by hypoxia in T47D cells. Source numerical data are available in.

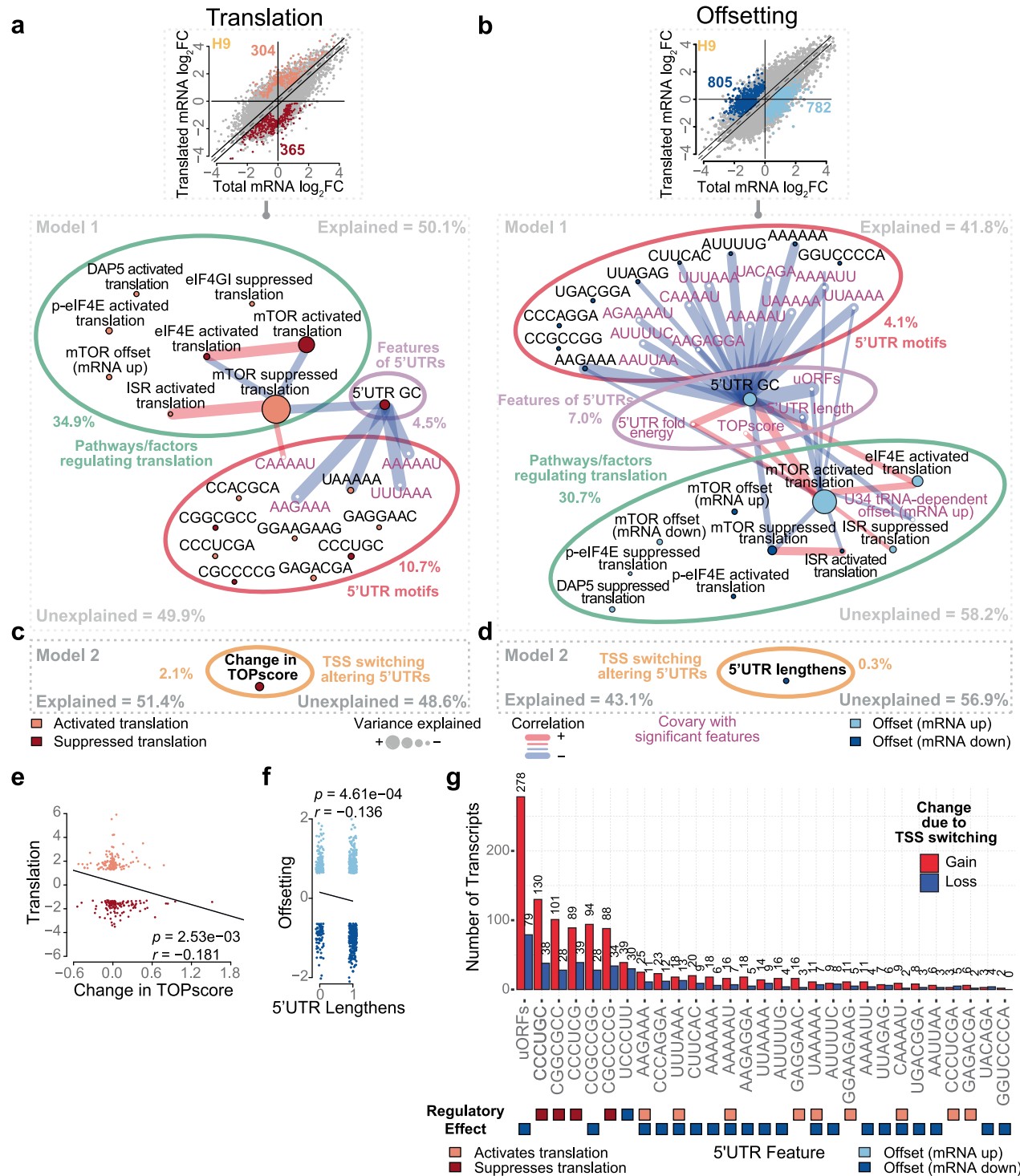

**Extended Data Fig. 4 | TSS switching alters regulatory 5′UTR features and shapes the hypoxia-induced translatome in H9 cells. a**, Network plot displaying the results of Model 1. Percentages of explained and unexplained variance in translation efficiency, and contributions from each input category, are indicated. Connections between features (nodes) indicate substantial correlations. Node colours indicate the mode of translational regulation under hypoxia the feature is associated with. **b**, As in **a**, but modelling translational offsetting induced by hypoxia in H9 cells. **c**, Selection of the full network plot for Model 2, displaying the additional impact of adding TSS-switching signatures to Model 1 in explaining changes in translation efficiency under hypoxia in H9 cells. **d**, As in **c**, but modelling translational offsetting induced by hypoxia in H9 cells.

**e**, Scatterplot comparing the change in TOPscore vs. the change in translation efficiency that occurs under hypoxia. Pearson's correlation coefficient and $P$ values from two-sided tests. $r$ = -0.181, $P$ = 2.53e-03. **f**, Scatterplot comparing 5′UTR lengthening events vs. translational offsetting that occurs under hypoxia. Pearson's correlation coefficient and p values from two-sided tests indicated. $r$ = -0.136, $P$ = 4.61e-04. **g**, Bar plot indicating the number of transcripts that gain or lose 5′UTR elements identified in either Model 1 or 2 in H9 cells. Transcripts were considered if more than 10% of the expressed pool of 5′UTR isoforms gained or lost the 5′UTR element. The translation mode associated with each element is indicated by coloured squares below. Source numerical data are available in.

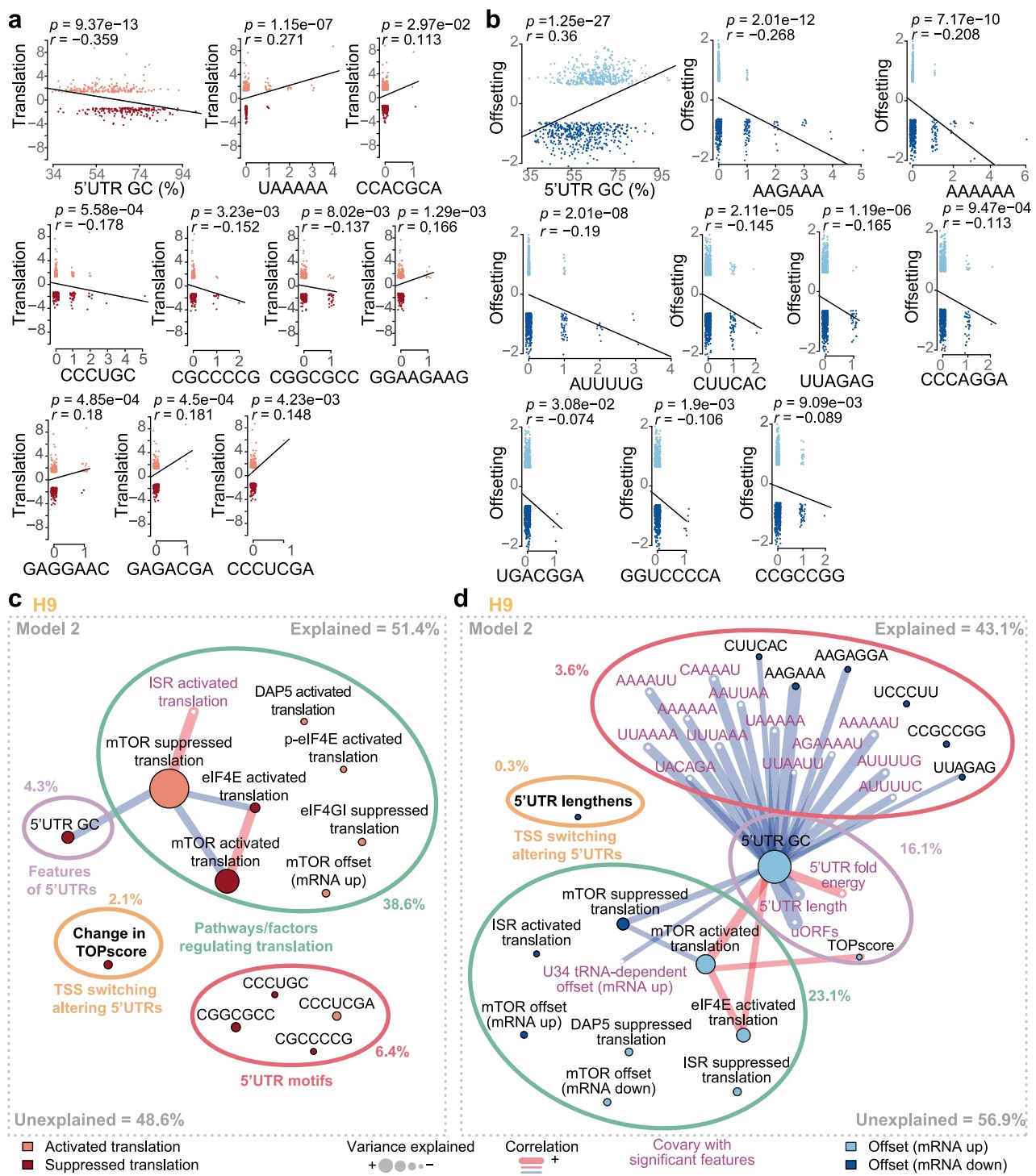

**Extended Data Fig. 5 | TSS switching alters regulatory 5´UTR features and shapes the hypoxia-induced translatome in H9 cells. a**, Scatterplot comparing the composition of each 5´UTR feature identified in Model 1 versus the change in translation efficiency that occurs under hypoxia. Pearson's correlation coefficient and $P$ values from two-sided tests indicated. %GC $r$ = -0.359, $P$ = 9.37e-13; UAAAAA $r$ = 0.271, $P$ = 1.15e-07; CCACGCA $r$ = 0.113, $P$ = 2.97e-02; CCCUGC $r$ = -0.178, $P$ = 5.58e-04; CGCCCCG $r$ = -0.152, $P$ = 3.32e-03; CGGCGCC $r$ = -0.137, $P$ = 8.02e-03; GGAAGAAG $r$ = 0.166, $P$ = 1.29e-03; GAGGAAC $r$ = 0.180, $P$ = 4.85e-04; GAGACGA $r$ = 0.181, $P$ = 4.5e-04; CCCUCGA $r$ = 0.148, $P$ = 4.23e-03. **b**, Same as **a** for translational offsetting. %GC $r$ = 0.36, $P$ = 1.25e-27; AAGAAA $r$ = -0.268, $P$ = 2.01e-12; AAAAAA $r$ = -0.208, $P$ = 7.17e-10; AUUUUG $r$ = -0.19, $P$ = 2.01e-08;

CUUCAC $r$ = -0.145, $P$ = 2.11e-05; UUAGAG $r$ = -0.165, $P$ = 1.19e-06; CCCAGGA $r$ = -0.113, $P$ = 9.47e-04; UGACGGA $r$ = -0.074, $P$ = 3.08e-02; GGUCCCCA $r$ = -0.106, $P$ = 1.9e-03; CCGCCGG $r$ = -0.089, $P$ = 9.09e-03. **c**, Network plot displaying the complete results of Model 2 in H9 cells. Percentages of explained and unexplained variance in translation efficiency, and contributions from each input category, are indicated. Connections between features (nodes) indicate substantial correlations. Node colours indicate the mode of translational regulation under hypoxia the feature is associated with. **d**, As in **c**, but modelling changes in translational offsetting induced by hypoxia in H9 cells. Source numerical data are available in.

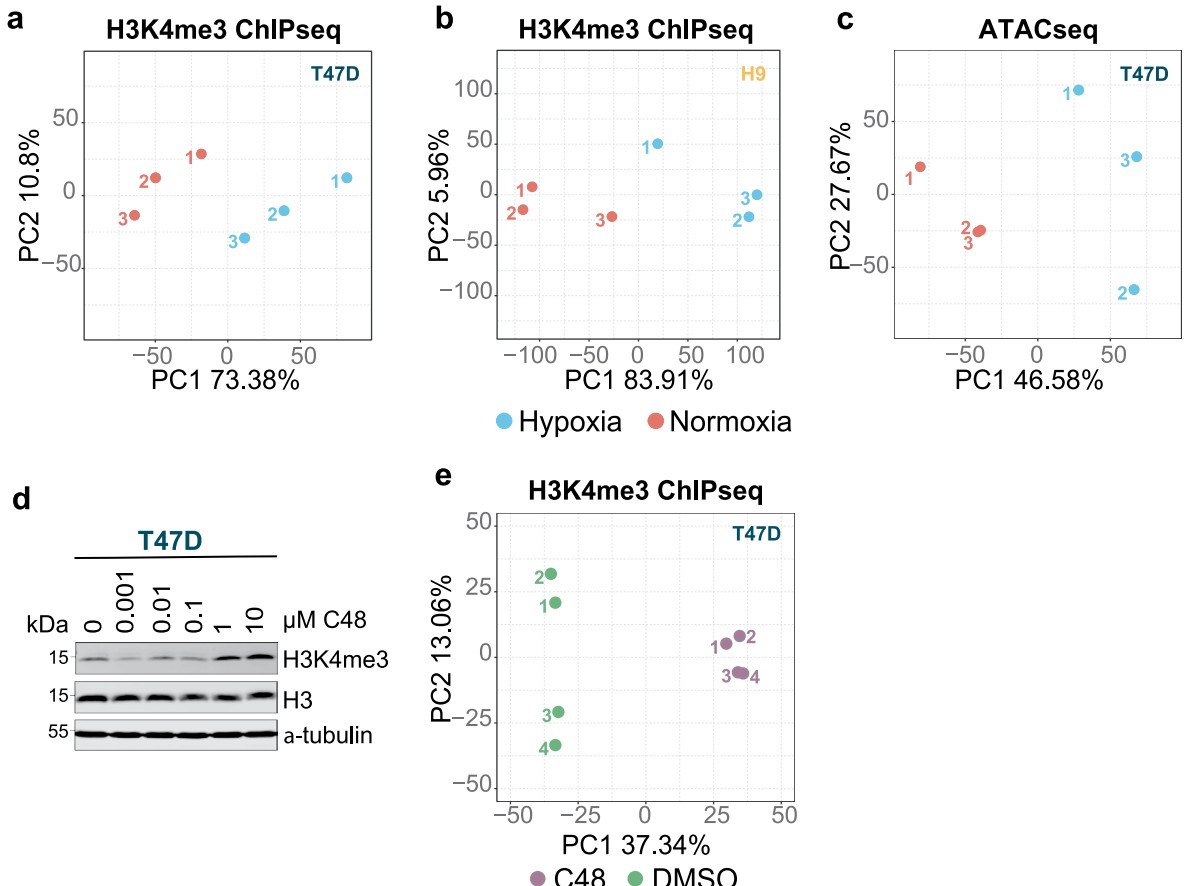

**Extended Data Fig. 6 | Hypoxia-induced TSS switching is associated with altered H3K4me3 and changes in nucleosome context. a,b**, PCA of quantification of H3K4me3 peaks surrounding TSSs from T47D (**a**) and H9 (**b**) cells grown in hypoxia and normoxia and quantified by ChIP-seq ($n = 3$ independent experiments). **c**, PCA of quantification of ATAC-seq peaks at TSSs from T47D cells grown in hypoxia and normoxia ($n = 3$ independent experiments). **d**, Immunoblots of H3K4me3, and H3, with α-tubulin as a loading control. C48 was titrated from 0 to 10 μM in T47D cells. Lysates were harvested after 24 h treatment ($n = 3$ independent experiments). **e**, PCA of quantification of H3K4me3 peaks surrounding TSSs from T47D cells treated with C48 or DMSO for 24 h quantified by ChIP-seq ($n = 4$ independent experiments). Unprocessed blots are available in.

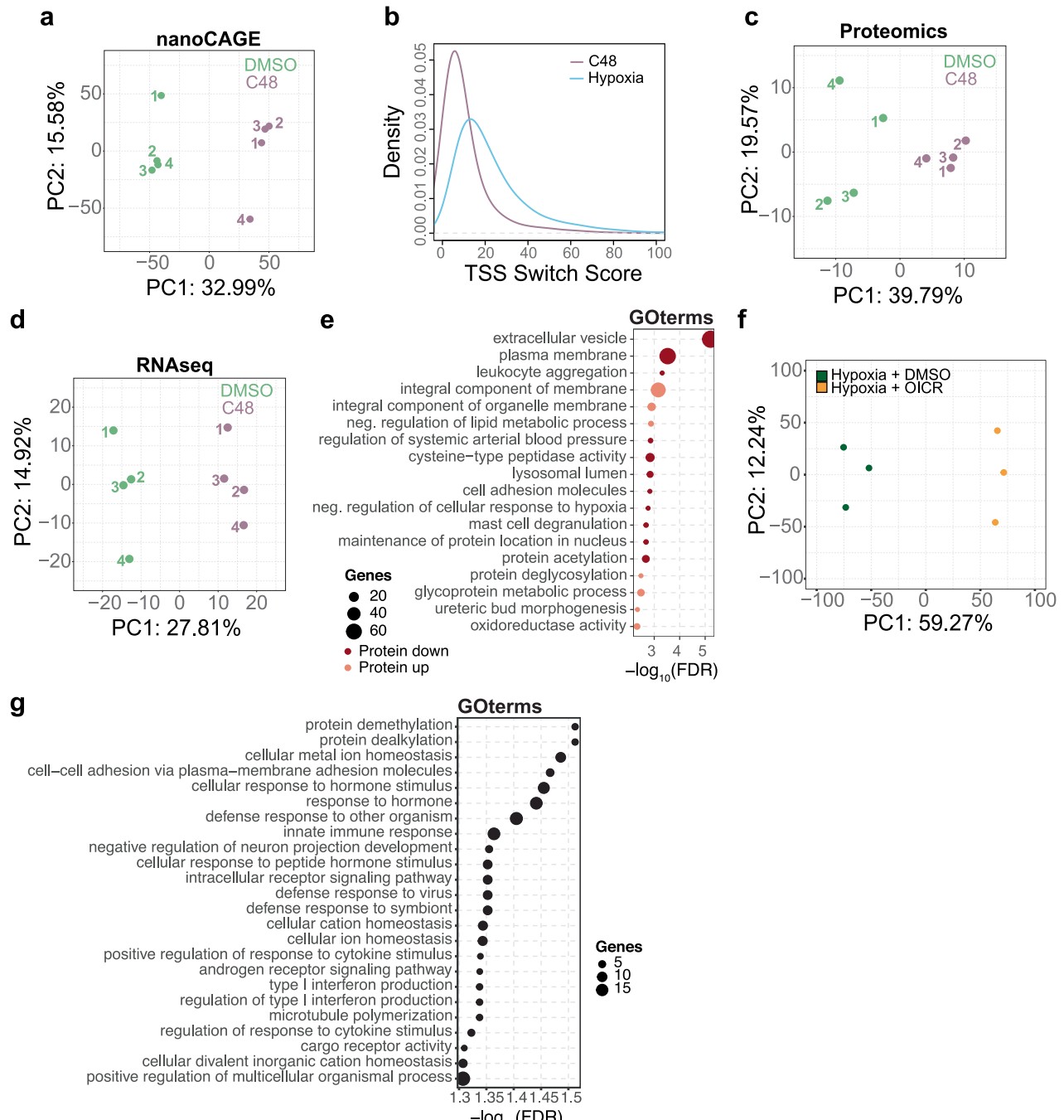

**Extended Data Fig. 7 | Inhibition of KDM5 induces TSS switching that remodels 5′ UTRs and is associated with proteome changes. a**, PCA of 5′ UTR isoform expression measured by nanoCAGE sequencing in T47D cells treated with 10 μM C48 for 24 h, and DMSO-treated controls (*n* = 4 independent experiments). **b**, Kernel density estimation change-point TSS switch score distributions for transcripts with significant (FDR < 0.15) TSS switching after C48 treatment (*n* = 3,287), compared to hypoxia in T47D cells (*n* = 2,552). Higher values indicate greater differences in 5′ UTR isoform expression. **c,d**, PCA of GPF-DIA proteomics data (**c**), and mRNA expression measured by RNA-seq (**d**) in T47D cells treated with 10 μM C48 for 24 h, and DMSO-treated controls (*n* = 4 independent experiments). **e**, Gene ontology enrichments for genes where protein levels were altered independent of changes in mRNA levels in C48-treated T47D cells. **f**, PCA of 5′ UTR isoform expression measured by nanoCAGE sequencing in T47D cells co-treated with hypoxia and either 25 μM OICR-9429 or DMSO vehicle control (*n* = 3 independent experiments). **g**, Gene ontology enrichments for genes where hypoxia-induced TSS switching was reversed by OICR-9429 treatment. Source numerical data are available in.

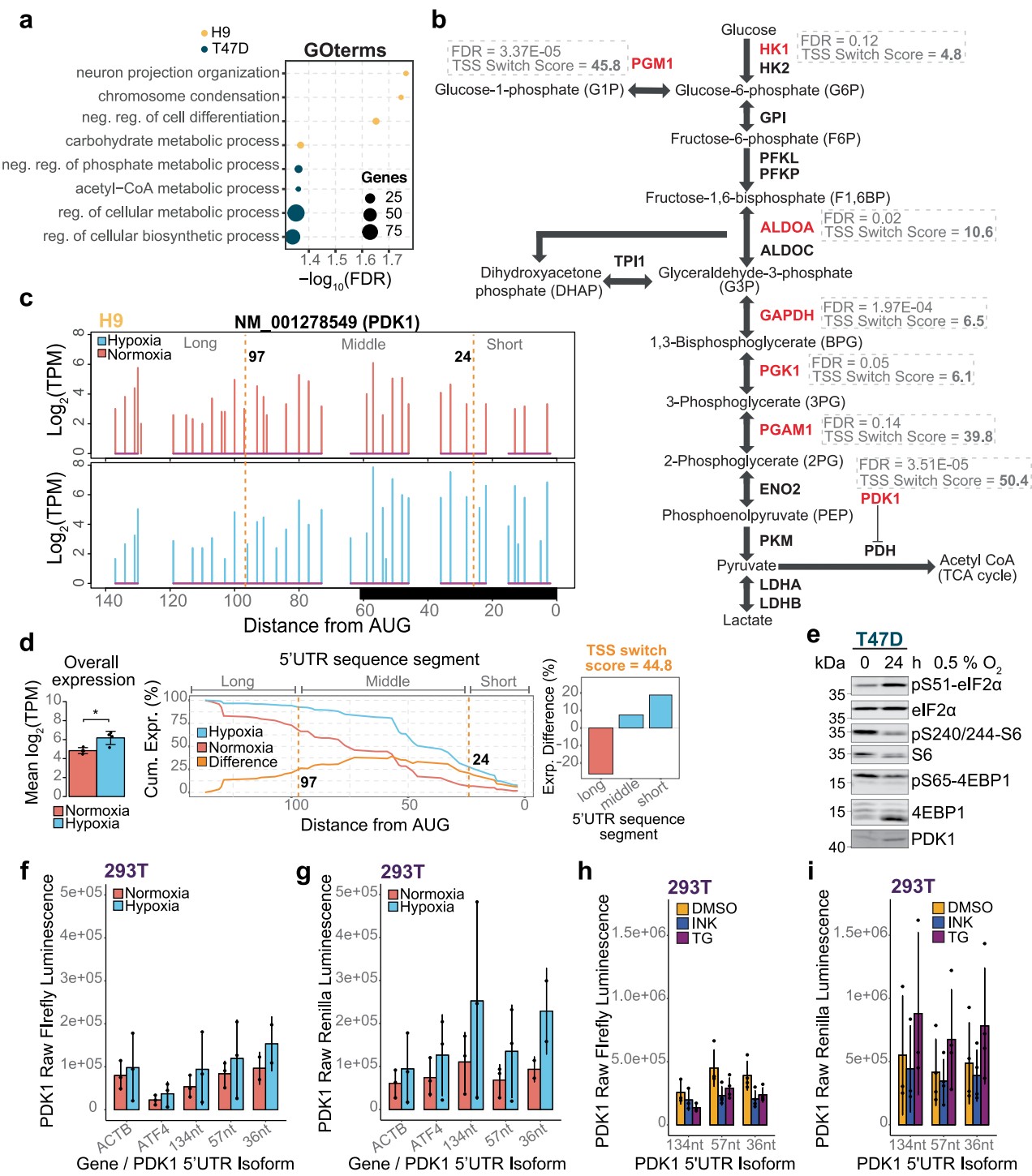

**Extended Data Fig. 8 | TSS switching orchestrates adaptation to hypoxia by regulating availability of differentially translated mRNA isoforms. a**, Gene Ontology enrichments for biological processes among genes that undergo significant TSS switching under hypoxia in T47D and H9 cells. **b**, Enzymes involved in glycolysis undergo TSS switching under hypoxia. Genes with significant TSS switching events in T47D cells are marked in red. *FDR* values, and TSS-switch scores (reflecting the degree of switching) are indicated. **c**, Quantification of 5´UTR isoforms for NM_001278549 (*PDK1* mRNA) in hypoxia and normoxia-treated H9 cells. Same outline as Fig. 1h but also indicating sequence segments identified by change-point analysis (dotted orange lines). **d**, Change-point analysis of NM_001278549 (*PDK1* mRNA) identified change-points at 24 and 97 nt upstream of the start codon (dotted orange lines)

(middle), defining shorter sequence segments with isoforms that are enriched under hypoxia and a longer segment that is depleted (right). Difference in total mRNA expression indicated (left). Mean ± s.d.; *P* = 0.0227; two-sided *t*-test. (*n* = 4 independent experiments). **e**, Representative immunoblots of phosphorylated rpS6, 4E-BP1 and eIF2α, and PDK1 from T47D cells used in polysome profiling in normoxia and hypoxia. 4E-BP1, rpS6, and eIF2α were used as loading controls (*n* = 2 independent experiments). **f**, Raw Firefly luminescence values relating to Fig. 7g. **g**, Raw Renilla luminescence values relating to Fig. 7g. **h**, Raw Firefly luminescence values relating to Fig. 7h. **i**, Raw Renilla luminescence values relating to Fig. 7h. Source numerical data and unprocessed blots are available in.

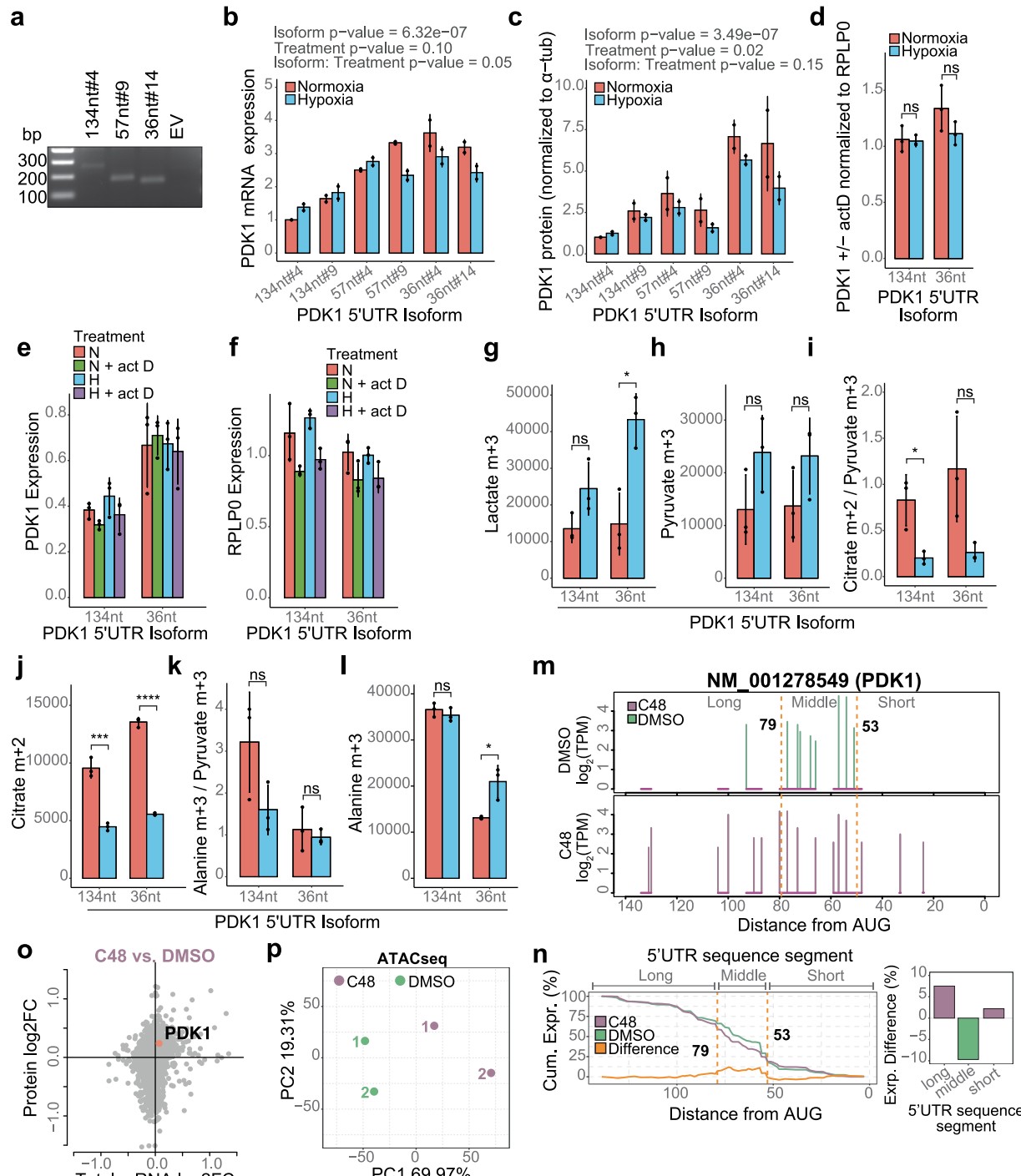

**Extended Data Fig. 9 | See next page for caption.**

**Extended Data Fig. 9 | TSS switching orchestrates adaptation to hypoxia by regulating availability of differentially translated mRNA isoforms.**
**a**, PCR confirmation of lentivirus-mediated integration of *PDK1* 5′ UTR isoforms in the gDNA of T47D-PDK1-KO cells. PCR spans the region between the SV40 promoter and the beginning of the *PDK1* ORF. **b**, Quantification of *PDK1* 5′ UTR isoform expression in hypoxia and normoxia-treated T47D cells using RT–qPCR and normalized to 134nt#4 normoxia. *P* values result from a linear model (ANOVA) with the design Expression - Isoform + Treatment + Replicate + Isoform:Treatment. Residual degrees of freedom = 16; mean ± s.d.; Isoform *P* = 6.32e-07; Treatment *P* = 0.10; Isoform:Treatment *P* = 0.05 (*n* = 2 independent experiments). **c**, Densitometry quantification of PDK1 protein normalized to α-tubulin loading control, measured by immunoblot in hypoxia and normoxia-treated T47D cells. *P* values are provided as in **b**. Residual degrees of freedom = 16; mean ± s.d.; Isoform *P* = 3.49e-07; Treatment *P* = 0.02; Isoform:Treatment *P* = 0.15 (*n* = 2 independent experiments). **d**, Stability of *PDK1* 5′ UTR mRNA isoforms is equal under hypoxia. T47D-PDK1-KO cells stably expressing either 134, or 36 nt 5′ UTR *PDK1* isoforms were maintained in normoxia or hypoxia for 24 h and treated with actinomycin D for 16 h. The ratio of *PDK1* transcripts (± actD) vs. *RPLP0* transcripts (± actD) is displayed. Mean ± s.d.; ns = not significant; *P* = 0.8674 for 134 nt, *P* = 0.183 for 36 nt; two-sided *t*-test (*n* = 3 independent experiments). **e**, Quantification of *PDK1* transcripts (± actD) determined by RT–qPCR (relative standard curve) used to calculate ratios in **d**. Mean ± s.d. **f**, Quantification of *RPLP0* transcripts (± actD) determined by RT–qPCR (relative standard curve) used to calculate ratios in **d**. Mean ± s.d. **g,h**, Labelled metabolite levels of lactate m + 3 (**g**), and pyruvate m + 3 (**h**) from T47D cells expressing different *PDK1* 5′ UTR mRNA isoforms grown in hypoxia or normoxia for 24 h. Metabolites were measured using stable isotope tracing

by GC–MS. Levels are calculated as areas normalized to cell count and internal standard (myristic acid-D27). Mean ± s.d.; ns = not significant; Lactate m + 3: *P* = 0.07952 for 134 nt, *P* = 0.01093 for 36 nt; Pyruvate m + 3: *P* = 0.1241 for 134 nt, *P* = 0.1694 for 36 nt; two-sided *t*-test (*n* = 3 independent experiments). **i**, The ratio of labelled citrate m + 2 / pyruvate m + 3 as in **g**, **h**. Mean ± s.d.; ns = not significant; *P* = 0.0193 for 134 nt, *P* = 0.05344 for 36 nt; two-sided *t*-test (*n* = 3 independent experiments). **j**, Labelled metabolite levels of citrate m + 2, as in **g**, **h**. Mean ± s.d.; *P* = 0.0006 for 134 nt, *P* = 5.14e⁻⁰⁶ for 36 nt; two-sided *t*-test (*n* = 3 independent experiments). **k**, The ratio of labelled alanine m + 3 / pyruvate m + 3, as in **i**. Mean ± s.d.; ns = not significant; *P* = 0.1036 for 134 nt, *P* = 0.5926 for 36 nt; two-sided *t*-test (*n* = 3 independent experiments). **l**, Labelled metabolite levels of alanine m + 3, as in **g**, **h**. Mean ± s.d.; ns = not significant; *P* = 0.3793 for 134 nt, *P* = 0.0190 for 36 nt; two-sided *t*-test (*n* = 3 independent experiments). **m**, Quantification of 5′ UTR isoforms for NM_001278549 (*PDK1* mRNA) in 10 μM C48 and DMSO-treated T47D cells. Change-point-identified sequence segments indicated by dotted orange lines. **n**, Change-point analysis of NM_001278549 (*PDK1* mRNA) identified change-points at positions 53 and 79 nt upstream of the start codon (left), which define shorter sequence segments with isoforms that are enriched and depleted with C48 treatment (right). **o**, Scatter-plot of protein (from GPF-DIA proteomics analysis) versus total mRNA log₂ fold changes in T47D cells (C48 versus DMSO) from Fig. 5k with PDK1 indicated. Anota2seq analysis identified a significant increase in protein level, independent of mRNA level. Benjamini-Hochberg Adjusted *FDR* = 0.0917 (*n* = 4 independent experiments) **p**, PCA of quantification of ATAC-seq peaks at TSSs in T47D cells treated with C48 or DMSO for 24 h (*n* = 2 independent experiments). Source numerical data and unprocessed gels are available in.

Dr. Ola Larsson

# Reporting Summary

## Statistics

For all statistical analyses, confirm that the following items are present in the figure legend, table legend, main text, or Methods section.

| n/a | Confirmed | |
|---|---|---|
| ☐ | ☒ | The exact sample size (*n*) for each experimental group/condition, given as a discrete number and unit of measurement |
| ☐ | ☒ | A statement on whether measurements were taken from distinct samples or whether the same sample was measured repeatedly |
| ☐ | ☒ | The statistical test(s) used AND whether they are one- or two-sided<br>*Only common tests should be described solely by name; describe more complex techniques in the Methods section.* |
| ☐ | ☒ | A description of all covariates tested |
| ☐ | ☒ | A description of any assumptions or corrections, such as tests of normality and adjustment for multiple comparisons |
| ☐ | ☒ | A full description of the statistical parameters including central tendency (e.g. means) or other basic estimates (e.g. regression coefficient) AND variation (e.g. standard deviation) or associated estimates of uncertainty (e.g. confidence intervals) |
| ☐ | ☒ | For null hypothesis testing, the test statistic (e.g. *F*, *t*, *r*) with confidence intervals, effect sizes, degrees of freedom and *P* value noted<br>*Give P values as exact values whenever suitable.* |
| ☒ | ☐ | For Bayesian analysis, information on the choice of priors and Markov chain Monte Carlo settings |
| ☒ | ☐ | For hierarchical and complex designs, identification of the appropriate level for tests and full reporting of outcomes |
| ☐ | ☒ | Estimates of effect sizes (e.g. Cohen's *d*, Pearson's *r*), indicating how they were calculated |

*Our web collection on statistics for biologists contains articles on many of the points above.*

## Software and code

Policy information about availability of computer code

| | |
|---|---|
| Data collection | Western blot data was collected using ImageStudio Acquisition Software (version 5.2.0) from LICOR (https://www.licor.com/bio/image-studio/).<br>NanoCAGE and Smartseq2 data were acquired using HiSeq Control Software (veriosn 2.2.58/RTA 1.18.64), and NovaSeq Control Software (version 1.6.0)<br>Raw proteomics data were converted to mzML using ProteoWizard Software (version 3.0.22137)<br>GC/MS data acquisition used MassHunter (version 10.2) |
| Data analysis | All preexisting software used in these studies is described in detail in the methods section, including version numbers and relevant citations.<br><br>Preexisting Pipelines and Software:<br>nf-core chipseq pipeline (version 2.0.0)<br>nf-core atacseq pipeline (version 2.1.2)<br>Bowtie (version 1.2.2)<br>BEDtools (version 2.29.1)<br>NucleoATAC (version 0.3.4)<br>IGV (version 2.8.3)<br>DeepTools (version 3.4.3)<br>deepStats (version 0.3.1)<br>bcl2fastq v2.19<br>bcl2fastq v2.20.0.422<br>BBTools v36.59 (http://sourceforge.net/projects/bbmap/) |

HISAT2 (version 2.1.0)
Cytoscape (version 3.8.2.)
ClueGO (version 2.5.8)
TagDust (version 2.33)
Cutadapt (version 1.18)
STREME (version 5.4.1)
Mfold (version 3.6)
Perl script for removal of strand invasion artifacts (from Tang et al. PMID: 23180801)
R (version 4.1.1)
Perl (version 5.16.3)
Agilent MassHunter Quantitative Analysis Software (version 10.2)
QuantStudio Design & Analysis software (version 1.5.2)
QuantStudio Real-time PCR software (version 1.6.1)
DIA-NN (version 1.8.1)

R/Bioconductor Packages used in stand-alone analyses (listed in methods):
RSubread (version 2.6.4)
Anota2seq (version 1.14.0)
changepoint (version 2.2.3)

R/Bioconductor packages that are dependencies of custom software (listed on Code Ocean capsule)
Boruta (version 8.0.0)
R.utils (version 2.13.0)
ROCR (version 1.0-11)
WriteXLS (version 6.7.0)
caret (version 7.0-1)
curl (version 4.3.2)
data.table (version 1.17.0)
dplyr (version 1.1.4)
ggplot2 (version 3.5.1)
ggrepel (version 0.9.6)
gplots (version 3.2.0)
gridExtra (version 2.3)
igraph (version 2.1.4)
phia (version 0.2-1)
plotrix (version 3.8-4)
plyr (version 1.8.9)
qvalue (version 1.26.0)
randomForest (version 4.7-12)
reshape2 (version 1.4.4)
seqinr (version 4.2-36)
shades (version 1.4.0)
stringr (version 1.5.1)
vioplot (version 0.5.1)
grid (version 4.1.1)
rlist (version 0.4.6.2)
matrixStats (version 0.62.0)
RColorBrewer (version 1.1-3)
Hmisc (version 4.7-0)
DESeq2 (version 1.38.3)
RSamtools (version 2.8.0)
GenomicRanges (version 1.44.0)
IRanges (version 2.26.0)
edgR (version 3.34.1)
biomaRt (version 2.48.3)
memes (version 1.0.4)
pqsfinder (version 2.8.0)
limma (version 3.48.3)

Analysis of TSS switching, 5'UTR features, and translatome modelling was carried out using original code in the form of two R software packages. These original R packages, along with data and scripts to reproduce the analysis are available on Code Ocean: https://doi.org/10.24433/CO.4525673.v1

For manuscripts utilizing custom algorithms or software that are central to the research but not yet described in published literature, software must be made available to editors and reviewers. We strongly encourage code deposition in a community repository (e.g. GitHub). See the Nature Portfolio guidelines for submitting code & software for further information.

## Data

Policy information about availability of data

All manuscripts must include a data availability statement. This statement should provide the following information, where applicable:

- Accession codes, unique identifiers, or web links for publicly available datasets
- A description of any restrictions on data availability
- For clinical datasets or third party data, please ensure that the statement adheres to our policy

Raw and processed RNA-seq, ChIP-seq, ATAC-seq, and nanoCAGE data have been deposited in the NCBI GEO database under accession number GSE243418. All sequencing datasets were aligned to the NCBI RefSeq GRCh38/hg38 genome assembly, using corresponding RefSeq transcript annotations (release 109, 2020-11-20; https://ftp.ncbi.nlm.nih.gov/refseq/H_sapiens/annotation/annotation_releases/109.20201120/GCF_000001405.39_GRCh38.p13/). Proteomics data have been deposited in the PRIDE database under accession number PXD058655. Metabolite data have been deposited in the MetaboLights repository under accession number MTBLS12086. Other published datasets used in this study can be accessed at GSE11011 (eIF4G-dependent translation signature), GSE115142 (DAP5-dependent translation signature), and GSE76766 (mTOR-dependent translation signature). All other experimental data is provided in the associated Supplementary Data and Source Data files.

## Research involving human participants, their data, or biological material

Policy information about studies with human participants or human data. See also policy information about sex, gender (identity/presentation), and sexual orientation and race, ethnicity and racism.

| Reporting on sex and gender | N/A |
|---|---|
| Reporting on race, ethnicity, or other socially relevant groupings | N/A |
| Population characteristics | N/A |
| Recruitment | N/A |
| Ethics oversight | N/A |

Note that full information on the approval of the study protocol must also be provided in the manuscript.

# Field-specific reporting

Please select the one below that is the best fit for your research. If you are not sure, read the appropriate sections before making your selection.

☒ Life sciences    ☐ Behavioural & social sciences    ☐ Ecological, evolutionary & environmental sciences

For a reference copy of the document with all sections, see nature.com/documents/nr-reporting-summary-flat.pdf

# Life sciences study design

All studies must disclose on these points even when the disclosure is negative.

| Sample size | No statistical methods were used to pre-determine sample sizes, but sample sizes are comparable to those previously published (PMID 32427827) and standard in the field. All analyses were performed on data from 3 or more independent experiments, with the exception of isoform-selective RT-qPCR data presented in Figure 7f, and protein/mRNA quantifications for CRISPR clones presented in Figure 7j, where 2 independent experiments were performed. In these instances, an ANOVA was used to identify if expression/translation of all isoforms is the same, or different. As such, the residual degrees of freedom in the models for data in Figures 7f and j were 6, and 16, respectively, which provide sufficient statistical power for this approach to be valid. Furthermore, for data presented in Figure 7j, multiple CRISPR clones were used for each 5'UTR isoform. Therefore, in addition to independent experimental replicates, we also have included biological replicates. |
|---|---|
| Data exclusions | Data from one sample of nanoCAGE sequencing in T47D cells under normoxia was excluded due to sequencing failure. This was determined by a lack of sequencing reads after data processing and alignment. Data from one replicate of nanoCAGE sequencing in H9 cells under hypoxia and normoxia was excluded due to being an outlier upon PCA. |
| Replication | All attempts to replicate data were successful. The number of repeats is described in the figure legends. For sequencing-based data, the reproducibility of replicates was assessed using Principle Component Analysis, hierarchical clustering, and correlation analyses between samples. |
| Randomization | No randomization was performed in this study. Covariates were controlled for in differential expression analyses by including batch/replicate in linear regression models (detailed in the methods). |
| Blinding | Blinding was not relevant to the study design as none of the analyses involved any type of scoring that could be subjective, or vary between individuals carrying out the analysis. |

# Reporting for specific materials, systems and methods

We require information from authors about some types of materials, experimental systems and methods used in many studies. Here, indicate whether each material, system or method listed is relevant to your study. If you are not sure if a list item applies to your research, read the appropriate section before selecting a response.

## Materials & experimental systems

| n/a | Involved in the study |
|-----|-----------------------|
| ☐ | ☒ Antibodies |
| ☐ | ☒ Eukaryotic cell lines |
| ☒ | ☐ Palaeontology and archaeology |
| ☒ | ☐ Animals and other organisms |
| ☒ | ☐ Clinical data |
| ☒ | ☐ Dual use research of concern |
| ☒ | ☐ Plants |

## Methods

| n/a | Involved in the study |
|-----|-----------------------|
| ☐ | ☒ ChIP-seq |
| ☐ | ☒ Flow cytometry |
| ☒ | ☐ MRI-based neuroimaging |

## Antibodies

**Antibodies used**

4E-BP1, phospho-S65 (Cell signaling; Cat# 9451S; 1:1000)
4E-BP1 (35H11) (Cell signaling; Cat# 9644; 1:7000)
S6 ribosomal protein, phospho-S240/244 (Cell signaling; Cat#2215; 1:1000)
S6 ribosomal protein (5G10) (Cell signaling; Cat# 2217; 1:1000)
EIF2S1 [E90] phospho-S51 (Abcam; Cat# ab32157; 1:1000)
eIF2alpha (Cell signaling; Cat# 9722; 1:1000)
HIF1alpha (D1S7W) XP (Cell signaling; Cat# 36169; 1:1000)
PDK1 [4A11] (Abcam; Cat# ab110025; 1:1000)
PGK1 (Abcam; Cat# ab38007; 1:1000
PDHE1alpha (D6) (Santa Cruz; Cat# sc-277092; 1:1000)
PDHE1alpha phospho-S232 (EMD milipore; Cat# AP1063-50ug; 1:1000)
PDHE1alpha phospho-S293 [EPR12200] (Abcam; Cat# ab177461; 1:1000)
Histone H3 (D1H2) XP (Cell signaling; Cat# 4499S; 1:1000)
Histone H3K4me3 (C42D8) (Cell signaling; Cat# 9751S; 1:1000)
H3K4me3 (ChIPseq) (Epigentek; Cat #A4033; 6 ug per ChIP sample)
Alpha-tubulin (mouse) (LICOR; Cat# 926-42213; 1:4000)
Beta-tubulin (rabbit) (LICOR; Cat# 926-42211; 1:4000)
Beta-actin (mouse) (LICOR; Cat# 926-42212; 1:4000)
Beta-actin (rabbit) (LICOR; Cat# 926-42210; 1:4000)
IRDye 800CW anti-mouse (LICOR; Cat# 926-32212; 1:10000)
IRDye 680RD anti-mouse (LICOR; Cat# 926-68072; 1:10000)
IRDye 800CW anti-rabbit (LICOR; Cat# 926-32213; 1:10000)
IRDye 680RD anti-rabbit (LICOR; Cat# 926-68073; 1:10000)

**Validation**

4E-BP1, phospho-S65 Cell signaling Cat# 9451S, 1:1000, Western blot analysis of extracts from 293 cells that were starved for 24 hours in serum-free medium and underwent a 1-hour amino acid deprivation. Amino acids were replenished for 1 hour. Cells were then either untreated or treated with 100 nM insulin for 30 minutes.

4E-BP1 (35H11) Cell signaling Cat# 9644, 1:7000, Western blot analysis of extracts from control HeLa cells or HeLa cells with a targeted mutation in the gene encoding 4E-BP1.

S6 ribosomal protein, phospho-S240/244 Cell signaling Cat#2215, 1:1000, Western blot analysis of extracts from 293 cells, untreated or treated with 20% FBS for 10, 20 or 30 min.

S6 ribosomal protein (5G10) Cell signaling Cat# 2217, 1:1000, Western blot analysis of extracts from HeLa, NIH/3T3, PC12 and COS cells.

EIF2S1 [E90] phospho-S51 Abcam Cat# ab32157, 1:1000, Western blot of whole cell lysates of RAW 264.7 treated with 5 ug/ml tunicamycin for 18 hours (positive control) or treated with 5 ug/ml tunicamycin for 18 hours and treated with alkaline phosphatase or. Lambda phosphatase (negative control).

eIF2alpha Cell signaling Cat# 9722, 1:1000, Western blot analysis of extracts from PC12 cells.

HIF1alpha (D1S7W) XP Cell signaling Cat# 36169, 1:1000, Western blot analysis of extracts from Hep G2 cells untreated or treated with cobalt chloride (100 μM, 4 h; +), Raji cells untreated or treated with cobalt chloride (100 μM, 4 h; +) and U-2 OS cells untreated or treated with DMOG (1 mM, 6 h; +)

PDK1 [4A11] Abcam Cat# ab110025, 1:1000, Western blot analysis of extracts from NIH/3T3, HeLa, Jurkat, HepG2,PC-12, COS7 cells.

PGK1 Abcam Cat# ab38007, 1:1000, discontinued, Western blot analysis of extracts from mouse stomach tissue, HepG2 cells.

PDHE1alpha (D6) Santa Cruz Cat# sc-277092,1:1000, Western blot analysis of PDH-E1α expression in Hep G2, HeLa, Sol8, C2C12, L8, L6, EOC 20, H4, IMR-3 and Hep G2 whole cell lysates.

PDHE1alpha phospho-S232 EMD milipore Cat# AP1063-50ug, 1:1000,  Whole tissue extract from mouse liver (20 μg) left untreated or treated with dichloroacetate (DCA, 5 mM for 4 h).

PDHE1alpha phospho-S293 [EPR12200] Abcam Cat# ab177461, 1:1000,  Western blot analysis of extracts from rat kidneys untreated or treated with phosphatase,  or HT-29 cells treated with 8 mM Sodium butyrate for 24 hours without or with phosphatase treatment.

Histone H3 (D1H2) XP Cell signaling Cat# 4499S, 1:1000, Western blot analysis of extracts from HeLa,  NIH/3T3, C6, COS cells.

Histone H3K4me3 (C42D8, for WB) Cell signaling Cat# 9751S, 1:1000, HeLa and NIH/3T3 cell lysates were probed with Tri-Methyl Histone H3 (Lys4) (C42D8) Rabbit mAb or Tri-Methyl Histone H3 (Lys4) Rabbit mAb pre-adsorbed with 1.5 μM of various competitor peptides, only the tri-methyl histone H3 (Lys4) peptide competed away binding of the antibody.

H3K4me3 (ChIPseq) Epigentek Cat #A4033, 6μg per ChIP sample,  Chromatin immunoprecipitation analysis of extracts of 293F cells was performed using Histone H3K4me3 (H3K4 Trimethyl) Polyclonal Antibody. The amount of immunoprecipitated DNA was checked by quantitative PCR. Additional validation was performed in PMID: 30244833.

# Eukaryotic cell lines

Policy information about cell lines and Sex and Gender in Research

| Cell line source(s) | T47D ductal carcinoma cells, female; ATCC (HTB-133) <br> HEK 293T cells, female; ATCC (CRL-3216) <br> H9 human embryonic stem cells, female; WiCell (WA09) |
|---|---|
| Authentication | All cell lines were authenticated by STR profiling at the SickKids Research Institute (Toronto, Ontario, Canada). |
| Mycoplasma contamination | All cells routinely tested negative for mycoplasma (Mycoplasma detection kit, ATCC). |
| Commonly misidentified lines <br> (See ICLAC register) | No commonly misidentified cell lines were used in this study. |

# Plants

| Seed stocks | N/A |
|---|---|
| Novel plant genotypes | N/A |
| Authentication | N/A |

# ChIP-seq

## Data deposition

☒ Confirm that both raw and final processed data have been deposited in a public database such as GEO.

☒ Confirm that you have deposited or provided access to graph files (e.g. BED files) for the called peaks.

| Data access links <br> *May remain private before publication.* | All raw and processed ChIP-seq data generated in this study is available on the GEO database with accession GSE243418. |
|---|---|
| Files in database submission | Raw sequencing data: <br> H9_H_H3K4me3_1_R1.fastq.gz <br> H9_H_H3K4me3_1_R2.fastq.gz <br> H9_H_H3K4me3_2_R1.fastq.gz <br> H9_H_H3K4me3_2_R2.fastq.gz <br> H9_H_H3K4me3_3_R1.fastq.gz <br> H9_H_H3K4me3_3_R2.fastq.gz <br> H9_H_input_1_R1.fastq.gz <br> H9_H_input_1_R2.fastq.gz <br> H9_H_input_2_R1.fastq.gz <br> H9_H_input_2_R2.fastq.gz |

```
H9_H_input_3_R1.fastq.gz
H9_H_input_3_R2.fastq.gz
H9_N_H3K4me3_1_R1.fastq.gz
H9_N_H3K4me3_1_R2.fastq.gz
H9_N_H3K4me3_2_R1.fastq.gz
H9_N_H3K4me3_2_R2.fastq.gz
H9_N_H3K4me3_3_R1.fastq.gz
H9_N_H3K4me3_3_R2.fastq.gz
H9_N_input_1_R1.fastq.gz
H9_N_input_1_R2.fastq.gz
H9_N_input_2_R1.fastq.gz
H9_N_input_2_R2.fastq.gz
H9_N_input_3_R1.fastq.gz
H9_N_input_3_R2.fastq.gz
T47D_H_H3K4me3_1_R1.fastq.gz
T47D_H_H3K4me3_1_R2.fastq.gz
T47D_H_H3K4me3_2_R1.fastq.gz
T47D_H_H3K4me3_2_R2.fastq.gz
T47D_H_H3K4me3_3_R1.fastq.gz
T47D_H_H3K4me3_3_R2.fastq.gz
T47D_H_input_1_R1.fastq.gz
T47D_H_input_1_R2.fastq.gz
T47D_H_input_2_R1.fastq.gz
T47D_H_input_2_R2.fastq.gz
T47D_H_input_3_R1.fastq.gz
T47D_H_input_3_R2.fastq.gz
T47D_N_H3K4me3_1_R1.fastq.gz
T47D_N_H3K4me3_1_R2.fastq.gz
T47D_N_H3K4me3_2_R1.fastq.gz
T47D_N_H3K4me3_2_R2.fastq.gz
T47D_N_H3K4me3_3_R1.fastq.gz
T47D_N_H3K4me3_3_R2.fastq.gz
T47D_N_input_1_R1.fastq.gz
T47D_N_input_1_R2.fastq.gz
T47D_N_input_2_R1.fastq.gz
T47D_N_input_2_R2.fastq.gz
T47D_N_input_3_R1.fastq.gz
T47D_N_input_3_R2.fastq.gz
NS.X0096.004.IDT_i7_34---IDT_i5_34.DSP1951_05_24h_IP_rep1_R1.fastq.gz
NS.X0096.004.IDT_i7_34---IDT_i5_34.DSP1951_05_24h_IP_rep1_R2.fastq.gz
NS.X0096.004.IDT_i7_40---IDT_i5_40.DSP1951_11_24h_IP_rep2_R1.fastq.gz
NS.X0096.004.IDT_i7_40---IDT_i5_40.DSP1951_11_24h_IP_rep2_R2.fastq.gz
NS.X0096.004.IDT_i7_46---IDT_i5_46.DSP1951_17_24h_IP_rep3_R1.fastq.gz
NS.X0096.004.IDT_i7_46---IDT_i5_46.DSP1951_17_24h_IP_rep3_R2.fastq.gz
NS.X0096.004.IDT_i7_52---IDT_i5_52.DSP1951_23_24h_IP_rep4_R1.fastq.gz
NS.X0096.004.IDT_i7_52---IDT_i5_52.DSP1951_23_24h_IP_rep4_R2.fastq.gz
NS.X0096.004.IDT_i7_31---IDT_i5_31.DSP1951_02_24h-_input_rep1_R1.fastq.gz
NS.X0096.004.IDT_i7_31---IDT_i5_31.DSP1951_02_24h-_input_rep1_R2.fastq.gz
NS.X0096.004.IDT_i7_37---IDT_i5_37.DSP1951_08_24h_input_rep2_R1.fastq.gz
NS.X0096.004.IDT_i7_37---IDT_i5_37.DSP1951_08_24h_input_rep2_R2.fastq.gz
NS.X0096.004.IDT_i7_43---IDT_i5_43.DSP1951_14_24h_input_rep3_R1.fastq.gz
NS.X0096.004.IDT_i7_43---IDT_i5_43.DSP1951_14_24h_input_rep3_R2.fastq.gz
NS.X0096.004.IDT_i7_49---IDT_i5_49.DSP1951_20_24h_input_rep4_R1.fastq.gz
NS.X0096.004.IDT_i7_49---IDT_i5_49.DSP1951_20_24h_input_rep4_R2.fastq.gz
NS.X0096.004.IDT_i7_33---IDT_i5_33.DSP1951_04_0h_IP_rep1_R1.fastq.gz
NS.X0096.004.IDT_i7_33---IDT_i5_33.DSP1951_04_0h_IP_rep1_R2.fastq.gz
NS.X0096.004.IDT_i7_39---IDT_i5_39.DSP1951_10_0h_IP_rep2_R1.fastq.gz
NS.X0096.004.IDT_i7_39---IDT_i5_39.DSP1951_10_0h_IP_rep2_R2.fastq.gz
NS.X0096.004.IDT_i7_45---IDT_i5_45.DSP1951_16_0h_IP_rep3_R1.fastq.gz
NS.X0096.004.IDT_i7_45---IDT_i5_45.DSP1951_16_0h_IP_rep3_R2.fastq.gz
NS.X0096.004.IDT_i7_51---IDT_i5_51.DSP1951_22_0h_IP_rep4_R1.fastq.gz
NS.X0096.004.IDT_i7_51---IDT_i5_51.DSP1951_22_0h_IP_rep4_R2.fastq.gz
NS.X0096.004.IDT_i7_30---IDT_i5_30.DSP1951_01_0h_input_rep1_R1.fastq.gz
NS.X0096.004.IDT_i7_30---IDT_i5_30.DSP1951_01_0h_input_rep1_R2.fastq.gz
NS.X0096.004.IDT_i7_36---IDT_i5_36.DSP1951_07_0h_input_rep2_R1.fastq.gz
NS.X0096.004.IDT_i7_36---IDT_i5_36.DSP1951_07_0h_input_rep2_R2.fastq.gz
NS.X0096.004.IDT_i7_42---IDT_i5_42.DSP1951_13_0h_input_rep3_R1.fastq.gz
NS.X0096.004.IDT_i7_42---IDT_i5_42.DSP1951_13_0h_input_rep3_R2.fastq.gz
NS.X0096.004.IDT_i7_48---IDT_i5_48.DSP1951_19_0h_input_rep4_R1.fastq.gz
NS.X0096.004.IDT_i7_48---IDT_i5_48.DSP1951_19_0h_input_rep4_R2.fastq.gz

Processed data, aligned to human 5'UTR regions:
H9_H_H3K4me3_R1.mLb.clN.sorted.trimmed95.cdf
H9_H_H3K4me3_R2.mLb.clN.sorted.trimmed95.cdf
H9_H_H3K4me3_R3.mLb.clN.sorted.trimmed95.cdf
H9_H_input_R1.mLb.clN.sorted.trimmed95.cdf
```

H9_H_input_R2.mLb.clN.sorted.trimmed95.cdf
H9_H_input_R3.mLb.clN.sorted.trimmed95.cdf
H9_N_H3K4me3_R1.mLb.clN.sorted.trimmed95.cdf
H9_N_H3K4me3_R2.mLb.clN.sorted.trimmed95.cdf
H9_N_H3K4me3_R3.mLb.clN.sorted.trimmed95.cdf
H9_N_input_R1.mLb.clN.sorted.trimmed95.cdf
II9_N_input_R2.mLb.clN.sorted.trimmed95.cdf
H9_N_input_R3.mLb.clN.sorted.trimmed95.cdf
T47D_H_H3K4me3_R1.mLb.clN.sorted.trimmed95.cdf
T47D_H_H3K4me3_R2.mLb.clN.sorted.trimmed95.cdf
T47D_H_H3K4me3_R3.mLb.clN.sorted.trimmed95.cdf
T47D_H_input_R1.mLb.clN.sorted.trimmed95.cdf
T47D_H_input_R2.mLb.clN.sorted.trimmed95.cdf
T47D_H_input_R3.mLb.clN.sorted.trimmed95.cdf
T47D_N_H3K4me3_R1.mLb.clN.sorted.trimmed95.cdf
T47D_N_H3K4me3_R2.mLb.clN.sorted.trimmed95.cdf
T47D_N_H3K4me3_R3.mLb.clN.sorted.trimmed95.cdf
T47D_N_input_R1.mLb.clN.sorted.trimmed95.cdf
T47D_N_input_R2.mLb.clN.sorted.trimmed95.cdf
T47D_N_input_R3.mLb.clN.sorted.trimmed95.cdf
C48_24hr_H3K4me3_rep1.mLb.clN.sorted.trimmed95.cumsum.cdf
C48_24hr_H3K4me3_rep2.mLb.clN.sorted.trimmed95.cumsum.cdf
C48_24hr_H3K4me3_rep3.mLb.clN.sorted.trimmed95.cumsum.cdf
C48_24hr_H3K4me3_rep4.mLb.clN.sorted.trimmed95.cumsum.cdf
C48_24hr_input_rep1.mLb.clN.sorted.trimmed95.cumsum.cdf
C48_24hr_input_rep2.mLb.clN.sorted.trimmed95.cumsum.cdf
C48_24hr_input_rep3.mLb.clN.sorted.trimmed95.cumsum.cdf
C48_24hr_input_rep4.mLb.clN.sorted.trimmed95.cumsum.cdf
C48_0hr_H3K4me3_rep1.mLb.clN.sorted.trimmed95.cumsum.cdf
C48_0hr_H3K4me3_rep2.mLb.clN.sorted.trimmed95.cumsum.cdf
C48_0hr_II3K4me3_rep3.mLb.clN.sorted.trimmed95.cumsum.cdf
C48_0hr_H3K4me3_rep4.mLb.clN.sorted.trimmed95.cumsum.cdf
C48_0hr_input_rep1.mLb.clN.sorted.trimmed95.cumsum.cdf
C48_0hr_input_rep2.mLb.clN.sorted.trimmed95.cumsum.cdf
C48_0hr_input_rep3.mLb.clN.sorted.trimmed95.cumsum.cdf
C48_0hr_input_rep4.mLb.clN.sorted.trimmed95.cumsum.cdf

Processed data, called peaks:
H9_H_H3K4me3_rep1_peaks.narrowPeak
H9_H_H3K4me3_rep2_peaks.narrowPeak
H9_H_H3K4me3_rep3_peaks.narrowPeak
H9_N_H3K4me3_rep1_peaks.narrowPeak
H9_N_H3K4me3_rep2_peaks.narrowPeak
H9_N_H3K4me3_rep3_peaks.narrowPeak
T47D_H_H3K4me3_rep1_peaks.narrowPeak
T47D_H_H3K4me3_rep2_peaks.narrowPeak
T47D_H_H3K4me3_rep3_peaks.narrowPeak
T47D_N_H3K4me3_rep1_peaks.narrowPeak
T47D_N_H3K4me3_rep2_peaks.narrowPeak
T47D_N_H3K4me3_rep3_peaks.narrowPeak
C48_24hr_H3K4me3_rep1_peaks.narrowPeak
C48_24hr_H3K4me3_rep2_peaks.narrowPeak
C48_24hr_H3K4me3_rep3_peaks.narrowPeak
C48_24hr_H3K4me3_rep4_peaks.narrowPeak
C48_0hr_H3K4me3_rep1_peaks.narrowPeak
C48_0hr_H3K4me3_rep2_peaks.narrowPeak
C48_0hr_H3K4me3_rep3_peaks.narrowPeak
C48_0hr_H3K4me3_rep4_peaks.narrowPeak

**Genome browser session**
(e.g. UCSC)

No longer applicable.

## Methodology

**Replicates**

N = 3 for each of two cell lines (T47D and H9) in hypoxia and normoxia. N = 4 for T47D cells treated with either DMSO (0 h) or compound-48 (C48) for 24 h. Reproducibility of the replicates was assessed using Principle Component Analysis (results provided in Extended Data Figure 6), hierarchical clustering, and correlations between samples.

**Sequencing depth**

Reads are 101bp, with a paired-end setup. Total raw reads are as follows:

H9_H_H3K4me3_R1: 74635732
H9_H_H3K4me3_R2: 80219516
H9_H_H3K4me3_R3: 78220806
H9_H_input_R1: 295713012
H9_H_input_R2: 255300374
H9_H_input_R3: 214645350

H9_N_H3K4me3_R1: 70275862
H9_N_H3K4me3_R2: 70588338
H9_N_H3K4me3_R3: 67492356
H9_N_input_R1: 223681710
H9_N_input_R2: 186776302
H9_N_input_R3: 228174104
T47D_H_H3K4me3_R1: 77663460
T47D_H_H3K4me3_R2: 79771312
T47D_H_H3K4me3_R3: 80211274
T47D_H_input_R1: 277232374
T47D_H_input_R2: 271966508
T47D_H_input_R3: 172351612
T47D_N_H3K4me3_R1: 86659118
T47D_N_H3K4me3_R2: 68534990
T47D_N_H3K4me3_R3: 88714902
T47D_N_input_R1: 274816836
T47D_N_input_R2: 312448620
T47D_N_input_R3: 258789032
C48_0hr_H3K4me3_rep1: 84394309
C48_0hr_H3K4me3_rep2: 79764369
C48_0hr_H3K4me3_rep3: 86199711
C48_0hr_H3K4me3_rep4: 76325166
C48_0hr_input_rep1: 90571962
C48_0hr_input_rep2: 80991338
C48_0hr_input_rep3: 77365219
C48_0hr_input_rep4: 88295985
C48_24hr_H3K4me3_rep1: 95161865
C48_24hr_H3K4me3_rep2: 78654392
C48_24hr_H3K4me3_rep3: 90864553
C48_24hr_H3K4me3_rep4: 81468184
C48_24hr_input_rep1: 89732670
C48_24hr_input_rep2: 94003924
C48_24hr_input_rep3: 88090559
C48_24hr_input_rep4: 88702605
C48_48hr_H3K4me3_rep1: 88191753
C48_48hr_H3K4me3_rep2: 83774451
C48_48hr_H3K4me3_rep3: 82371767
C48_48hr_H3K4me3_rep4: 79573865
C48_48hr_input_rep1: 91941255
C48_48hr_input_rep2: 114668477
C48_48hr_input_rep3: 95640688
C48_48hr_input_rep4: 70614428

Uniquely mapped reads are as follows (following removal of ENCODE blacklist regions):
H9_H_H3K4me3_R1: 37317866
H9_H_H3K4me3_R2: 40109758
H9_H_H3K4me3_R3: 39110403
H9_H_input_R1: 147856506
H9_H_input_R2: 127650187
H9_H_input_R3: 107322675
H9_N_H3K4me3_R1: 35137931
H9_N_H3K4me3_R2: 35294169
H9_N_H3K4me3_R3: 33746178
H9_N_input_R1: 111840855
H9_N_input_R2: 93388151
H9_N_input_R3: 114087052
T47D_H_H3K4me3_R1: 38831730
T47D_H_H3K4me3_R2: 39885656
T47D_H_H3K4me3_R3: 40105637
T47D_H_input_R1: 138616187
T47D_H_input_R2: 135983254
T47D_H_input_R3: 86175806
T47D_N_H3K4me3_R1: 43329559
T47D_N_H3K4me3_R2: 34267495
T47D_N_H3K4me3_R3: 44357451
T47D_N_input_R1: 137408418
T47D_N_input_R2: 156224310
T47D_N_input_R3: 129394516
T47D_N_input_R3_T1 145243816
C48_0hr_H3K4me3_rep1 65610320
C48_0hr_H3K4me3_rep2 62061305
C48_0hr_H3K4me3_rep3 66398742
C48_0hr_H3K4me3_rep4 59306964
C48_0hr_input_rep1 72241012
C48_0hr_input_rep2 64409241

C48_0hr_input_rep3 62040919
C48_0hr_input_rep4 70378628
C48_24hr_H3K4me3_rep1 73383442
C48_24hr_H3K4me3_rep2 62086313
C48_24hr_H3K4me3_rep3 70315745
C48_24hr_H3K4me3_rep4 63418790
C48_24hr_input_rep1 71742298
C48_24hr_input_rep2 75450342
C48_24hr_input_rep3 70456718
C48_24hr_input_rep4 70800600
C48_48hr_H3K4me3_rep1 68261352
C48_48hr_H3K4me3_rep2 65382533
C48_48hr_H3K4me3_rep3 64106221
C48_48hr_H3K4me3_rep4 62782519
C48_48hr_input_rep1 73325561
C48_48hr_input_rep2 91978308
C48_48hr_input_rep3 76865323
C48_48hr_input_rep4 56512534

**Antibodies**

The antibody used for ChIP-seq was H3K4me3, Epigentek Cat# A4033.

**Peak calling parameters**

The results presented in the study did not employ traditional peak-calling-based analysis of ChIP-seq data. Our methods are describe in detail in the methods section, and all processed files required to reproduce the analysis are provided with the GEO submission above. However, the quality of the data was assessed by the traditional metrics based on peak-calling with MACS2. To do this, the nf-core ChIP-seq pipeline (version 2.0.0) was used with default settings, using the command:
nextflow run nf-core/chipseq --fasta GCF_000001405.39_GRCh38.p13_genomic.fna --gtf
GCF_000001405.39_GRCh38.p13_genomic.gtf --macs_gsize 2805636231 --blacklist hg38-blacklist.v2.refseq.bed  ---aligner star --narrow_peak --macs_fdr 0.15 --min_reps_consensus 3 (n replicates - 1)

**Data quality**

The quality of the data was assessed using fingerprint plots and read distribution profiles after alignment, annotation and filtering.

The number of peaks called by MACS2 passing thresholds are as follows:
T47D_H_H3K4me3_R1: 73993
T47D_H_H3K4me3_R2: 54848
T47D_H_H3K4me3_R3: 57825
T47D_N_H3K4me3_R1: 42231
T47D_N_H3K4me3_R2: 30953
T47D_N_H3K4me3_R3: 38387
H9_H_H3K4me3_R1: 54884
H9_H_H3K4me3_R2: 24479
H9_H_H3K4me3_R3: 23075
H9_N_H3K4me3_R1:64942
H9_N_H3K4me3_R2: 39403
H9_N_H3K4me3_R3: 25606
C48_0hr_H3K4me3_rep1: 112791
C48_0hr_H3K4me3_rep2: 111564
C48_0hr_H3K4me3_rep3: 112058
C48_0hr_H3K4me3_rep4: 117266
C48_24hr_H3K4me3_rep1: 135962
C48_24hr_H3K4me3_rep2: 134304
C48_24hr_H3K4me3_rep3: 131786
C48_24hr_H3K4me3_rep4: 135549

**Software**

Raw H3K4me3 fastq files were processed using the nf-core/chipseq pipeline v.2.0.0 (available at https://github.com/nf-core/chipseq). Reads were aligned to the NCBI RefSeq GRCh38/hg38 genome assembly (release 109, 2020-11-20; https://ftp.ncbi.nlm.nih.gov/refseq/H_sapiens/annotation/annotation_releases/109.20201120/GCF_000001405.39_GRCh38.p13/) in paired-end mode with the following settings: --macs_gsize 2805636231 --blacklist hg38-blacklist.v2.refseq.bed  ---aligner star --narrow_peak --macs_fdr 0.15 --min_reps_consensus 3 (n replicates - 1). The output of the pipeline includes BAM files with aligned reads with duplicates, multi-mapping, unpaired, and ENCODE blacklist reads filtered out. In order to assess shifts in the position of H3K4me3 marks relative to TSSs of protein-coding genes, filtered aligned reads were extracted from the BAM files and re-aligned to a custom index of 5'UTR genomic sequences using Bowtie (version 1.2.2) (settings: -v 2 -X 1000 -a). Bedtools genomeCoverageBed (v2.29.1) was then used to compute the genome coverage of H3K4me3 with -dz and -pc options, and -scale to normalize for library size. The cumulative sum of H3K4me3 coverage was calculated for each genomic region corresponding to RefSeq transcripts, and genomic regions were trimmed at the nucleotide position where 95% of the cumulative H3K4me3 signal had already occurred. All fully process ChIP-seq files used in the analysis are provided at the GEO accession GSE243418.

# Flow Cytometry

## Plots

Confirm that:

☒ The axis labels state the marker and fluorochrome used (e.g. CD4-FITC).

☒ The axis scales are clearly visible. Include numbers along axes only for bottom left plot of group (a 'group' is an analysis of identical markers).

☒ All plots are contour plots with outliers or pseudocolor plots.

☒ A numerical value for number of cells or percentage (with statistics) is provided.

## Methodology

| | |
|---|---|
| Sample preparation | See Methods section for origin and genetic manipulation of T47D cells. For fluorescence-activated cell sorting (FACS) cells were passaged three times after stable transduction, collected after trypsinization, and resuspended in PBS containing 2% FBS. Subsequently, cells were delivered to Lady Davis Institute (LDI) Flow Cytometry Facility for sorting. Approximately 1 x 104 cells were sorted at 30 psi with a 100 µm nozzle into duplicate wells of a 6-well plate containing 2 mL of growth media. |
| Instrument | BD FACSAria Fusion cell sorter |
| Software | FACSDiva Software Version 8.0.2 |
| Cell population abundance | ~ 2.0e4 |
| Gating strategy | Gating strategy was determined by the Lady Davis Institute Flow Cytometry Facility, where successive gating identified FSC Singlets, SSC Singlets, and finally GFP positive cells (see explanatory panels in Supplementary Fig. 1 in Supplementary Information). |

☒ Tick this box to confirm that a figure exemplifying the gating strategy is provided in the Supplementary Information.

