## [Peer Review File · Nature Cell Biology]

Epigenetic alterations facilitate transcriptional and translational programs in hypoxia

Corresponding Author: Dr Lynne Postovit

Version 0:

Decision Letter:

Dear Dr Postovit,

Thank you for your interest in submitting your work to Nature Cell Biology.

I have discussed the information you provided with my colleagues, and we think that the study sounds interesting and could be appropriate for this journal. However, given the limited information provided, we would like to evaluate the complete manuscript before deciding whether to formally review it.

Please use this link to submit the complete manuscript:

Link Redacted

Please feel free to contact me if you have any questions.

Kind regards,

Sabrya Carim

Sabrya Carim, PhD
(she/her/hers)
Associate Editor, Nature Cell Biology
Nature Portfolio

Springer Nature
The Campus, 4 Crinan Street, London N1 9XW, UK
sabrya.carim@springernature.com
<https://orcid.org/0000-0001-9485-1938>

Version 1:

Decision Letter:

Dear Dr Postovit,

Your manuscript "Epigenetic coordination of transcriptional and translational programs in hypoxia", has now been seen by 3 referees, who are experts in hypoxia (referee 1); epigenetics and transcription (referee 2); and translation in stress (referee 3), and whose comments are pasted below. In light of their advice, we regret that we cannot offer to publish the study in Nature Cell Biology.

As you will see, although the reviewers find this work interesting, they raise a number of concerns that question the strength of the data and of the conclusions that can be drawn, and in light of the points they raise, we find the present data-set too preliminary to pursue at this stage.

In particular, among the limitations of the dataset the referees note limited insight into the mechanism by which an increase in H3K4me3 levels should lead to changes in TSS selection and how changes in TSS leads to translational control.

We would be open to the possibility of considering a revised manuscript that would fully address the referee concerns. However, any decision to re-review such a revised study would depend on the strength of the revisions and the published literature at the time of resubmission.

We would be happy to consult with our colleagues at Nature Communications and EMBO journals to see whether they would be interested in taking this manuscript further with the existing peer review history. Please do let me know if you would like us to pursue this option in the new year.

Although we cannot publish your paper, it may be appropriate for another journal in the Nature Portfolio. If you wish to explore the journals and transfer your manuscript please use our manuscript transfer portal. You will not have to re-supply manuscript metadata and files, unless you wish to make modifications. For more information, please see our http://www.nature.com/authors/author_resources/transfer_manuscripts.html?WT.mc_id=EMI_NPG_1511_AUTHORTRANSF&WT.ec_id=AUTHOR manuscript transfer FAQ page.

We are very sorry that we could not be more positive on this occasion, but we thank you for the opportunity to consider this work.

With kind regards,
Sabrya Carim

Sabrya Carim, PhD
(she/her/hers)
Associate Editor, Nature Cell Biology
Nature Portfolio

Springer Nature
The Campus, 4 Crinan Street, London N1 9XW, UK
sabrya.carim@springernature.com
<https://orcid.org/0000-0001-9485-1938>

Reviewers' comments:

Reviewer #1 (Remarks to the Author):

This is an exciting paper that has explored the effects of hypoxia on changes in transcription and translation through a series of sophisticated and well performed biological and analytical methods. The paper reveals at least three important and novel insights.

First, the authors have revealed a substantial change in TSS sites utilized by the transcriptional machinery during hypoxia. Through sequencing of UTR's under each condition, the data convincingly demonstrate changes in large numbers of genes in two different cell lines. This is a novel and interesting observation that is further explored in the paper. They go on to show these differential TSS result in substantial changes in UTR composition and are able to map these changes to some of the known mechanisms of mRNA translation regulation under hypoxia including mTOR regulation and ISR/UPR activation.

Second, the paper explores in detail, at the gene level, the impact of hypoxia on the efficiency of mRNA translation. While it has been known for many years that hypoxia inhibits mRNA translation through mTOR inhibition and UPR/ISR activation, there is relatively little experimental data that examines the direct impact of those effect on the translation of individual mRNAs. This is due in part to the technical complexity of both the biological and analytical methods. The authors should be congratulated on the quality of the work that was done and as a result the results are convincing. This unique gene level mRNA translation dataset has allowed the authors to examine the relationship between translation, TSS switching, and specific UTR elements. This analysis reveals that TSS switching drives a component of the change in translation and have linked this to the acquisition or loss of various UTR elements.

Third, and perhaps most importantly, the authors explore a potential mechanism that underlies the changes in TSS, UTR composition, and mRNA translation efficiency by mapping hypoxia induced changes in H3K4me3. Previous work has shown that hypoxia causes increases in the levels and genomic distribution of this mark through inhibition of KDM5, an enzyme required for its removal. Here, the authors demonstrate changes in H3K4me3 around TSS that correlate with changes in TSS in at least some of the hypoxia altered genes. Furthermore, they demonstrate concordant changes in a fraction (27%) of genes that show H3K4me3 changes under hypoxia in cells treated with the KDM5 inhibitor alone.

Finally, PDK1 is used as a key example that demonstrates that a key hypoxia adaptive gene is influenced by TSS start site changes. The data here are also convincing, including the additional methods used to validate the UTR changes and mRNA translation changes.

Overall, this manuscript provides a significant advance to the field, and connects and extend previous discoveries of gene regulation during hypoxia that had already identified contributions from regulation of the mRNA translation machinery and the epigenetic machinery. This is the first paper to show a substantial impact on TSS start site and its consequential impact on UTR composition and mRNA translation. It also highlights the importance of further exploring and understanding histone

methylation changes that drive at least a component of the TSS changes.

Major comments:

1) The authors have selected two rather unrelated cell lines and have conducted their experiments using different hypoxia exposures. The basis for this is unclear and not explained in the manuscript. The ability to compare and contrast these cell lines would be much stronger if the experimental conditions used to generate hypoxia adaptation were the same. The rationale for this needs to be addressed.

More importantly, the authors have not done a detailed comparison of the similarity or differences in these two lines. While the aggregate effects are similar (ie there are changes in TSS and changes in gene level mRNA translation) there is no comparison at the gene level on how these changes overlap. For example how many individual genes show common TSS shifts. When there is overlap, are the TSS changes the same or are they different? Similarly, how many individual genes between the two lines show common changes in mRNA translation efficiency? This more detailed analysis should be completed at the gene and UTR level.

This is important, because the impact of mTOR inhibition, ISR activation should impact individual genes to the same extent. However, it is less clear if one would expect shifts in TSS sites to be the same at the gene level when driven by changes in H3K4me3. Existing epigenetic marks are likely to vary across the two lines and thus the TSS switching may be highly cell type and cell line dependent. The data exist to explore and answer this to some degree.

2) The manuscript would be significantly strengthened by exploring in more detail the relationship between the H3K4me3 changes and the TSS changes during hypoxia at the individual gene/UTR level. The analyses are mostly done in aggregate and show significant changes in both and to some extent the overlap between upstream and downstream changes. However, the more interesting question is to examine how many genes show a defined H3K4me3 and TSS change, ie where a TSS switch can be directly attributed/correlated to a corresponding H3K4me3 change around that TSS site. What fraction of the TSS changes can this attribution be made, and for those genes what is the impact on mRNA translation relative to those where it is not?

3) A similar opportunity to explore specific changes in H3K4me3 and TSS change exist in the KDM5 experiments. The data comparing KDM5 inhibition to hypoxia resulted in an overlap of genes showing TSS changes of 27%. This is considered significant compared to a Monte Carlo simulation by random of 20%. However, what is key to understand in those 27% of genes is if the TSS switching and the H3K4me3 changes at the gene/sequence level are similar in genomic position. At a minimum this should look at upstream versus downstream, but it would be far more convincing if the actual TSS start sites and H3K4me3 changes were compared at the gene level here to see if the H3K4me3 change was driving this at the sequence context.

4) Finally, the authors should consider an effort to explore the biological relevance of these unique insights. Biological impact has not been investigated. The manuscript would have a much larger impact on the scientific if there was a demonstration that the TSS switching, driven by H3K4me3 changes leads to functionally relevant phenotypes under hypoxia. There are some great examples of how it is influencing the translation of key genes like PDK1 and PLOD2, but it is unclear to what magnitude this is actually affecting protein levels, enzyme activity, and most importantly cellular phenotype.

Minor comments:

Introduction

1) Page 2 Line 3 – this is a rather specific and unusual definition of plasticity

2) Page 2 Line 10 – there are earlier publications on the impact of hypoxia on the epigenome - PMID: 27800026

3) Page 2 Line 25 – the inhibition of translation during hypoxia is also due in large part to hypoxia induced ER stress (independent of energy demands)

Results

4) Page 5 Line 7 – Figure 2c and its description should be explained in more detail. There is an important normalization impact on the way that data is scaled given that hypoxia has caused a very massive overall suppression of protein synthesis of up to 80%. In this figure that impact is essentially normalized out, and thus it appears from figure 2c that there are similar overall changes in up and down regulated translation. The absolute amount of translation (ie that producing protein) will be shifted downward by the overall average inhibition.

Discussion

5) Page 12 Lines 36-37. It is stated: "This altered pool of 5'UTRs works in concert with reprogramming of the translational apparatus to mediate a survival-enhancing adaptive translational program." This has not been demonstrated in this manuscript. No exploration of the contribution of this unique mode of gene regulation on cellular phenotypes, including cell survival, have been evaluated. As it stands now, the discussion should focus more on the unique discoveries of gene regulation during hypoxia rather than on the impact this has on biological properties.

Reviewer #2 (Remarks to the Author):

Watt et al., address the potential function of hypoxia in regulating transcription and post-transcriptional programs and provide results suggesting that H3K4me3 is essential for remodeling of 5' UTR selection in hypoxia. Moreover, they suggest that this process is independent of HIF1 dependent transcriptional mechanisms and requires mTOR signaling.

Although the manuscript addresses an interesting topic - attempting to reveal how hypoxia links epigenetics to TSS selection, some of the data are ambiguous and there are multiple overstatements not well supported by experiments. In addition, the biological relevance of the reported observations is unclear, and mechanistic insights into how 5'UTR selection by H3K4me3 is regulated are lacking. Therefore, we are unable to recommend publication of the manuscript.

1. A major question is regarding the biological relevance of the observations. How important are the 5'-UTR switches? Most often, the same amount of protein is produced (they are not measured in the manuscript, however mRNAs associated with ribosomes are). The same protein is predicted to be produced as well because the difference in the 5'-UTR most often does not contain an alternative start codon. Therefore, although the observation of these different 5'-UTR is interesting, the biological impact of the small changes in the 5'UTRs remains unclear.
2. The authors try to link changes in TSS selection induced by hypoxia with H3K4me3 and KDM5 activity. Although the observations are interesting, the authors fail to provide mechanistic insights into why and how a general increase in H3K4me3 levels should lead to changes in TSS selection.
 - H3K4me3 is abundant at expressed genes. Why would a change in H3K4me3 levels change the TSS? The authors do not address that or provide results suggesting a potential mechanism.
 - KDM5 demethylation is not known to regulate nucleosome-specific localization of H3K4me3 – only the overall levels of H3K4me3. Instead of ChIP-seq, the authors should perform experiments that can give single nucleosome resolution. Would that show nucleosome specific changes in H3K4me3 levels and have the nucleosomes moved relative to the DNA.
3. The experimental approaches and analyses do not appear to be consistent throughout the manuscript.
 - NanoCAGE-seq, only two samples are shown for T47D-normoxia and three samples for hypoxia condition in Fig1b, however, five samples were found for H9 in Fig. S1b and four in Fig. S1a.
 - The authors switch between 0.5% for hypoxia and 1%. Why?
 - There is no big difference in nanoCAGE-seq between normoxia and hypoxia conditions (Fig. 1g-j), which may be the reason the authors use FDR < 0.15 for differential analysis. An FDR of this value may give a lot of false positives. The authors should test their results using a more normal FDR < 0.05.
 - Why is 48h hypoxia used for T47D and 24h for H9?
4. The authors claim there is a “high” degree of coordination between some pathways and 5'-UTR translation in hypoxia (Fig. 2 and Fig. S2,3), however, these claims are not supported by any experiments. Moreover, the differential co-regulated genes by 5 UTR and translation under hypoxia are few and more like a random event.

Reviewer #3 (Remarks to the Author):

All cells must be able to quickly respond to their environment by tuning gene expression. Cancer cells are exposed in the tumor microenvironment to hypoxia, and the ability of cancer cells to adapt through transcription and translation-regulated gene programs is correlated with higher cancer stem-cell phenotypes. Under hypoxia, cancer cells turn off general protein synthesis by blocking eIF4E and turning off eIF2a. Therefore, how stress response proteins are unregulated is a critical question. Here, authors use sequencing and bioinformatics to identify changes in transcription start site usage during hypoxia. They find that certain RNA features such as 5' UTR length or RNA motifs correlate with translation efficiency during hypoxia. They further find that there are epigenome changes around the hypoxia-regulated transcription start sites.

Authors note that “remarkably TSS switching is a previously unappreciated mechanism facilitating coordination between mRNA sequence features and translation initiation.” Given that there is little functional work in the manuscript, this is supposed to be the major novel finding of the paper. However, this statement is untrue. Indeed work from the Brar and Unal labs have previously revealed extensive transcription start site changes that lead to translational control (29474919,33826921,36622328,30016623), including during meiosis and ER stress. Therefore, excitement for the work is limited without substantial addition of functional studies including (1) proof that TSS switching is essential for cell survival during hypoxia, and (2) mechanism of action by which TSS switching (and in particular the new RNA motifs and RNA-binding proteins identified by bioinformatics analysis) leads to enhanced translation during shutoff.

Specific comments:

- It is unclear why the two different cell lines were chosen. The cell lines display different characteristics in terms of TSS usage (e.g. H9 cells do not display a specific preference for short or long UTRs) but the authors do not explain how these changes lead to differences in cell-type specific physiology during hypoxia.

- Authors use bioinformatics to identify different RNA motifs but provide no evidence that motifs are regulatory, for example authors should use reporter assays to show resistance to hypoxia-induced translation shutoff, and additionally show if these motifs are coordinately enriched in specific gene ontologies. Do the motifs contribute to changes in what RNA-binding proteins are found loaded onto the RNA during translation? Do the motifs contribute to structure? In addition, the authors should examine RNA structure, given substantial data implicating RNA structure to translational control.

-Authors propose a model where hypoxia leads to shorter UTRs that are able to more competitively use the translation machinery. However, 5' TOP RNAs are known to be very short and are translationally inhibited during hypoxia. Therefore this simplified model is not sufficient to explain the translation phenotype. Can authors explain?

- The only translation function data is in found in Figure 4, and does not make a lot of sense.

(1) 4F - RT-PCR is not able to distinguish between the isoforms correctly. As stated in the paper, the short isoform RT-PCR includes long isoform in the quantification. Therefore, authors should perform a northern blot to actually perform the experiment correctly.

(2) 4G - The way that the authors are interpreting this data is very confusing. Authors say that beta actin is the control cellular gene that is translationally shutoff during hypoxia. However, their shorter UTR (57 and 37nt) RNAs are also translationally down regulated to similar levels to beta-actin, and therefore the short UTR reporters are not showing a different "hypoxia resistant" phenotype.

At the same time, authors say that the shorter UTR RNA is more highly translated than the 134nt RNA. While this is true, again this is similar to levels of beta actin translation, which is the negative control. Therefore, this data instead shows that the UTR length of PDK1 is not sufficient to allow for resistance of shutoff during hypoxia and that this "RNA determinant" does not explain the ability of PDK1 to be translationally upregulated during hypoxia.

- There are no experiments showing that TSS switching functionally changes the hypoxia response. For example, authors could engineer a cell line which exclusively expresses a short versus long UTR version of PDK1 (or another identified gene of interest) and show that the cells are no longer able to survive during hypoxia.

- Given that the finding of TSS switching is not novel, authors need to show why the change in TSS leads to translational control. For example, it would be important to show how other initiation factors or RNA-binding proteins are involved, or even together with longer range interactions. Indeed, there is beautiful work from Uniacke...Lee (22678294) showing how a switch in usage of eIF4E to eIF4E2 regulates translation during hypoxia via an RNA binding site in the 3' UTR. Could TSS switching (via the enriched RNA motifs) be involved in this kind of well characterized translational control pathway? The existing bioinformatics based mining, although extensive, seems to have missed out on detecting known pathways and identifying new biochemical mechanisms.

- Authors do not do experiments to distinguish between why there are changes with TSS start site enrichment, but note it could transcription or stability. Authors should measure stability of target candidates using a pulse chase analysis and if it is stability, identify the RNA motifs (such as the proposed G/C richness) that contribute by using RNA mutants.

Minor comments:

- Figure 1B - why is one replicate discarded?

- Figure 2 - why are the conditions between Fig 1 and 2 changed from 0.5% to 1% O₂?

Fig S4 - It is confusing to annotate the gel "polysome" as it is a western blot from cytoplasmic lysate although matched with the polysomes (e.g. it's not protein isolated from a polysome)

Fig 4F - There needs to be error bars in the quantification and the matching polysome profile trace.

Fig 4G - This way of expressing the p value assignment is confusing, it's unclear what the comparison is. Please use the standard stars and lines to show comparisons.

**For Nature Portfolio general information and news for authors, see <http://npg.nature.com/authors>.

Version 2:

Decision Letter:

Our ref: NCB-A52589B-Z

8th May 2025

Dear Dr. Postovit,

Thank you for submitting your revised manuscript "Epigenetic coordination of transcriptional and translational programs in hypoxia" (NCB-A52589B-Z) and for your patience with the peer review process. It has now been seen by some of the original referees and their comments are below. Reviewer#3 was unable to review the revised manuscript and we sought advice from an arbitrator reviewer (Reviewer#4) who commented on the responses to reviewer#3's original concerns. The reviewers find that the paper has improved in revision, and therefore we'll be happy in principle to publish it in Nature Cell Biology, pending minor revisions to satisfy the referees' final requests and to comply with our editorial and formatting guidelines.

Please note that our articles must have 6 to 8 main figures and they can have up to 10 ED figures. Please convert all supplementary figures to extended data figures. Please ensure that all figures fit into a single standard page and adhere to a maximum page size of roughly 180mm wide x 200mm high. To ensure legibility once figures are re-sized, please use a font size of no smaller than 7pt Arial or Helvetica throughout the figures. Please incorporate the model figure (figure 8) into one of the main figures as we do not allow the model figure to be alone in a figure.

We are now performing detailed checks on your paper and will send you a checklist detailing our editorial and formatting requirements in about 2 weeks. **Please do not upload the final materials and make any revisions until you receive this additional information from us**.

Thank you again for your interest in Nature Cell Biology Please do not hesitate to contact me if you have any questions.

Sincerely,

Sabrya Carim, PhD
(she/her/hers)
Senior Editor, Nature Cell Biology
Nature Portfolio

Springer Nature
The Campus, 4 Crinan Street, London N1 9XW, UK
sabrya.carim@springernature.com
<https://orcid.org/0000-0001-9485-1938>

Reviewer #1 (Remarks to the Author):

The authors have thoughtfully and carefully addressed the criticisms raised in the initial review. The addition of new data demonstrating both mechanistic links between epigenetic changes and TSS, and the work demonstrating consequential downstream phenotypic changes significantly strengthen the manuscript.

The rationale for using different cell lines and different time points is appreciated but nonetheless leaves weaknesses in the ability to understand the overall importance of the specific changes identified in each line and their importance to our overall understanding of these observations. However, the additional analyses identifying many common patterns of regulation mitigate this.

This reviewer also commented on the responses to reviewer#2's previous concerns and their thoughts are below:

Reviewer#2's points on the revised manuscript:

1. Inconsistencies in Hypoxia Conditions: The authors still present experiments in which they have used varying hypoxia conditions, even within the same cell line. This raises concerns about the interpretations of the data.

>>Ref#1's thoughts: This remains a 'limitation' of the results but I do not agree that it raises significant concerns about the interpretation of the data. The authors responded to the same criticism that I made originally, and in their updated manuscript address this through looking at common changes across the two lines and different hypoxia conditions. I think it is unreasonable to expect them to conduct or repeat new experiments at common oxygen conditions given the findings that have been observed.

2. The manuscript lacks a clear and logical explanation for how H3K4me3 modifications directly regulate TSS switching under hypoxia. The authors should explore whether specific subregions of H3K4me3 peaks drive TSS relocation and how pre-existing chromatin states influence this process.

>>Ref#1's thoughts: This is certainly an interesting question and one that arises due to the findings in this paper. The discovery of TSS switching under hypoxia raises a number of interesting follow up questions but I do not think that identifying this mechanism is required for publication.

3. The C48 inhibitor has not been well-characterized, and we have not been able to find data in the literature that has tested the specificity of the compound. Therefore, a KDM5 knockout model is needed to confirm whether H3K4me3 redistribution is

necessary and sufficient for TSS switching. This would clarify whether the observed effects are due to H3K4me3 modulation itself or secondary effects from pharmacological inhibition. KDM5B another H3K4me3 specific demethylase should also be involved and discussed.

>>Ref#1's thoughts: I don't agree with the premise here. A knockout would create longer term phenotypic changes and would be difficult to interpret with respect to the effect of hypoxia. An inhibitor, which creates transient inhibition in a functional cell is much more equivalent to the hypoxic experiment. One could construct and inducible knockdown to address the transient changes but this also goes beyond the scope needed here. The authors should add some description and evidence of the KDM5 inhibitor from previous work.

4. Despite these insights, the study acknowledges challenges in directly correlating sequence-level TSS shifts with H3K4me3 modifications, as H3K4me3 redistributions typically span hundreds to thousands of bases, whereas TSS changes occur on a smaller scale. While preliminary analyses suggest correlations between H3K4me3 shifts and 5'UTR length changes, a simple upstream/downstream model does not fully capture the complexity of the mechanism. The authors should consider mapping changes to H3K4me3 levels at nucleosome resolution using Cut&Run or Cut&Tag, instead of ChIP-seq.

>>Ref#1's thoughts: This is a similar criticism to point 2. Speaks to a deeper understanding of mechanism. I don't think it's needed for publication but authors could address the need to understand this in the discussion.

Reviewer #2 (Remarks to the Author):

This is a revised version of a manuscript, which we found interesting, but contained several overstatements and included experimental data that did not fully support the conclusions. Now, the authors have incorporated new data to address our and other reviewers' concerns, correcting several of these deficiencies. The updated data better explain their results, and the revisions to the text more clearly highlight the significance of this work. However, issues remain, including questionable interpretations and logical gaps that need to be addressed in a further revision.

1. Inconsistencies in Hypoxia Conditions: The authors still present experiments in which they have used varying hypoxia conditions, even within the same cell line. This raises concerns about the interpretations of the data.

2. The manuscript lacks a clear and logical explanation for how H3K4me3 modifications directly regulate TSS switching under hypoxia. The authors should explore whether specific subregions of H3K4me3 peaks drive TSS relocation and how pre-existing chromatin states influence this process.

3. The C48 inhibitor has not been well-characterized, and we have not been able to find data in the literature that has tested the specificity of the compound. Therefore, a KDM5 knockout model is needed to confirm whether H3K4me3 redistribution is necessary and sufficient for TSS switching. This would clarify whether the observed effects are due to H3K4me3 modulation itself or secondary effects from pharmacological inhibition. KDM5B another H3K4me3 specific demethylase should also be involved and discussed.

4. Despite these insights, the study acknowledges challenges in directly correlating sequence-level TSS shifts with H3K4me3 modifications, as H3K4me3 redistributions typically span hundreds to thousands of bases, whereas TSS changes occur on a smaller scale. While preliminary analyses suggest correlations between H3K4me3 shifts and 5'UTR length changes, a simple upstream/downstream model does not fully capture the complexity of the mechanism. The authors should consider mapping changes to H3K4me3 levels at nucleosome resolution using Cut&Run or Cut&Tag, instead of ChIP-seq.

Reviewer #4 (Remarks to the Author) [Reviewer#3 was unable to review the revised manuscript]:

This is a very interesting and novel manuscript, and I focused my review on the comments raised by reviewer 3. The authors have addressed all the comments and added new bioinformatics analysis and functional experiments to corroborate their original results. I agree with the authors that some of the comments raised were either not feasible to address or outside the scope of this work. I found this manuscript significant and inspiring, providing future directions in the field of gene expression, stress response, and cancer biology. The authors showed a novel interplay between epigenetic mechanisms and translation control in regulating the homeostasis of cancer cells under hypoxia. We still do not understand the adaptive mechanisms of cancer cells to different environments, and this work provides new insights in this context that have the potential to open compelling new directions in research on gene expression, cancer development, and therapeutic opportunities. This manuscript is ready to be published.

Version 3:

Decision Letter:

Dear Dr Postovit,

I am pleased to inform you that your manuscript, "Epigenetic alterations facilitate transcriptional and translational programs in hypoxia", has now been accepted for publication in Nature Cell Biology. Congratulations!

Please note that *Nature Cell Biology* is a Transformative Journal (TJ). Authors may publish their research with us through the traditional subscription access route or make their paper immediately open access through payment of an article-processing charge (APC). Authors will not be required to make a final decision about access to their article until it has been accepted. [Find out more about Transformative Journals](https://www.springernature.com/gp/open-research/transformative-journals)

Authors may need to take specific actions to achieve compliance with funder and institutional open access mandates. If your research is supported by a funder that requires immediate open access (e.g. according to [Plan S principles](https://www.springernature.com/gp/open-science/plan-s-compliance) or the [NIH public access policy](https://www.springernature.com/gp/open-science/us-federal-agency-compliance)) then you should select the gold OA route, and we will direct you to the compliant route where possible. Because authors warrant under our subscription licensing terms that they haven't committed to licensing any version of their article under a licence inconsistent with the terms of our agreement – including the applicable embargo period – publication under the subscription model isn't suitable for authors whose funders require no embargo.

If you have not already done so, we strongly recommend that you upload the step-by-step protocols used in this manuscript to protocols.io (<https://protocols.io>), an open online resource that allows researchers to share their detailed experimental know-how. All uploaded protocols are made freely available and are assigned DOIs for ease of citation. Protocols and Nature Portfolio journal papers in which they are used can be linked to one another, and this link is clearly and prominently visible in the online versions of both. Authors who performed the specific experiments can act as primary authors for the Protocol as they will be best placed to share the methodology details, but the Corresponding Author of the present research paper should be included as one of the authors. By uploading your Protocols onto protocols.io, you are enabling researchers

to more readily reproduce or adapt the methodology you use, as well as increasing the visibility of your protocols and papers. You can also establish a dedicated workspace to collect your lab Protocols. Further information can be found at <https://www.protocols.io/help/publish-articles>.

Nature Cell Biology encourages authors presenting evidence for cell, biological, molecular, and genetic interactions to consider communicating these findings using Biofactoid (<https://biofactoid.org/>). This tool helps users share a searchable representation of interactions (e.g. binding, gene expression, post-translational modification) between genes, gene products, or chemicals. Information added to Biofactoid, with author attribution, is shared on social media and public databases, such as Pathway Commons, where it can be discovered and analyzed in the context of a large and growing corpus of knowledge.

With kind regards,

Sabrya Carim, PhD
(she/her/hers)
Senior Editor, Nature Cell Biology
Nature Portfolio

Springer Nature
The Campus, 4 Crinan Street, London N1 9XW, UK
sabrya.carim@springernature.com
<https://orcid.org/0000-0001-9485-1938>

** Visit the Springer Nature Editorial and Publishing website at http://editorial-jobs.springernature.com?utm_source=ejp_NCB_email&utm_medium=ejp_NCB_email&utm_campaign=ejp_NCB for more information about our career opportunities. If you have any questions please click [here](mailto:editorial.publishing.jobs@springernature.com).

Rebuttal:

Reviewer #1 (Remarks to the Author):

This is an exciting paper that has explored the effects of hypoxia on changes in transcription and translation through a series of sophisticated and well performed biological and analytical methods. The paper reveals at least three important and novel insights.

First, the authors have revealed a substantial change in TSS sites utilized by the transcriptional machinery during hypoxia. Through sequencing of UTR's under each condition, the data convincingly demonstrate changes in large numbers of genes in two different cell lines. This is a novel and interesting observation that is further explored in the paper. They go on to show these differential TSS result in substantial changes in UTR composition and are able to map these changes to some of the known mechanisms of mRNA translation regulation under hypoxia including mTOR regulation and ISR/UPR activation.

Second, the paper explores in detail, at the gene level, the impact of hypoxia on the efficiency of mRNA translation. While it has been known for many years that hypoxia inhibits mRNA translation through mTOR inhibition and UPR/ISR activation, there is relatively little experimental data that examines the direct impact of those effect on the translation of individual mRNAs. This is due in part to the technical complexity of both the biological and analytical methods. The authors should be congratulated on the quality of the work that was done and as a result the results are convincing. This unique gene level mRNA translation dataset has allowed the authors to examine the relationship between translation, TSS switching, and specific UTR elements. This analysis reveals that TSS switching drives a component of the change in translation and have linked this to the acquisition or loss of various UTR elements.

Third, and perhaps most importantly, the authors explore a potential mechanism that underlies the changes in TSS, UTR composition, and mRNA translation efficiency by mapping hypoxia induced changes in H3K4me3. Previous work has shown that hypoxia causes increases in the levels and genomic distribution of this mark through inhibition of KDM5, an enzyme required for its removal. Here, the authors demonstrate changes in H3K4me3 around TSS that correlate with changes in TSS in at least some of the hypoxia altered genes. Furthermore, they demonstrate concordant changes in a fraction (27%) of genes that show H3K4me3 changes under hypoxia in cells treated with the KDM5 inhibitor alone.

Finally, PDK1 is used as a key example that demonstrates that a key hypoxia adaptive gene is influenced by TSS start site changes. The data here are also convincing, including the additional methods used to validate the UTR changes and mRNA translation changes.

Overall, this manuscript provides a significant advance to the field, and connects and extend previous discoveries of gene regulation during hypoxia that had already identified contributions from regulation of the mRNA translation machinery and the epigenetic machinery. This is the first paper to show a substantial impact on TSS start site and its consequential impact on UTR composition and mRNA translation. It also highlights the importance of further exploring and understanding histone methylation changes that drive at least a component of the TSS changes.

We thank the reviewer for acknowledging the novelty and significance of our study.

Major comments:

1) The authors have selected two rather unrelated cell lines and have conducted their experiments using different hypoxia exposures. The basis for this is unclear and not explained in the manuscript. The ability to compare and contrast these cell lines would be much stronger if the experimental conditions used to generate hypoxia adaptation were the same. The rationale for this needs to be addressed.

It was not our intention to directly compare the cell lines, but rather to determine if TSS switching occurs in highly divergent cell types in response to hypoxia. As we and others have shown (PMID: 32427827; PMID: 22031289; PMID: 17685448), divergent cell types respond to hypoxia with differing kinetics, modifying gene expression and regulators of mRNA translation at different time points and oxygen concentrations. Accordingly, we chose time points and oxygen concentrations that induced LOX1 mRNA and HIF1 α protein levels, suppressed mTOR signalling, and induced ISR in both cell lines. We have added additional clarification in the text and have highlighted this in **Figure 2a-b**.

More importantly, the authors have not done a detailed comparison of the similarity or differences in these two lines. While the aggregate effects are similar (ie there are changes in TSS and changes in gene level mRNA translation) there is no comparison at the gene level on how these changes overlap. For example how many individual genes show common TSS shifts. When there is overlap, are the TSS changes the same or are they different? Similarly, how many individual genes between the two lines show common changes in mRNA translation efficiency? This more detailed analysis should be completed at the gene and UTR level.

This is important, because the impact of mTOR inhibition, ISR activation should impact individual genes to the same extent. However, it is less clear if one would expect shifts in TSS sites to be the same at the gene level when driven by changes in H3K4me3. Existing epigenetic marks are likely to vary across the two lines and thus the TSS switching may be highly cell type and cell line dependent. The data exist to explore and answer this to some degree.

Thank you for the excellent suggestions. Indeed, the rationale for selecting these cell lines was to determine whether hypoxia-induced TSS switching is maintained across divergent cell types. Our modeling of translation changes showed that both cell types modulate translation downstream of mTOR and ISR with expected directionality (**Figure 3b-c** and **Figure S4a-b**; further described below under point 3). We further address this below:

- 1) We analyzed how hypoxia affects the translation of mRNAs in T47D breast cancer cells and H9 hESCs and then compared the modes of regulation for individual genes (translation, offsetting, and mRNA abundance) between the cell types (**Reviewer Figure 1**). Using a threshold-independent approach, we found substantial similarities in regulation between T47D and H9 cells for genes altered at the levels of mRNA abundance (**Reviewer Figure 2a-b**), and translation (**Reviewer Figure 2c-d**). In contrast, marginal overlap in translational offsetting was observed in H9 but not T47D cells (**Reviewer Figure 2e-f**). Furthermore, while gene ontologies were shared between hypoxia-regulated genes in T47D and H9 cells (**Fig S2a-b**), some ontologies and regulatory modes were distinct between cell types (**Reviewer Figure 3a-d**). For example, genes involved in oxidative phosphorylation were translationally suppressed in T47D cells (**Reviewer Figure 3b**), while in the more glycolytic H9 cells (PMID: 24651542), these genes were induced at the level of mRNA abundance but translationally offset (**Reviewer Figure 3c**). Due to spatial constraints, these results were not included in the revised manuscript. However, if the

reviewer feels this is of sufficient interest we would be happy to add this additional comparison to the manuscript.

- 2) We have compared the genes that undergo hypoxia-induced TSS switching in T47D versus H9 cells and examined whether these changes result in similar alterations in 5'UTR patterns. Despite the highly divergent cell types used, we found that ~11% of the transcripts that underwent TSS switching were shared between the cell lines and that ~37% of these switching events resulted in very similar changes in 5'UTR isoforms (**Figure 1I**). Importantly, genes that shared TSS switching patterns were enriched for functions related to core hypoxia-related processes including glucose metabolism, and chromatin remodelling (Supplemental **Figure S1j**). We also carried out a pilot study using nanoCAGE sequencing of total and polysome-associated mRNA. This approach captures changes in TSS usage as well as isoform-level changes in polysome association. We found that in addition to TSS switching, there is a significant increase in the number of genes with 5'UTR isoforms that differ in their polysome-association (after adjusting for total mRNA levels) under hypoxia in T47D cells (**Reviewer Figure 4a**). Although this comparison between UTR-level translation across cell lines is possible, we feel this area requires further methodological refinement that is beyond the scope of the current study where we focused on switching events that were most likely to lead to changes in the proteome. However, this is an area we are actively pursuing in our ongoing work.
- 3) When modeling hypoxia-associated changes in mRNA translation in T47D breast cancer cells and H9 hESCs, we confirmed that mTOR suppression and ISR induction underlie changes in hypoxia-associated translation in both cell types (**Figure 3b-c** and **Figure S4a-b**). For example, for mRNAs harbouring TOP motifs whose translation is highly mTOR-dependent as an example, we observed significant translational offsetting for these genes in both T47D and H9 cells, whereby increased mRNA levels are counteracted by decreased translation efficiency (**Figure S2c-d**). However, in H9 cells we found that TSS switching impacting TOP motifs could independently explain changes in translation efficiency, and was significantly associated with translational suppression (**Figure S4c and e**). This suggests that hypoxia-induced TSS switching impacting different genes between divergent cell types can lead to differential translational regulation, yet rely on common pathway alterations.

2) The manuscript would be significantly strengthened by exploring in more detail the relationship between the H3K4me3 changes and the TSS changes during hypoxia at the individual gene/UTR level. The analyses are mostly done in aggregate and show significant changes in both and to some extent the overlap between upstream and downstream changes. However, the more interesting question is to examine how many genes show a defined H3K4me3 and TSS change, ie where a TSS switch can be directly attributed/correlated to a corresponding H3K4me3 change around that TSS site. What fraction of the TSS changes can this attribution be made, and for those genes what is the impact on mRNA translation relative to those where it is not?

As comments #2 and 3 both address mechanistic questions relating to the connection between changes in H3K4me3 and TSS usage, we address them together below.

3) A similar opportunity to explore specific changes in H3K4me3 and TSS change exist in the KDM5 experiments. The data comparing KDM5 inhibition to hypoxia resulted in an overlap of genes showing TSS changes of 27%. This is considered significant compared to a Monte Carlo simulation by random of 20%. However, what is key to understand in those 27% of genes is if the TSS switching and the H3K4me3 changes at the gene/sequence level are similar in genomic

position. At a minimum this should look at upstream versus downstream, but it would be far more convincing if the actual TSS start sites and H3K4me3 changes were compared at the gene level here to see if the H3K4me3 change was driving this at the sequence context.

We thank the reviewer for these comments. We have added several new experiments and analyses to better understand how TSS switching under hypoxia may arise. Specifically, we examined H3K4me3 distribution, nucleosome positioning and occupancy, and TSS selection by exposing cells to hypoxia or treating them with a KDM5a inhibitor (C48) and then conducting ChIP-seq (for H3K4me3), ATAC-seq, and nanoCAGE.

KDM5a inhibition by C48 resulted in comparable changes in H3K4me3 distribution to what was observed under hypoxia around the TSSs of ~59% of protein-coding genes (**Figure 4i**). Furthermore, C48 treatment resulted in the same pattern of TSS switching as observed under hypoxia in ~28% of cases (see revised text). Importantly, for the majority of genes this occurred without a change in overall mRNA abundance (**Figure 5f**), suggesting that the H3K4me3 mark plays a role in TSS selection, independent of changes in transcriptional output. To further confirm the link between modulation of H3K4me3 distributions and TSS selection, we treated T47D cells grown in hypoxia with either DMSO, or OICR-9429, which inhibits the interaction between WDR5 and MLL in MLL-containing COMPASS H3K4 methylases (**Figure 6a-b**). NanoCAGE sequencing showed that OICR-9429 prevents close to 30% of hypoxia-induced TSS switching (**Figure 6**). Therefore, we confirmed the role for H3K4me3 in TSS selection under hypoxia by using two independent pharmacological modulators of H3K4me3 (KDM5a and MLL methyltransferase inhibitors).

Furthermore, under hypoxia, we examined the chromatin state for genes that undergo TSS switching relative to those that do not. This revealed what appears to be priming (i.e., differences in nucleosome positioning/occupancy under normoxia) as well as distinct alterations affecting both upstream and downstream nucleosomes under hypoxia (**Figure 4e**). This is exemplified by PDK1 (**Figure 7k**) where a broadening of the H3K4me3 mark associates with an upstream shift in the dyad position of the +1 nucleosome, increased occupancy at the upstream nucleosome, and expression of shorter 5'UTR isoforms under hypoxia. We found that these changes were recapitulated with remarkable similarity with C48 treatment in the absence of hypoxia (**Figure 7k**). This suggests that the modulation of H3K4me3 resulted in changes in nucleosome conformation around the PDK1 TSS that created a chromatin state permissive for transcription of the shorter 5'UTR isoforms, in a manner that does not require HIF1. However, transcriptional activation by HIF1 may further contribute to the TSS switching resulting in decreased expression of longer 5'UTR isoforms that we observed under hypoxia, but not with C48 treatment. Together these findings demonstrate concomitant changes in H3K4me3, nucleosome conformation (position and occupancy), and TSS switching that occur both with hypoxia and inhibition of KDM5a alone. Hence, we posit that altered H3K4me3 deposition drives nucleosome repositioning and TSS switching for a subset of genes.

As changes in H3K4me3 distributions occur on the scale of hundreds to thousands of bases (see **Figure 4c and Figure 7k**), and most TSS switching we report occurs on a smaller scale (see examples in **Figure 1e-g, Figure 5i-j, Figure 6h and j, and Figure 7a**), it is challenging to correlate the sequence context of the acquired TSS changes to corresponding changes in H3K4me3. Accordingly, while our preliminary analyses have revealed correlations between upstream and downstream shifts in H3K4me3 and 5'UTR lengthening and shortening, we feel this simplified model is not adequate to capture the complexity of the mechanism activated under hypoxia. Furthermore, although our C48 experiments have demonstrated that modulating H3K4me3 is sufficient to induce TSS switching, under hypoxia there were also many examples

where H3K4me3 was altered in the absence of TSS switching, indicating there are both H3K4me3-dependent and -independent subsets of switching events. Due to the differential patterns of TSS switching we observed between cell types, and the “priming” effect we observed with nucleosome conformations (**Figure 4e**), we posit that in addition to the potential effect of the above-mentioned sequence context determinants, pre-existing epigenetic states likely also play an important role in determining both the pattern and extent of TSS switching upon H3K4me3 alterations. Substantial additional data is required to understand the epigenetic states that may predispose genes to TSS switching, and the complex subsets of these states that result in different patterns of TSS switching. This is a primary topic of our ongoing work. However, we feel that it falls beyond the scope of the current manuscript.

4) Finally, the authors should consider an effort to explore the biological relevance of these unique insights. Biological impact has not been investigated. The manuscript would have a much larger impact on the scientific if there was a demonstration that the TSS switching, driven by H3K4me3 changes leads to functionally relevant phenotypes under hypoxia. There are some great examples of how it is influencing the translation of key genes like PDK1 and PLOD2, but it is unclear to what magnitude this is actually affecting protein levels, enzyme activity, and most importantly cellular phenotype.

Thank you for the suggestion. We explored the biological relevance of epigenetically-driven TSS switching using several approaches:

- 1) We assessed if the switch in PDK1 5'UTR isoform abundance impacts glucose carbon flux during the transition from oxidative phosphorylation to glycolytic metabolism that occurs under hypoxia (PMID: 35642641). To this end, we expressed 134, 57, and 36 nt PDK1 5'UTR isoforms followed by CRISPR-mediated knockout of the endogenous PDK1 gene in T47D cells (**Figure 7h**). Notably, the shorter 5'UTR variant was associated with higher PDK1 protein abundance independently of mRNA level, consistent with more efficient translation (**Figure 7i**). Cells expressing PDK1 5'UTR variants were subjected to [¹³C] pyruvate labelling followed by stable isotope tracing analysis by GC-MS after 24 hours in hypoxia or normoxia (0.5% or 20% O₂). This revealed that T47D cells expressing the shorter, more efficiently translated, 36 nucleotide 5'UTR isoform of PDK1 exhibit increased conversion of pyruvate to lactate under hypoxia as compared to the less efficiently translated 134 nucleotide isoform (**Figure 7j**). Furthermore, we found that a number of other enzymes involved in glycolysis upstream of PDK1 also underwent TSS switching in T47D cells (**Figure S8b**), suggesting complex coordination between 5'UTR isoform expression, translation efficiency, and metabolic adaptations.
- 2) We determined whether modifying H3K4me3 patterns in the absence of hypoxia can remodel the proteome. Remarkably, in **Figure 5k-m**, we show that C48 treatment leads to proteome changes independent of alterations in total mRNA levels, and without changes in mTOR or ISR signaling (**Figure 5a**). Furthermore, these mRNA-independent protein level-changes associate with TSS switching. This is exemplified by PDK1, which gains shorter 5'UTR isoforms and exhibits increased protein level independent of corresponding mRNA expression (**Figure S9o**). Globally, these proteome adaptations impacted functions related to hypoxia response, lipid metabolism, protein modifications, and extracellular vesicles, among others (**Figure S7f**). These findings link TSS switching downstream of KDM5 inhibition to proteome adaptations arising independently of mRNA-level changes, mTOR or ISR.
- 3) We hypothesized that if TSS switching is important for cellular adaptation to hypoxia, blocking the switching may affect key adaptive phenotypes including cell proliferation and/or survival under hypoxia. To assess this, we treated T47D cells grown in hypoxia

with either DMSO, or OICR-9429 (an inhibitor of the interaction between WDR5 and MLL in COMPASS methyltransferases that deposit a subset of H3K4me3 marks). NanoCAGE sequencing revealed that OICR-9429 treatment blocked around 30% of hypoxia-induced TSS switching, and significantly reduced cell proliferation under hypoxia relative to the same treatment under normoxia (**Figure 6**).

Together, these data support that hypoxia-induced TSS switching is an important mechanism that impacts key cellular processes and phenotypes during adaptation to hypoxia. These additions are described in the revised text, and presented in **Figures 5, 6, 7h-j, and Figures S7-9**.

In addition, with regard to PLOD2, we generated a new nanoCAGE dataset for the C48 treatment, such that our proteomics, RNAseq, ChIPseq, and ATACseq could all be derived from the same samples. In this new dataset, PLOD2 underwent the same significant TSS switching we reported in the previous version of our manuscript (see **Supplemental File S1**). However, in light of useful feedback from Reviewers, in the revised version we chose to display examples where the TSS switching was more visually obvious in an effort to present our findings more clearly (**Figure 5i-j**).

Minor comments:

Introduction

1) Page 2 Line 3 – this is a rather specific and unusual definition of plasticity

Thank you, we have revised the text for clarity.

2) Page 2 Line 10 – there are earlier publications on the impact of hypoxia on the epigenome - PMID: 27800026

Apologies for the oversight, we have added the appropriate citation.

3) Page 2 Line 25 – the inhibition of translation during hypoxia is also due in large part to hypoxia induced ER stress (independent of energy demands)

Thank you, we have clarified this in the text.

Results

4) Page 5 Line 7 – Figure 2c and its description should be explained in more detail. There is an important normalization impact on the way that data is scaled given that hypoxia has caused a very massive overall suppression of protein synthesis of up to 80%. In this figure that impact is essentially normalized out, and thus it appears from figure 2c that there are similar overall changes in up and down regulated translation. The absolute amount of translation (ie that producing protein) will be shifted downward by the overall average inhibition.

Thank you for bringing this point to our attention. We have tried to clarify in the text that this analysis is carried out in a manner that accounts for differences in global protein synthesis between different conditions to identify transcript-selective alterations in gene expression. To this end, we sequenced the same amount of RNA originating from fractions containing more than three ribosomes across all conditions and identified different modes of gene expression alterations using anota2seq (i.e., abundance, translation and offsetting). Importantly, in the

absence of selective alterations in gene expression, this approach would not identify changes due to a global decrease. Therefore, our approach allows capturing relative (transcript-selective) differences in translation efficiency and other modes of gene expression regulation (e.g., offsetting), which underlie the newly synthesized proteome under hypoxia.

Discussion

5) Page 12 Lines 36-37. It is stated: "This altered pool of 5'UTRs works in concert with reprogramming of the translational apparatus to mediate a survival-enhancing adaptive translational program." This has not been demonstrated in this manuscript. No exploration of the contribution of this unique mode of gene regulation on cellular phenotypes, including cell survival, have been evaluated. As it stands now, the discussion should focus more on the unique discoveries of gene regulation during hypoxia rather than on the impact this has on biological properties.

Thank you for the constructive suggestions, we have revised the discussion accordingly, also reflecting the addition of our new data.

Reviewer #2 (Remarks to the Author):

Watt et al., address the potential function of hypoxia in regulating transcription and post-transcriptional programs and provide results suggesting that H3K4me3 is essential for remodeling of 5' UTR selection in hypoxia. Moreover, they suggest that this process is independent of HIF1 dependent transcriptional mechanisms and requires mTOR signaling.

We demonstrate that TSS switching can occur independently of HIF1-dependent transcription as C48 treatment, which modulates H3K4me3 without stabilizing HIF1, was sufficient to induce extensive TSS switching that in large part recapitulated that observed under hypoxia. However, we wish to clarify that we do not show data suggesting that mTOR signaling is involved directly in mediating TSS switching under hypoxia. Rather, we show that C48, which does not inhibit mTOR and does not activate the ISR, induces TSS switching that associates with changes in protein levels that cannot be explained by altered mRNA levels (**Figure 5a**). Hence, we suggest that TSS switching-induced alterations in 5'UTR features can regulate the translation of a subset of transcripts in the absence of mTOR-regulated remodelling of the translational machinery. We have clarified these points in the revised version of the manuscript.

Although the manuscript addresses an interesting topic - attempting to reveal how hypoxia links epigenetics to TSS selection, some of the data are ambiguous and there are multiple overstatements not well supported by experiments. In addition, the biological relevance of the reported observations is unclear, and mechanistic insights into how 5'UTR selection by H3K4me3 is regulated are lacking. Therefore, we are unable to recommend publication of the manuscript.

As described above, we have performed a multitude of experiments to understand the biological relevance of the hypoxia-associated alterations in TSS selection. We now show that the hypoxia-induced shorter 5'UTR variant of PDK1, but not longer variants, leads to increased PDK1 protein level (independent of mRNA level) and promotes reduction of pyruvate to lactate (**Figure 7h-j**); that KDM5 inhibition using C48 results in proteome alterations, that are independent of mRNA level changes but associated with changes in TSS selection, targeting key adaptive functions as determined by gene ontology analysis (**Figure 5k-m, Figure S7f**); and that suppression of H3K4me3 using OICR-9429 under hypoxia, which blocks ~30% of hypoxia-induced TSS switching events, leads to reduced proliferation (**Figure 6k**). We have thereby linked TSS

alterations to cellular phenotypes both at single gene (PDK1) as well as genome-wide levels. To better understand how TSS switching arises, we studied nucleosome positioning and occupancy for genes that undergo TSS switching under hypoxia relative to those that do not. This revealed both priming (i.e. differences in nucleosome positioning/occupancy under normoxia) as well as distinct alterations affecting both upstream and downstream nucleosomes under hypoxia (**Figure 4e**). This is exemplified by PDK1 (**Figure 7k**) where a broadening of the H3K4me3 mark associates with altered nucleosome positioning and expression of shorter 5'UTR isoforms under hypoxia, which is recapitulated with C48 treatment in the absence of hypoxia. These data thereby indicate a link between H3K4me3, nucleosome positioning and/or occupancy and TSS changes. This link was further confirmed by reducing the deposition of H3K4me3 under hypoxia with OICR-9429, which prevented a subset of TSS switching events (**Figure 6**).

1. A major question is regarding the biological relevance of the observations. How important are the 5'-UTR switches? Most often, the same amount of protein is produced (they are not measured in the manuscript, however mRNAs associated with ribosomes are). The same protein is predicted to be produced as well because the difference in the 5'-UTR most often does not contain an alternative start codon. Therefore, although the observation of these different 5'-UTR is interesting, the biological impact of the small changes in the 5'UTRs remains unclear.

Thank you for the suggestion. Please see our response to comment #4 from reviewer #1 above, wherein we have addressed the functional relevance of the observed TSS switching using three complementary experimental approaches.

Regarding the link between TSS-switching and changes in protein levels, we have now added proteomics for C48-treated T47D cells under normoxia, demonstrating that TSS switching is significantly associated with changes in protein levels occurring independently of alterations in total mRNA levels (**Figure 5k-m**). However, we concur that TSS switching may also occur without accompanying alterations in corresponding proteins. This mode of regulation is equivalent to translational offsetting, whereby protein levels are maintained despite changes in 5'UTR isoform expression (PMID: 31556460). This suggests that TSS switching-mediated translational offsetting may serve as a stress-adaptive response (PMID: 34599983). Indeed, in our pilot study using nanoCAGE sequencing of both total and polysome-associated mRNA we identified instances where the TSS switching under hypoxia is offset by altered translation efficiency (**Reviewer Figure 4b-c**). Although we have edited the text of the manuscript to clarify this point, we have not included these figures, as we feel isoform-level offsetting and the temporal dynamics at which this occurs require further development that is beyond the scope of the current study, which focuses on TSS switching events most likely to alter protein levels. However, this is an area we are actively pursuing in future studies.

2. The authors try to link changes in TSS selection induced by hypoxia with H3K4me3 and KDM5 activity. Although the observations are interesting, the authors fail to provide mechanistic insights into why and how a general increase in H3K4me3 levels should lead to changes in TSS selection.

- H3K4me3 is abundant at expressed genes. Why would a change in H3K4me3 levels change the TSS? The authors do not address that or provide results suggesting a potential mechanism.
- KDM5 demethylation is not known to regulate nucleosome-specific localization of H3K4me3 – only the overall levels of H3K4me3. Instead of ChIP-seq, the authors should perform experiments that can give single nucleosome resolution. Would that show nucleosome specific changes in H3K4me3 levels and have the nucleosomes moved relative to the DNA.

Thank you for the comments. Please see our response to Reviewer #1's comments #2-3. Briefly, we found the inhibition of KDM5a by C48 resulted in comparable changes in H3K4me3 distribution

to what was observed under hypoxia around the TSSs of a large proportion (~59%) of protein-coding genes (**Figure 4i**). Importantly, for the majority of the genes TSS switching was not accompanied by changes in overall mRNA abundance (**Figure 5f**), suggesting that the H3K4me3 mark plays a role in TSS selection, independent of changes in transcriptional output. To further confirm the link between modulation of H3K4me3 distributions and TSS selection, we treated T47D cells grown in hypoxia with either DMSO, or OICR-9429 which resulted in reduced H3K4me3 accumulation (**Figure 6a-b**). NanoCAGE sequencing allowed us to determine that OICR-9429 treatment prevented close to 30% of hypoxia-induced TSS switching (**Figure 6**). Therefore, we have shown that modulation of H3K4me3 through two independent mechanisms results in extensive TSS switching that could both recapitulate and block changes in TSS usage observed under hypoxia, confirming a role for H3K4me3 in TSS selection.

Examination of the chromatin state for genes that undergo TSS switching revealed both priming (i.e., differences in nucleosome positioning/occupancy under normoxia) and distinct alterations affecting both upstream and downstream nucleosomes under hypoxia (**Figure 4e**). This is exemplified by PDK1 (**Figure 7k**) where a broadening of the H3K4me3 mark associates with an upstream shift in the dyad position of the +1 nucleosome, increased occupancy at the upstream nucleosome, and expression of shorter 5'UTR isoforms under hypoxia. We found that these changes were recapitulated with remarkable similarity with C48 treatment in the absence of hypoxia (**Figure 7k**). This suggests that H3K4me3 accumulation alters nucleosome conformation around the PDK1 TSS thus creating a chromatin state permissive for transcription of the shorter 5'UTR isoforms, in a HIF1-independent manner. Together these findings demonstrate concomitant changes in H3K4me3, nucleosome conformation (position and occupancy), and TSS switching that occur both with hypoxia and inhibition of KDM5a alone. Hence, we propose a mechanism whereby altered H3K4me3 deposition is sufficient to drive nucleosome repositioning and TSS switching for a subset of genes.

Importantly, although both hypoxia and C48 treatment lead to a global increase in H3K4me3, our analysis shows that these treatments also drive alterations in the genomic position of H3K4me3 distributions around TSSs. This is highlighted with examples at the gene-level (**Figure 4c**), but also in our aggregate analyses (**Figure 4d and h**) that showed many significant directional shifts in the positions of the H3K4me3 signal around TSSs. For this reason, we would not refer to the changes in H3K4me3 as a “general increase” but rather a context-dependent change in distribution (often accompanied by an overall increase in magnitude). As we observed these directional shifts both under hypoxia, and with C48 treatment, this suggests that KDM5a demethylation does not impact all nucleosomes equally. However, the determinants of this selection appear to be highly complex and are a topic of our ongoing and future work.

Finally, our preliminary analyses have revealed correlations between upstream and downstream shifts in H3K4me3 and 5'UTR lengthening and shortening. We however feel that this simplified model is not adequate to capture the complexity of the mechanism. Furthermore, although C48 experiments demonstrated that modulating H3K4me3 is sufficient to induce TSS switching, under hypoxia there were also many examples where H3K4me3 was altered in the absence of TSS switching. This indicates the existence of both H3K4me3-dependent and -independent subsets of switching events. Due to the differential patterns of TSS switching we observed between cell types, and the “priming” effect we observed with nucleosome conformations (**Figure 4e**), we posit that in addition to the potential effect of the above-mentioned sequence context determinants, pre-existing epigenetic states likely also play an important role in determining both the pattern and extent of TSS switching upon H3K4me3 alterations. Substantial additional data is required to understand the epigenetic states that may predispose genes to TSS switching, and the

combinations of states that result in different patterns of TSS switching. We feel that this is beyond the scope of the current manuscript, which is already rather complicated.

3. The experimental approaches and analyses do not appear to be consistent throughout the manuscript.

- NanoCAGE-seq, only two samples are shown for T47D-normoxia and three samples for hypoxia condition in Fig1b, however, five samples were found for H9 in Fig. S1b and four in Fig. S1a.

One of the T47D nanoCAGE replicates for normoxia failed sequencing QC, and unfortunately had to be removed from the analysis. As the H9 cells are significantly more challenging to culture, we initially performed five replicates. One of these replicates also failed QC and was removed from the analysis. We thank you for identifying our error in **Figure S1b**. We have updated the figure to reflect that replicate five was removed leaving only the four replicates for the H9 cells originally stated in the manuscript.

- The authors switch between 0.5% for hypoxia and 1%. Why?

Please see our response to reviewer #1's comment #1.

- There is no big difference in nanoCAGE-seq between normoxia and hypoxia conditions (Fig. 1g-j), which may be the reason the authors use $FDR < 0.15$ for differential analysis. An FDR of this value may give a lot of false positives. The authors should test their results using a more normal $FDR < 0.05$.

We apologize that we have not been clear enough in demonstrating the extent of the differences in TSS usage between hypoxia and normoxia. We have added panel **f** of **Figure S1**, to show the p-value densities in addition to FDRs, which may make the large enrichment of significant TSS switching events more visually obvious. The FDR threshold of 0.15 was not selected due to a lack of significant differences, as at a threshold of 0.05 there were 1,276, and 2,216 transcripts with significant TSS switching in T47D and H9 cells, respectively (**Supplemental File S1**). In developing these methods for identifying differential TSS usage, we have applied a very similar statistical approach as to what is used in our published aNota2seq algorithm (PMID: 30926999). There, using controlled simulations we showed that an FDR threshold of 0.15 was very effective for capturing many more true positives at the cost of the expected false positives, as compared to a threshold of 0.05. Therefore, we have used the same approach here. We also would like to emphasize the importance of the context in which these thresholds are applied. For interrogating patterns and integrating data of different modalities and dynamic ranges (nanoCAGE, RNAseq, ChIPseq, proteomics), false negatives are also an issue. A robust analysis requires sufficient true positives, and the difference between a given gene having an 85% or 95% chance of being a true positive is weighed against the cost of it being a false negative. So, for aggregate analyses to identify broader patterns, we have shown that using these statistical methods it is appropriate to use the 0.15 threshold in terms of balancing true positives and false negatives (PMID: 30926999). However, in the context of examining individual transcripts (such as the TSS switching for PDK1, $FDR = 3.51E-05$ in T47D and $3.22E-08$ in H9), the use of a more stringent FDR threshold is appropriate, as we have done in this study.

- Why is 48h hypoxia used for T47D and 24h for H9?

Please again see our response to reviewer #1's comment #1.

4. The authors claim there is a “high” degree of coordination between some pathways and 5'-UTR translation in hypoxia (Fig. 2 and Fig. S2,3), however, these claims are not supported by any experiments. Moreover, the differential co-regulated genes by 5 UTR and translation under hypoxia are few and more like a random event.

We apologise for the confusion and feel that it was due to the way in which we presented the data. We have substantially reworked the presentation of the data in the revised version of the manuscript. To establish context, all network graphs shown (e.g., **Figure 3**) illustrate a modelling of gene expression changes occurring in hypoxia (quantified in the current study) as depending on specific pathways (by using mRNAs previously established as sensitive to these pathways [e.g. mTOR and ISR]; full gene lists are provided in **Supplemental File S4**, and their sources described in the methods section) and/or 5'UTR features. With this in mind, we have below attempted to address the specific comments:

“The authors claim there is a “high” degree of coordination between some pathways and 5'-UTR translation in hypoxia (Fig. 2 and Fig. S2,3), however, these claims are not supported by any experiments. Moreover, the differential co-regulated genes by 5 UTR and translation under hypoxia are few and more like a random event.” To further demonstrate that the changes in translation efficiency occurring downstream of hypoxia-induced TSS switching are not random events, we have added **Figure 2g-j**, which indicates that TSS switching is significantly associated with alterations in translation efficiency in both cell types. The results presented in **Figure 3**, and **Figures S3-5** were obtained through nanoCAGE experiments that precisely determine the 5'UTR, and transcriptome-wide changes in translation efficiency. Hypoxia-associated changes in translation efficiency were then modelled to understand what could be explained by *i*) alterations in the translome mediated by pathways and factors known to impact translation efficiency (e.g., mTOR and ISR that are perturbed under hypoxia) (**Supplemental File S4**), and *ii*) 5'UTR features identified and/or quantified in the present study. These models largely confirmed what is known about translational regulation under hypoxia, identifying a prominent role for mTOR suppression and activation of the ISR (PMID: 15545625, PMID: 12370288). Importantly, our approach also identified additional pathways, factors and 5'TUR features that have not previously been implicated in translational reprogramming under hypoxia.

We show that TSS switching influences the translome in a manner that is both dependent and independent of the known translation-regulating pathways and factors we examined (**Figure 3b-c**, **Figure S4a-b**). For instance, TSS switching impacts 5'UTR features that are known to mediate mTOR- and ISR-dependent translational regulation including TOP motifs and uORFs. For example, in **Figure 3b**, we show that 5'UTR features including length, GC content, fold energy, presence of uORFs, and AAGAAA motifs that are all correlated with mTOR-sensitive translation. In the case of length and GC content, these findings independently confirm what is already known regarding mTOR-sensitive translation initiation (PMID: 26984228). In these omnibus models, the percentages (now shown in **Supplemental File S4**, to increase the visual clarity of the figures) represent the contribution of a given variable to the measured changes in translation efficiency. To clarify this point, we have provided an example of a model including only mTOR-activated translation and the 5'UTR features that are substantially correlated. In a univariate analysis (**Reviewer Figure 5a**), the mTOR variable alone explained 18.4% of the variance in translation efficiency under hypoxia. After adding the 5'UTR features to the model, the independent contribution of mTOR (adjusted for the contribution of the included 5'UTR features) decreased to ~12%, meaning that ~33% of hypoxia-associated changes in translation of mRNAs described as showing mTOR-sensitive translation could be explained by these 5'UTR features (notably, these estimates are limited by measurement error) (**Reviewer Figure 5b-c**). Therefore, when

interpreting the network plots, connections between nodes indicate substantial positive or negative correlations, in this case indicating a relationship, or what we previously termed “coordination”, between mTOR-dependent translation initiation and specific 5’UTR features (**Reviewer Figure 5d**). Notably, TSS switching impacted all of the above-mentioned 5’UTR features to varying degrees (**Figure S1h-i, Figure 3f-j, Figure S4e-g**), suggesting that TSS switching may fine-tune mTOR-regulated translation.

To further illustrate the impact of TSS switching on the 5’UTR elements that we identified as being associated with hypoxia induced changes in translation, we have added **Figure 3j and Figure S4g**. For example, in both cell types hundreds of transcripts gain or lose one or more uORFs in > 10% of the expressed pool of isoforms as a result of hypoxia-induced TSS switching. To this end, TSS switching affects ISR-directed translation that critically depends on the presence of uORFs. We have also extensively revised the text to clarify our findings.

Finally, we have also added experiments demonstrating that TSS switching downstream of C48 treatment modulating H3K4me3 is associated with changes in protein levels, independent of altered total mRNA levels (**Figure 5k-m**). Together, we feel these findings indicate that the changes in translation efficiency that result from altered 5’UTRs are not random events but rather play a substantial role in shaping the proteome as cells respond to stress.

Reviewer #3 (Remarks to the Author):

All cells must be able to quickly respond to their environment by tuning gene expression. Cancer cells are exposed in the tumor microenvironment to hypoxia, and the ability of cancer cells to adapt through transcription and translation-regulated gene programs is correlated with higher cancer stem-cell phenotypes. Under hypoxia, cancer cells turn off general protein synthesis by blocking eIF4E and turning off eIF2a. Therefore, how stress response proteins are unregulated is a critical question. Here, authors use sequencing and bioinformatics to identify changes in transcription start site usage during hypoxia. They find that certain RNA features such as 5’ UTR length or RNA motifs correlate with translation efficiency during hypoxia. They further find that there are epigenome changes around the hypoxia-regulated transcription start sites.

Authors note that “remarkably TSS switching is a previously unappreciated mechanism facilitating coordination between mRNA sequence features and translation initiation.” Given that there is little functional work in the manuscript, this is supposed to be the major novel finding of the paper. However, this statement is untrue. Indeed work from the Brar and Unal labs have previously revealed extensive transcription start site changes that lead to translational control (29474919,33826921,36622328,30016623), including during meiosis and ER stress. Therefore, excitement for the work is limited without substantial addition of functional studies including (1) proof that TSS switching is essential for cell survival during hypoxia, and (2) mechanism of action by which TSS switching (and in particular the new RNA motifs and RNA-binding proteins identified by bioinformatics analysis) leads to enhanced translation during shutoff.

We apologize for failing to cite these previous findings and have corrected this in the revised version of the manuscript. Importantly, we believe that the long undecoded transcript isoforms (LUTIs) described in the above-mentioned studies carried out in yeast are not comparable to the extent and diversity of TSS switching and associated regulatory mechanisms that we report here. Namely, the reported LUTIs are based on a uniform mechanism whereby the use of an alternative TSS leads to the inclusion or exclusion of a uORF, thereby altering translation efficiency. The TSS switching we report here is considerably more extensive, and we found changes to a plethora of 5’UTR features across thousands of transcripts that would impact rates of translation initiation

through diverse mechanisms. Moreover, the induction of LUTIs is mediated by the induction of specific transcription factors, while under hypoxia we observed the changes in TSS selection can be driven by changes in epigenetic marks. We found that about 14% and 10% of hypoxia-induced TSS switching events impacted the presence of uORFs in T47D and H9 cells, respectively (resulting in a gain or loss in at least 10% of 5'UTR isoforms for that transcript). It is indeed possible that TSS switching may induce a "LUTIs-like" effect in mammalian cells for a subset of genes under hypoxia, but the changes we report here extend well beyond this specific mechanism. Based on this, although a small subset of TSS switching in mammalian cells exposed to hypoxia may impact LUTIs identified in yeast during meiosis or ER stress, we believe that our study significantly improves the understanding of the stress-induced regulation of gene expression in highly disease-relevant mammalian systems.

Specific comments:

- It is unclear why the two different cell lines were chosen. The cell lines display different characteristics in terms of TSS usage (e.g. H9 cells do not display a specific preference for short or long UTRs) but the authors do not explain how these changes lead to differences in cell-type specific physiology during hypoxia.

Please see our response to reviewer #1's comment #1 above.

In short, the rationale for selecting these cell lines was to determine whether hypoxia-induced TSS switching is maintained across highly divergent cell types, although different genes may be affected based on cell type and fate. We posit that the differences in hypoxia-induced TSS switching observed between the cell lines (note that H9 cells had a strong preference for expression of longer 5'UTR isoforms) may be due to the divergence of existing epigenetic marks between cell types. In light of the excellent suggestions, we added a comparison of hypoxia-induced TSS switching between cell types (**Figure 1I**). However, as described above, the intention of our study was not to compare the different cell types, but rather to investigate cell-specific and common mechanisms of TSS switching between these divergent cell lines. We also appended a more extensive comparison of the cell lines (**Reviewer Figures 1-3**). We are happy to add this to the manuscript if it is of interest to the reviewers, despite not being the original intention of the study.

- Authors use bioinformatics to identify different RNA motifs but provide no evidence that motifs are regulatory, for example authors should use reporter assays to show resistance to hypoxia-induced translation shutoff, and additionally show if these motifs are coordinately enriched in specific gene ontologies. Do the motifs contribute to changes in what RNA-binding proteins are found loaded onto the RNA during translation? Do the motifs contribute to structure? In addition, the authors should examine RNA structure, given substantial data implicating RNA structure to translational control.

Thank you for the suggestions. We have added **Supplementary File S5**, which includes gene ontology enrichments for the identified motifs, and information regarding evidence-based predicted RNA-binding proteins in cases where such information is available (**Supplemental File S4**). We have also performed a pilot study where we used nanoCAGE sequencing of total and polysome-associated mRNA isolated from T47D cells under both hypoxia and normoxia that showed that the presence of several of the motifs we identified in our analysis (**Figure 3b-c, and Figure S4a-b**) appear to significantly impact translation under hypoxia. For example, the 5'UTR of CBX5 harbours the UCCUCU motif which was associated with translational activation under hypoxia in T47D cells (**Reviewer Figure 6a**). Upon dividing 5'UTR isoforms into two groups (those

having the motif, and those without it), we found that the isoforms that included the motifs were more enriched in polysomes under hypoxia as compared to normoxia (**Reviewer Figure 6b**). Notably, the UCCUCU was gained and lost due to TSS switching under hypoxia in > 90 transcripts (**Figure 3j**). This was also the case for the CGGACGU motif, where 5'UTR isoforms harbouring the isoform appeared to be better translated under hypoxia than those without it (**Reviewer Figure 6c-d**). We extended this analysis of differences in 5'UTR isoform-level translation efficiency across transcripts for several of the identified motifs and found enrichments of low p-values under hypoxia (indicating differential translation), more often than under normoxia (**Reviewer Figure 7a**). Furthermore, these differences in translation efficiency were also accompanied by significant TSS switching leading to the inclusion or exclusion of the motif under hypoxia (**Reviewer Figure 7b**). This supports that the identified RNA elements play a role in modulating translation under hypoxia, and are altered by hypoxia-induced TSS switching. Although these results are consistent with the tenet of the manuscript, they have not been included in the revised version as we feel that this area requires further development, which we are actively pursuing in our ongoing and future work.

In regard to examining RNA structure, we included the predicted folding energy of the 5'UTR in our models (it can be seen in the networks in **Figure 3b**, **Figure S4b**), as well as the change in folding energy resulting from TSS switching, and the difference in GC content (**Supplemental File S4**). We did not find the differences in 5'UTR folding energy or GC content to be significant in explaining changes in the translome. As we have reported alterations to thousands of 5'UTRs, there are many open questions and directions that could be pursued. We feel that examining the resulting differences in RNA structure from hypoxia-induced TSS switching is an interesting topic for future work, but requires extensive time and experimental approaches that exceed the scope of the current study. Similarly, with changes in the loading of RNA-binding proteins, we have identified a number of motifs (> 40) that appear to impact translation efficiency under hypoxia. We have shown that TSS switching events lead to alterations in known RNA motifs that impact translation (e.g. TOP motifs, uORFs) but also revealed previously unknown mRNA features that are likely to modulate protein synthesis under hypoxia. These include the SGCSGCS motif that was gained or lost in > 1000 transcripts in T47Ds as a result of TSS switching (**Figure 3c and j**), and the AAGAAA motif that was correlated with translational suppression of mTOR-dependent genes (**Figure 3b and j**, and **Figure S4a-b and g**), among others. Hence, hypoxia-induced TSS switching is expected to regulate protein synthesis through a variety of parallel mechanisms that we intend to study in the future including by employing appropriate reporters.

-Authors propose a model where hypoxia leads to shorter UTRs that are able to more competitively use the translation machinery. However, 5' TOP RNAs are known to be very short and are translationally inhibited during hypoxia. Therefore this simplified model is not sufficient to explain the translation phenotype. Can authors explain?

We apologize if the way we have presented our findings was unclear. Our results show that TSS switching causes not only shortening, but also lengthening of 5'UTRs. In T47Ds there is a relatively equal propensity for shortening or lengthening, while in H9s there is a stronger preference for lengthening. This is highlighted in **Figure 1k**. We have also shown that a change in 5'UTR length resulting from TSS switching is significantly associated with altered translation efficiency. In both cell types, for the most part, we found that mRNAs with longer 5'UTRs are translated more efficiently under hypoxia (**Figure 3b and Figure S4b**). In T47D cells, although 5'UTR shortening was mirrored by increased translation of a subset of mRNAs under hypoxia (**Figure 3h**), a decrease in 5'UTR length was also associated with reduced translation efficiency counteracting increased mRNA abundance (translational offsetting) (**Figure 3i**). In turn, in H9 cells, 5'UTR lengthening was associated with increased translation efficiency counteracting

decreased mRNA abundance (translational offsetting) (**Figure S4f**). Therefore, both shortening and lengthening of 5'UTRs were associated with altered translation efficiencies. Although there was a significant overall directionality to these effects, there were many exceptions as can be seen in the scatterplots in **Figure 3h-l** and **Figure S4f**. It is also important to note that “shortening” and “lengthening” are relative terms, for example a long 5'UTR may undergo shortening but still remain long as compared to other 5'UTRs in the cell. We clarified these points in the revised version of the manuscript.

Finally, although we concur that classical TOP mRNAs usually have relatively short 5'UTRs, these have a highly specific motif and mode of regulation that we feel may be difficult to extrapolate to all short 5'UTRs that do not contain 5'TOP motifs. We unexpectedly observed that 5'TOP mRNAs are strongly translationally offset under hypoxia suggesting that their regulation under these conditions may be more nuanced than previously appreciated (**Figure S2c-d**). To support functional relevance, we have shown PDK1 as an example of a gene highly relevant to hypoxia phenotypes that undergoes TSS switching under hypoxia. It is coincidental that for PDK1 the short 5'UTR isoforms are those that are more efficiently translated, but there were many transcripts where this was not the case. Accordingly, we do not propose a model whereby hypoxia exclusively leads to shorter 5'UTRs that are better translated, which we clarified in the revised manuscript.

- The only translation function data is in found in Figure 4, and does not make a lot of sense. (1) 4F - RT-PCR is not able to distinguish between the isoforms correctly. As stated in the paper, the short isoform RT-PCR includes long isoform in the quantification. Therefore, authors should perform a northern blot to actually perform the experiment correctly.

We apologize that our findings were not presented clearly enough and we have thus revised the text accordingly. It was impossible to design a probe or primers that would detect shorter isoforms without including longer ones. We are not aware of Northern blotting methods that would allow resolving mRNAs with differences in 5'UTR isoforms at sufficient resolution to quantify many of the differences in 5'UTR isoforms we report here. For PDK1, for example, the difference in length between the 134 nt and 36 nt, and 57 nt and 36 nt 5'UTR isoforms would amount to ~2% and ~0.4% of the full transcript, respectively. Furthermore, in many cases the Northern blot approach would be inappropriate due to very small differences in 5'UTR lengths within the same TSS cluster (often just a few nucleotides). Lastly, Northern blots would suffer from the same caveat as our RT-qPCR approach in that the shorter isoforms are contained within longer ones. Based on this, we believe that the RT-qPCR approach was the best available approach in this instance. Notably, this approach will underestimate the enrichment of the shorter 5'UTR isoforms of PDK1. For this reason, we deemed it to be conservative, although not absolutely quantitative. Due to the fact these isoforms were also detected and demonstrated the same expression patterns in multiple cell lines and across several nanoCAGE datasets (**Figure 7a** and **Figure S8c**), as well as using 5'RACE (**Figure 7a**), we feel our approach has been sufficiently rigorous to confirm both the presence and enrichment of shorter 5'UTR isoforms of PDK1 under hypoxia. To validate the difference in translation efficiency even further, we have now used cells that were engineered by a CRISPR/overexpression approach to express only one PDK1 5'UTR isoform. In these cells we also observed increased protein levels as compared to mRNA expression for the 36 nt 5'UTR isoform of PDK1 (**Figure 7i**), consistent with all other data supporting its enhanced translation efficiency (distribution across polysomes of endogenous PDK1 5'UTR isoforms in **Figure 7e**, and validation by luciferase reporter constructs in **Figure 7f-g**).

(2) 4G - The way that the authors are interpreting this data is very confusing. Authors say that beta actin is the control cellular gene that is translationally shutoff during hypoxia. However, their shorter UTR (57 and 37nt) RNAs are also translationally down regulated to similar levels to beta-

actin, and therefore the short UTR reporters are not showing a different “hypoxia resistant” phenotype.

The reduced translation of beta-actin mRNA (not shut-off) and all isoforms of PDK1 is consistent with the global decrease in protein synthesis that is expected under hypoxia (see **Figures 2c-d**). We do not suggest that the shorter 5'UTRs of PDK1 demonstrate a “hypoxia resistant” phenotype, but rather state that these isoforms are more efficiently translated both in hypoxia and normoxia relative to other isoforms, whereby their availability is increased under hypoxia due to changes in TSS usage (**See Figure 7**). We apologize for the confusion and have revised the text to clarify these points.

At the same time, authors say that the shorter UTR RNA is more highly translated than the 134nt RNA. While this is true, again this is similar to levels of beta actin translation, which is the negative control. Therefore, this data instead shows that the UTR length of PDK1 is not sufficient to allow for resistance of shutoff during hypoxia and that this “RNA determinant” does not explain the ability of PDK1 to be translationally upregulated during hypoxia.

As above, we did not mean to suggest that the shorter 5'UTR PDK1 isoforms are resistant to translational suppression under hypoxia. They are simply more efficiently translated than the longer 5'UTR PDK1 isoforms under both normoxia and hypoxia. Under hypoxia, expression of the shorter PDK1 5'UTR isoforms is induced which, in combination with its relatively higher translational efficiency as compared to longer isoforms, allows sufficient synthesis of PDK1 protein. We clarified these points in the manuscript.

- There are no experiments showing that TSS switching functionally changes the hypoxia response. For example, authors could engineer a cell line which exclusively expresses a short versus long UTR version of PDK1 (or another identified gene of interest) and show that the cells are no longer able to survive during hypoxia.

Thank you for the constructive suggestion. Please see the response to comment #4 from reviewer 1 above wherein we have addressed functional relevance with three complementary experimental approaches, including the approach recommended for PDK1.

- Given that the finding of TSS switching is not novel, authors need to show why the change in TSS leads to translational control. For example, it would be important to show how other initiation factors or RNA-binding proteins are involved, or even together with longer range interactions. Indeed, there is beautiful work from Uniacke...Lee (22678294) showing how a switch in usage of eIF4E to eIF4E2 regulates translation during hypoxia via an RNA binding site in the 3' UTR. Could TSS switching (via the enriched RNA motifs) be involved in this kind of well characterized translational control pathway? The existing bioinformatics based mining, although extensive, seems to have missed out on detecting known pathways and identifying new biochemical mechanisms.

Thank you for the comments. We have now included a signature of eIF4E2-dependent translation in our translome models. This signature was derived from ribosome profiling data comparing eIF4E2 KO and control HEK293 cells (PMID: 29412140). We found that this signature was not able to explain changes in translation efficiency and was not significant in either univariate or multivariate regression analyses, in either T47D or H9 cells (see **Reviewer Figures 8 and 9**). As such, we concluded that this eIF4E2-dependent mechanism does not appear to be active in our experimental systems. This is consistent with the findings of Ratcliffe, who extensively examined

this mechanism in the same experimental model as the original study cited above and challenged the conclusions of the original study (PMID: 36097292).

Our analysis identified the most well-characterized pathways that have known roles in the hypoxic translome (mTOR, and ISR, PMID: 18846101). In addition, we have also identified previously unknown potential roles for factors such as DHX9, DAP5, MNK1/2/eIF4E, eIF4G1, and ER-alpha which we previously demonstrated regulates tRNA U34-modifying enzymes (PMID: 31556460). Importantly, many of these factors were previously associated with other types of stress response (e.g. DAP5; PMID: 38828390).

“...authors need to show why the change in TSS leads to translational control”. As we have discussed in the manuscript, we do not feel it is possible to show a singular mechanism of translational control, as is the case with the TSS switching impacting LUTIs cited above, which rely on the inclusion or exclusion of uORFs. 5'UTRs are regulatory hubs that interact with a myriad of factors impacting rates of translation initiation. The mechanism we report here is that hypoxia induces TSS switching that alters the sequence composition of 5'UTRs on a broad scale, impacting their potential for interactions with regulatory factors. We have identified numerous sequence features of 5'UTRs that are altered by TSS switching, and associated with changes in translation efficiency. Some of these were known, for example the gain and loss of uORFs and TOP motifs (**Figure 3j, Figure S4e and g**), however many were unknown such as the SGCSGCS motif frequently altered by TSS switching in T47D cells (**Figure 3c and j, Figure S3b**). We have also produced transcriptome-wide preliminary data which suggests that a number of these 5'UTR motifs are significantly associated with altered translation efficiency at the isoform level (**Reviewer Figures 6 and 7**). Concurrently, we have shown that hypoxia reprograms the translational machinery via alterations to both known and previously unappreciated pathways mentioned in the response above. Therefore, the hypoxia-adaptive proteome is shaped in part by the convergence of changes to the translation initiation machinery that interacts with 5'UTRs, and changes in TSS usage producing 5'UTRs with different potentials for interactions with translation initiation machinery. Furthermore, we have shown that modulation of H3K4me3 through inhibition of KDM5 alone is sufficient to alter the composition of 5'UTRs and the proteome (independently of mRNA levels), even in the absence of translational reprogramming via mTOR inhibition and ISR activation (**Figure 5**). As such, the 5'UTR elements and pathways identified here are exciting avenues for future work.

- Authors do not do experiments to distinguish between why there are changes with TSS start site enrichment, but note it could transcription or stability. Authors should measure stability of target candidates using a pulse chase analysis and if it is stability, identify the RNA motifs (such as the proposed G/C richness) that contribute by using RNA mutants.

Thank you for the suggestions. We have added **Figure S9d**, in which we used actinomycin D to monitor the stability of the different PDK1 5'UTR isoforms. We found that there is no difference in the stability of PDK1 5'UTR isoforms under hypoxia. For this reason, we concluded that the enrichment of shorter PDK1 isoforms seen under hypoxia is indeed due to increased transcription and not enhanced stability. Beyond this, we have used an approach for measuring 5'UTR isoform stability transcriptome-wide using 4-thiouridine incorporation in combination with nanoCAGE sequencing. However, as concurrently measuring TSS switching and 5'UTR isoform stability transcriptome-wide has (to our knowledge) not been attempted before, significant additional methods development is required to properly analyze and interpret this dataset. In light of the scope of the results already included in our study, we believe that it is reasonable for this to remain a topic of our ongoing work.

Minor comments:

- Figure 1B - why is one replicate discarded?

One of our T47D nanoCAGE replicates for normoxia failed sequencing QC, and unfortunately had to be removed from the analysis for this reason.

- Figure 2 - why are the conditions between Fig 1 and 2 changed from 0.5% to 1% O₂?

Please see our response to reviewer #1's comment #1.

Fig S4 - It is confusing to annotate the gel "polysome" as it is a western blot from cytoplasmic lysate although matched with the polysomes (e.g. it's not protein isolated from a polysome)

Thank you for the comment, we agree it was confusing and have revised the labelling for clarity.

Fig 4F - There needs to be error bars in the quantification and the matching polysome profile trace.

We have revised how this data is plotted to increase clarity.

Fig 4G - This way of expressing the p value assignment is confusing, it's unclear what the comparison is. Please use the standard stars and lines to show comparisons.

As described in the figure caption, the p-values are derived from a linear model with the design: $\log_2(\text{Luminescence}) \sim \text{Replicate} + \text{Isoform} + \text{Treatment} + \text{Isoform:Treatment}$

The hypotheses tested were:

- 1) Are the isoforms translated differently?
- 2) Does the treatment alter isoform translation?
- 3) Are the isoforms translated differently between treatment conditions?

Hypothesis 1 is addressed by the Isoform term in the model, which indicates that indeed they are translated at different efficiencies. Hypothesis 2 is similarly addressed by the treatment term, which confirms all isoforms have reduced translation efficiency under hypoxia. Hypothesis 3 is the interaction term between treatment and isoform, and was not significant, indicating there are no differences in how isoforms are translated between hypoxia and normoxia, i.e., despite the global decrease under hypoxia, the translation efficiencies of the isoforms are the same relative to each other. As we are not performing pairwise comparisons, stars and lines are not appropriate in this instance. However, we have tried to more clearly describe the appropriate statistics in the caption.

REVIEWER FIGURE CAPTIONS:

Reviewer Figure 1: Comparison of translome changes between T47D and H9 cells under hypoxia.

Bar plots showing how genes categorized into regulatory modes (by anota2seq analysis) under hypoxia in H9 cells are regulated in hypoxic T47D cells (left), and how genes categorized into regulatory modes under hypoxia in T47D cells are regulated in hypoxic H9 cells (right).

Reviewer Figure 2: Threshold-independent comparison of translome changes between T47D and H9 cells under hypoxia.

a) Scatterplots from anota2seq analysis of T47D cells with the location of genes categorized into mRNA abundance up and down modes in hypoxic H9 cells colored (left panel). Empirical cumulative distribution functions showing the \log_2 fold changes for these mRNAs in polysome-associated (middle panel), and total mRNA fractions (right panel). Grey lines correspond to the background (i.e., all other genes). Significant differences in gene expression were determined by Wilcoxon rank-sum test. Differences in distributions at quantiles are indicated.

b) Same as **(a)** but showing genes categorized into mRNA abundance up and down modes in T47D cells, assessed in H9 cells.

c) Same as **(a)** but showing genes categorized into translation up and down modes in H9 cells, assessed in T47D cells.

d) Same as **(a)** but showing genes categorized into translation up and down modes in T47D cells, assessed in H9 cells.

e) Same as **(a)** showing genes in the offsetting modes identified in hypoxic H9 cells, assessed in T47D cells.

f) Same as **(a)** showing genes in the offsetting modes identified in hypoxic T47D cells, assessed in H9 cells.

Reviewer Figure 3: Comparison of gene ontology enrichments for genes regulated under hypoxia in T47D and H9 cells.

For comparisons of enriched gene ontology terms between the T47D and H9 cells, ClueGO analysis was performed using two gene lists at a time, corresponding to the regulatory categories identified by anota2seq analysis in each cell line. The background used was the combination of all genes passing anota2seq quality thresholds in both cell types. Enrichments were selected based on the following criteria, additional to those stated above: Combine Clusters With 'Or' = TRUE, Percent for a Cluster to be Specific = 60.0, GO Fusion = TRUE.

a-d) Gene Ontology enrichments among genes that were translationally activated **(a)**, suppressed **(b)**, offset (mRNA up) **(c)**, and offset (mRNA down) **(d)** under hypoxia in T47D and H9 cells. Terms more enriched among genes in T47D are shaded blue, while those more enriched among genes regulated in H9 cells are shaded yellow. Terms commonly enriched (zero specific enrichment) among both cell lines are shaded in grey.

Reviewer Figure 4: Isoform-level translational offsetting under hypoxia.

a) T47D cells were subjected to hypoxia (1% O₂) for 48 hrs before being subjected to polysome fractionation. Total cytosolic mRNA, as well as polysome-associated mRNA were subjected to

nanoCAGE sequencing allowing transcriptome-wide quantification of 5'UTR isoforms (n = 3). Kernel density estimation of FDRs indicate an enrichment in significant differences in 5'UTR isoform-level translation in hypoxia compared to normoxia.

b) 5'UTR isoform expression for RCE1 in total cytosolic, and polysome-associated mRNA fractions from normoxia and hypoxia. Numbers in bold indicate the % of total transcript expression in the respective mRNA fractions. Under hypoxia, there is a switch towards expression of the shorter TSS cluster (40% to 82%), and a loss in expression of the longer cluster (44% to 18%). This change in expression is offset at the level of translation, as the proportion of longer and shorter isoforms remains quite steady in the polysomal fraction.

c) Overall expression (all 5'UTR isoforms) of RCE1 in total cytosolic and polysomal mRNA fractions was unchanged, despite the switch in isoform abundance.

Reviewer Figure 5: Coordination between mTOR-dependent translation and mRNA features of 5'UTRs.

a) Translatome changes in hypoxia-treated T47D cells were modelled using anota2seqUtils (as in manuscript **Figure 3b**), examining only the contribution of mTOR-activated translation, and five mRNA features of 5'UTRs (length, GC content, AAGAAA motif, fold energy, and uORFs). The table indicates the p-value, FDR, and proportion of hypoxia-sensitive translation that can be explained by each variable in a univariate model.

b) Table shows the changes in F-value for each variable during stepwise regression modelling. The feature explaining the largest proportion of variance in translation is indicated in green, and features removed from the model due to the inability to significantly explain additional changes in translation are indicated in red. Orange highlights substantial changes in F value, indicating co-linearity between variables.

c) Table providing p-values for each variable from the stepwise regression model, the proportion of variance in translation explained by each variable (in the final "Omnibus" model), and the proportion of variance explained after adjusting for the contribution of the other variables in the model ("Adjusted"). The mTOR-activated translation signature could explain 18.4% of variance, but this decreased to 12.2% after adjusting for the contributions of the 5'UTR features. Therefore, ~33.5% of hypoxia-induced changes in translation to mTOR-activated genes can be explained by the features of 5'UTRs included in the model. This supports a high degree of coordination between mTOR-dependent translational regulation and 5'UTR features.

d) Network plot indicates the percentage of hypoxia-dependent translation changes in T47D cells explained by each feature that was significant in the final omnibus regression model. Connections between features indicate substantial correlations. Colors of the nodes indicate if the feature is associated with translation activation or suppression under hypoxia.

Figure 6: TSS switching under hypoxia leads to a gain or loss of regulatory 5'UTR feature.

As described in Reviewer Figure 1, total cytosolic and polysome-associated mRNA fractions from T47D cells treated with hypoxia or normoxia (1% or 20% O₂) were subjected to nanoCAGE sequencing. Shown are two examples of transcripts containing regulatory 5'UTR motifs that were identified as being associated with translational activation under hypoxia.

a) 5'UTR isoforms for NM_001127322 (CBX5) were divided into two groups either containing the UCCUCU motif (purple) and those without the motif (green), and quantified in both total and

translated mRNA fractions in hypoxia and normoxia. The position of the putative regulatory motif is indicated. The x-axis represents the distance to the AUG start codon, with the black bar denoting the RefSeq-annotated 5'UTR length.

b) Bar plots indicate the difference in translation efficiency between 5'UTR isoforms with, and without the motif, in both normoxia and hypoxia. Isoforms containing the motifs are less efficiently translated than those without under normoxia, and this effect is reversed under hypoxia, consistent with that this motif is associated with translation activation.

c) Same as **a** for the CGGACGU motif that was found to be associated with translational activation under hypoxia in T47D cells in NM_144732 (HNRNPUL1).

d) Same as **b** for the CGGACGU motif in NM_144732 (HNRNPUL1).

Reviewer Figure 7: 5'UTRs containing identified motifs are differentially translated under hypoxia.

a) Kernel density estimation of p-values from the analysis of differential translation of 5'UTR isoforms containing different motifs identified in translome modelling of hypoxia-treated T47D cells in **Figure 3b-c**. The enrichment of low p-values indicates that 5'UTR isoforms containing these motifs are more significantly differentially translated under hypoxia compared to normoxia. Note that these significant differences can represent both translational activation and suppression of 5'UTR isoforms.

b) Kernel density estimation of p-values from the analysis of both differential TSS usage (total mRNA) and differential translation of 5'UTR isoforms (polysome-associated mRNA) containing the identified regulatory motifs. Enrichment of low p-values indicates that there are transcripts where TSS switching impacts the inclusion or exclusion of the motif in the 5'UTR, as well as translation of the resulting isoforms is significantly different between hypoxia and normoxia.

Reviewer Figure 8: eIF4E2-dependent translation does not independently contribute to translome changes in T47D cells under hypoxia.

a) Translatome modelling using anota2seqUtils shown in **Figure 3b** was repeated adding signatures of eIF4E2-dependent translation (from PMID: 29412140). The table indicates the p-value, FDR, and proportion of hypoxia-sensitive translation that can be explained by each variable in a univariate model. The eIF4E2-dependent translation signatures were not significant in explaining changes in the translome under hypoxia in univariate models.

b) Table shows the changes in F-value for each variable during stepwise regression modelling. The feature explaining the largest proportion of variance in translation is indicated in green, and features removed from the model due to the inability to significantly explain additional changes in translation are indicated in red. Signatures of eIF4E2-dependent translation are indicated by red arrows, and are removed from the model in the first two steps.

Reviewer Figure 9: eIF4E2-dependent translation does not independently contribute to translome changes in H9 cells under hypoxia.

The same analysis as presented in Reviewer Figure 5 was repeated for the H9 cells, with the same finding that eIF4E2 signatures were not significant in explaining changes in translation in univariate models, and were removed from the stepwise regression model in the first step.

Note that while results are shown for the translation mode of regulation, the eIF4E2-dependent translation signatures were also not significant in univariate or multivariate analyses for the offsetting regulation mode in either cell type.

a**b****c****d**

a

Features	Pvalue_Univariate	FDRvalue_Univariate	VarianceExplained_Univariate
mTOR_activated_translation	4.6e-33	2.7e-32	18.40
T47D_UTR5_gc	1.1e-15	5.4e-15	8.71
T47D_UTR5_length	3.9e-15	1.5e-14	8.38
UTR5_AAGAAA	5.0e-07	1.5e-06	3.52
fold_energy_UTR5	2.6e-06	5.1e-06	3.09
uORFs	3.8e-04	3.8e-04	1.78

b

	step1	step2	step3
mTOR_activated_translation	159.24		
T47D_UTR5_gc	67.36	43.86	
T47D_UTR5_length	64.59	29.71	10.12
uORFs	12.76	5.15	0.13
fold_energy_UTR5	22.48	7.88	2.21
UTR5_AAGAAA	25.76	16.13	3.54

c

Features	Pvalue	VarianceExplained_Omnibus	VarianceExplained_Adjusted
mTOR_activated_translation	3.7e-35	18.40	12.24
T47D_UTR5_gc	5.3e-11	4.78	2.57
T47D_UTR5_length	1.5e-03	1.09	1.09

33.47% of hypoxia-induced translation changes to mTOR-activated genes can be explained by these five mRNA features of 5'UTRs

Total variance explained: 24.27%

d

a**b**
a

T47D

Features	Pvalue_Univariate	FDRvalue_Univariate	VarianceExplained_Univariate
mTOR_activated_translation	1.3e-31	1.1e-29	19.14
mTOR_suppressed_translation	3.0e-24	2.6e-22	14.80
T47D_UTR5gc	2.3e-16	1.9e-14	9.92
high_eIF4E_activated_translation	1.6e-14	1.3e-12	8.75
T47D_UTR5length	1.1e-13	8.6e-12	8.22
ISR_activated_translation	8.4e-12	6.8e-10	6.99
ISR_suppressed_translation	5.4e-10	4.3e-08	5.80
UTR5_AAGAAA	1.3e-07	9.9e-06	4.24
Offset_mRNADown_upon_ERa_KD	1.2e-06	9.0e-05	3.60
UTR5_UUCUUU	1.4e-06	1.1e-04	3.54
fold_energy_UTR5	1.5e-06	1.1e-04	3.53
UTR5_AAAAAA	8.9e-06	6.7e-04	3.02
PRTE	1.1e-05	8.1e-04	2.96
uORFs	1.3e-05	9.5e-04	2.91
UTR5_introns	6.6e-05	4.7e-03	2.44
UTR5_UCCUCU	9.4e-05	6.6e-03	2.34
UTR5_ACUUCU	1.3e-04	8.9e-03	2.25
UTR5_AAAAGA	2.7e-04	1.9e-02	2.03
high_phosho_eIF4E_activated_translation	2.9e-04	2.0e-02	2.01
UTR5_ACCAAA	7.1e-04	4.8e-02	1.76

b

	step1	step2	step3	step4	step5	step6	step7	step8	step9	step10	step11	step12	step13	step14
mTOR_activated_translation	152.83													
mTOR_suppressed_translation	112	79.31												
high_eIF4E_activated_translation	61.85	39.32	29.78											
T47D_UTR5gc	71.02	49.64	28.96	23.8										
mTOR_offset_mRNUp	11.29	9.62	14.68	13.62	15.29									
ISR_translationDown	39.72	22.96	19.47	16.76	13.19	14.07								
ISR_translationUp	48.44	29.91	17.49	14.05	11.91	13.37	12.24							
UTR5_UCCUCU	15.46	14.52	11.8	9.76	6.86	7.63	7.88	9.79						
UTR5_UUCGUA	10.92	9.17	12.05	11.45	6.82	7.35	7.67	9.62	9.19					
Offset_mRNADown_upon_ERa_KD	24.11	17.49	14.37	12.05	9.28	8.99	8.07	7.6	7.54	7.42				
DHX9_KD_suppressed_translation	5.55	4.2	5.03	6.42	6.31	6.05	7.13	6.8	6.81	6.42	6.51			
UTR5_CGGCAGC	7.43	6.24	7.04	6.54	6.36	6.76	6.53	5.5	5.99	6.29	6.24	6.08		
UTR5_AAAAAA	20.05	17.74	11.53	11.17	4.66	5.01	4.9	5.25	4.89	5.26	5.25	5.74	6.13	
DHX9_KD_activated_translation	10.1	6.8	4.35	4.94	4.87	5.33	5.07	6.05	5.89	5.76	5.45	5.05	5.29	5.45
UTR5_UUCUUU	23.68	17.43	13.52	12.16	4.83	4.26	4.31	4.35	3.5					
T47D_UTR5length	57.75	24.91	16.76	15.07	4.93	5.58	5.03	5.02	2.78					
fold_energy_UTR5	23.58	8.2	4.38	4.09	1.11									
UTR5_introns	16.15	10.42	7.79	8.27	1.42									
PRTE	19.65	14.83	9.58	7.16	2.88									
UTR5_AAGAAA	28.56	19.18	7.4	7.11	1.44									
UTR5_ACUUCU	14.87	9.8	5.78	4.51	1.04									
UTR5_ACCUUC	8.54	5.27	4.42	4.66	2.46									
UTR5_CGGGCC	8.72	7.69	6.26	5.58	2.46									
uORFs	19.32	10.46	4.2	3.78										
DAP5KO_KO_suppressed_translation	8.09	5.45	4.1	3.49										
UTR5_GAAGA	11.13	7.78	3.2											
UTR5_ACCAAA	11.57	6.43	1											
UTR5_AAAAGA	13.39	8.71	1.94											
UTR5_UAGGCA	7.82	5.37	2.67											
high_phosho_eIF4E_activated_translation	13.25	9.9	2.97											
UTR5_AGGGAA	8.35	3.8												
UTR5_ACUUGC	5.34	2.57												
UTR5_GGAGUG	4.17	1.6												
UTR5_GCGGUC	6.26	3.75												
UTR5_UCGUG	7.98	3.26												
UTR5_AUUUUC	6.56	3.75												
EIF4E2_KO_suppressed_translation	4.46	2.81												
DAP5KO_KO_activated_translation	5.31	2.25												
T47D_TOPscore	2.61													
CERT	0.23													
G4	0.14													
UTR5_SGCGSCS	0.05													
UTR5_CCCCTGGGCC	0													
UTR5_CGUUGUU	0.34													
UTR5_UCGUGC	0.06													
UTR5_UAAGC	1.79													
UTR5_GAUGC	0													
UTR5_CGCGCC	0.45													
UTR5_CCCGGC	2.08													
UTR5_ACGGUC	1.38													
UTR5_ACGGG	2.2													
UTR5_CUACAAC	2.43													
UTR5_CGUGGG	2.18													
UTR5_CGCGGG	0.11													
UTR5_CGCGGCG	1.74													
UTR5_CCCGCG	2.25													
UTR5_CCGGCG	0.07													
UTR5_CGCGCG	0.37													
UTR5_CGCUCCG	3.82													
UTR5_CCCUUGA	1.92													
UTR5_CGAAGGA	2.07													
UTR5_CGACCGA	2.18													
UTR5_CGUUGUC	0.08													
UTR5_CGCGGAC	1.26													
UTR5_CAUCAG	0.08													
UTR5_GAAGGAG	0.14													
UTR5_CCGCGCC	0.01													
UTR5_CCCUCCU	0.88													
UTR5_CAGUAC	1.6													
UTR5_CUCGCUUU	1.21													
UTR5_UCUGCUGU	1.33													
UTR5_CGUGCUGU	1.19													
UTR5_CGUGCUUC	0.1													
UTR5_CUUGCUGC	0.37													
UTR5_UGGGGU	3.29													
UTR5_GCUAGUA	0.92													
UTR5_UGGGGAA	1.84													
UTR5_UUGGAGG	0.02													
EIF4E2_KO_activated_translation	1.12													
SNAT2_suppressed_translation	3.26													
SNAT2_activated_translation	1.67													
high_phosho_eIF4E_suppressed_translation	0.33													
Offset_mRNUp_upon_ERa_KD	2.02													
sL_eIF4G_suppressed_translation	3.06													
sL_eIF4G_activated_translation	1.82													
sL_eIF4A1_suppressed_translation	1.58													
mTOR_offset_mRNADown	0.07													

a **H9**

Features	Pvalue_Univariate	FDRvalue_Univariate	VarianceExplained_Univariate
mTOR_translationDown_InsvsInsTor_updated	7.6e-18	5.5e-16	18.16
mTOR_translationUp_InsvsInsTor_updated	1.3e-15	9.5e-14	15.87
H9_UTR5gc	9.4e-13	6.6e-11	12.89
high4Eactivated	3.2e-10	2.2e-08	10.16
ISR_translationUp	5.4e-08	3.7e-06	7.69
UTR5_UAAAAA	1.2e-07	7.7e-06	7.32
UTR5_UUUAAA	2.5e-07	1.6e-05	6.95
UTR5_AAAAAU	1.3e-06	8.6e-05	6.13
high4EpActivated	1.8e-06	1.1e-04	5.99
UTR5_UUAAAA	1.3e-05	7.9e-04	5.03
UTR5_meme_AAGAAA	1.3e-05	8.3e-04	5.00
UTR5_CAAAAU	1.5e-05	8.9e-04	4.96
UTR5_UUAAUU	2.2e-05	1.3e-03	4.75
UTR5_AAAAUU	6.3e-05	3.7e-03	4.24
uORFs	1.2e-04	6.9e-03	3.93
UTR5_AAUAAU	2.0e-04	1.2e-02	3.66
UTR5_AGAAAAU	2.3e-04	1.3e-02	3.60
UTR5_AAAAAA	3.9e-04	2.2e-02	3.34
UTR5_GAGACGA	4.5e-04	2.4e-02	3.28
UTR5_GAGGAAC	4.8e-04	2.6e-02	3.24
UTR5_CCCUGC	5.6e-04	2.9e-02	3.17
UTR5_UACAUC	8.8e-04	4.5e-02	2.95

b

	step1	step2	step3	step4	step5	step6	step7	step8	step9	step10	step11	step12	step13	step14	step15	step16	step17	step18
mTOR_suppressed_translation	82.1																	
mTOR_activated_translation	69.79	46.21																
H9_UTR5gc	54.73	30.62	24.24															
high_phosho_eIF4E_activated_translation	23.56	19.67	15.52	11.06														
UTR5_UAAAAA	23.24	24.45	21.64	10.25	9.95													
UTR5_CCCUGC	12.12	7.57	9.89	9.39	9.13	10.54												
high_eIF4E_activated_translation	41.82	27.14	12.45	8.49	8.66	8.75	11.03											
UTR5_GAGACGA	12.54	9.15	9.48	8.73	9.94	6.57	8.03	8.34										
mTOR_offset_mRNAup	6.35	11.84	9.61	9.95	9.29	9.36	8.45	7.77	7.97									
DAP5_KO_suppressed_translation	9.86	7.94	6.56	6.62	5.68	6.97	6.74	6.33	6.53	7.08								
UTR5_CGGCGCC	7.11	6.16	10.81	7.74	7.47	10.25	6.01	6	7.06	6.73	7.38							
UTR5_CCCUGCA	8.28	5.26	5.91	8.96	5.82	5.92	5.95	5.88	6.03	6.28	6.8	6.7						
UTR5_CCCAGCA	4.76	4.81	4.28	3.9	4.53	5.26	5.72	5.65	5.87	6.14	6.52	6.29	6.36					
UTR5_GGAAGAAG	10.52	10.91	9.02	5.56	7.2	5.14	7.66	7.8	6.09	6.45	5.41	5.99	5.79	6.14				
UTR5_CGCCCGC	8.79	7.81	6.78	5.81	6.11	6.25	6.14	7.04	7.01	6.74	6.44	6.14	6.06	5.96	5.8			
UTR5_GAGGAAC	12.39	6.74	5.28	5.76	7.41	5.59	6.35	6.72	5.62	5.98	5.45	4.98	5.17	5.53	4.68	5.87		
s_eIF4G_activated_translation	3.99	7.45	6	7.14	6.04	6.52	6.65	6.15	6.22	6.75	5.14	4.69	4.95	5.1	5.4	5.23	5.49	
ISR_translationUp	30.63	8.52	6.57	5.95	6.74	7.46	6.67	5.82	5.53	6.18	4.37	4.18	4.78	5.47	4.27	5.02	4.62	4.31
UTR5_CCCCGCG	8.97	7.47	9.74	6.24	5.96	5.78	4.23	4.5	4.35	4.03	3.65							
UTR5_UUAAUU	18.44	10.86	9.81	4.25	5.94	1.35												
UTR5_AAAAAU	24.17	17.94	14.06	5.05	5.39	0.02												
UTR5_UUUAAA	27.64	20.39	17.8	7.04	7.65	2.06												
UTR5_CAAAAU	19.31	10.64	10.16	5.66	7.25	3.52												
UTR5_UCCGGA	8.69	6.17	4.26	4.63	3.72													
uORFs	15.13	11.02	7.65	0.59														
UTR5_AAGAAA	19.47	16.35	11.77	3.07														
UTR5_AAGAGGA	5.38	5.33	4.55	2.48														
UTR5_UACAUC	11.24	7.61	5.98	2.05														
UTR5_UUAAUUU	6.76	5.72	4.94	3.23														
UTR5_CACAGAC	7.09	8.04	5.88	3.63														
UTR5_AAAUUU	16.37	9.58	7.59	1.83														
UTR5_AAUAUU	14.07	7.77	5.86	1.77														
UTR5_AAAAAA	12.8	8.09	5.79	1.41														
UTR5_AGAAAAU	13.8	10.02	7.9	2.46														
UTR5_UUUUAC	9.68	8.61	6.38	1.75														
UTR5_UUAAAA	19.61	11.9	9.77	1.84														
UTR5_UCCUUU	4.32	6.24	5.58	3.59														
H9_TOPscore	7.88	5.38	2.37															
UTR5_ATTTTG	10.32	5.78	3.63															
UTR5_AUUUUUG	10.32	5.78	3.63															
UTR5_AAUUUA	8.83	4.53	3.54															
UTR5_AUUUUC	9.12	4.54	3.71															
H9_UTR5length	10.37	5.4	2.96															
DAP5_KO_activated_translation	10.15	6.87	3.46															
UTR5_introns	4.51	2.83																
UTR5_UACAGA	6.74	1.78																
Offset_mRNAup_upon_ERa_KD	10.67	3.48																
s_eIF4G_suppressed_translation	6.64	0.76																
ISR_translationDown	4.92	1.31																
fold_energy_UTR5	3.3																	
CERT	2.62																	
PRTE	0.29																	
G4	1.21																	
UTR5_UCUCUCG	0.09																	
UTR5_GGUCCCCA	0.12																	
UTR5_AUGAGCG	0.63																	
UTR5_CUUCAC	0.58																	
UTR5_CGGCGGG	1.27																	
UTR5_CCCAGGA	0.41																	
UTR5_GAGUAGG	3.04																	
UTR5_AAAUUC	0																	
UTR5_UUAGAG	0.32																	
UTR5_UGACGGA	0																	
UTR5_AGGGAU	0.07																	
EIF4E2_KO_suppressed_translation	0.32																	
EIF4E2_KO_activated_translation	1.26																	
SNAT2_suppressed_translation	2.68																	
SNAT2_activated_translation	1.9																	
DHX9_KD_suppressed_translation	0.66																	
DHX9_KD_activated_translation	2.62																	
high_phosho_eIF4E_suppressed_translation	2.69																	
Offset_mRNAdown_upon_ERa_KD	1.26																	
s_eIF4A1_suppressed_translation	0.72																	
mTOR_offset_mRNAdown	3.38																	

Reviewer #1 (Remarks to the Author):

The authors have thoughtfully and carefully addressed the criticisms raised in the initial review. The addition of new data demonstrating both mechanistic links between epigenetic changes and TSS, and the work demonstrating consequential downstream phenotypic changes significantly strengthen the manuscript.

The rationale for using different cell lines and different time points is appreciated but nonetheless leaves weaknesses in the ability to understand the overall importance of the specific changes identified in each line and their importance to our overall understanding of these observations. However, the additional analyses identifying many common patterns of regulation mitigate this.

We thank Reviewer #1 for their time and constructive feedback that has significantly strengthened our study. We are very grateful for their expertise and efforts in reviewing our manuscript.

This reviewer also commented on the responses to reviewer#2's previous concerns and their thoughts are below:

Reviewer#2's points on the revised manuscript:

1. Inconsistencies in Hypoxia Conditions: The authors still present experiments in which they have used varying hypoxia conditions, even within the same cell line. This raises concerns about the interpretations of the data.

>>Ref#1's thoughts: This remains a 'limitation' of the results but I do not agree that it raises significant concerns about the interpretation of the data. The authors responded to the same criticism that I made originally, and in their updated manuscript address this through looking at common changes across the two lines and different hypoxia conditions. I think it is unreasonable to expect them to conduct or repeat new experiments at common oxygen conditions given the findings that have been observed.

Thank you for the feedback. As mentioned by Reviewer #1, we have added additional explanation and comparisons to the manuscript to address this point. We would again emphasize that the difference in hypoxia conditions between cell lines is due to that each cell line responds very differently to hypoxia, with differing kinetics (PMID: 32427827; PMID: 22031289; PMID: 17685448). Timepoints for each cell type were selected based on the biology we are interested in (mRNA translation), not with the intention of directly comparing the cell types at the same time points. Experiments were carried out over the course of several years using different hypoxia chambers. Therefore, for each

experiment, we used oxygen concentrations that induced HIF1 α protein levels and LOX1 mRNA, suppressed mTOR signalling, and induced ISR in a manner comparable to other experiments included in the study.

2. The manuscript lacks a clear and logical explanation for how H3K4me3 modifications directly regulate TSS switching under hypoxia. The authors should explore whether specific subregions of H3K4me3 peaks drive TSS relocation and how pre-existing chromatin states influence this process.

>>Ref#1's thoughts: This is certainly an interesting question and one that arises due to the findings in this paper. The discovery of TSS switching under hypoxia raises a number of interesting follow up questions but I do not think that identifying this mechanism is required for publication.

We agree these are important questions and we have provided preliminary directions in the manuscript that we are now following up on in our current and future studies. However, given the scope of the data already included in a single publication, and the complexity and time-frame of the additional work required to robustly define these mechanisms, we feel it is reasonable for these questions to be pursued in follow-up studies.

3. The C48 inhibitor has not been well-characterized, and we have not been able to find data in the literature that has tested the specificity of the compound. Therefore, a KDM5 knockout model is needed to confirm whether H3K4me3 redistribution is necessary and sufficient for TSS switching. This would clarify whether the observed effects are due to H3K4me3 modulation itself or secondary effects from pharmacological inhibition. KDM5B another H3K4me3 specific demethylase should also be involved and discussed.

>>Ref#1's thoughts: I don't agree with the premise here. A knockout would create longer term phenotypic changes and would be difficult to interpret with respect to the effect of hypoxia. An inhibitor, which creates transient inhibition in a functional cell is much more equivalent to the hypoxic experiment. One could construct and inducible knockdown to address the transient changes but this also goes beyond the scope needed here. The authors should add some description and evidence of the KDM5 inhibitor from previous work.

Compound 48 (C48) was selected as it has been shown to be a potent, selective, and orally bioavailable inhibitor of KDM5 (PMID: 27406798), making it very useful in our ongoing studies that aim to extend on these findings by modulating H3K4me3 in *in vivo* systems. It has IC₅₀ values of 15.1 nM, 4.7 nM, and 65.5 nM for KDM5A, KDM5B, and KDM5C, respectively, and was shown to be significantly less potent against other KDM enzymes (>100-fold selectivity). We have revised the text to make it clear that C48

specifically inhibits KDM5 (A and B, as well as C to a lesser extent). Furthermore, C48 was tested at 10 uM concentration in kinase and Cerep panels and did not inhibit any target at >50% (PMID: 27406798). C48 was compared to *KDM5A* KO in SU-DHL-6 cells and resulted in similar upregulation of the genes examined (PMID: 33786580). C48 has also been used to inhibit KDM5 in MCF7 breast cancer cells (PMID: 30472020), HCC1428 and HMEC cells (Lau *et al.* 2025, eLife) leading to significantly increased H3K4me3. We have now added some additional citations to the manuscript describing the characterization and previous use of C48 to inhibit KDM5 and modulate H3K4me3.

We agree with the response from Reviewer #1. The rationale for using pharmacological inhibition of KDM5 activity rather than a KO system was to more closely mimic the transient nature of hypoxia, and to avoid the off-target adaptive changes and clonal effects that would occur with establishing and culturing cells with stable KO of KDM5.

4. Despite these insights, the study acknowledges challenges in directly correlating sequence-level TSS shifts with H3K4me3 modifications, as H3K4me3 redistributions typically span hundreds to thousands of bases, whereas TSS changes occur on a smaller scale. While preliminary analyses suggest correlations between H3K4me3 shifts and 5'UTR length changes, a simple upstream/downstream model does not fully capture the complexity of the mechanism. The authors should consider mapping changes to H3K4me3 levels at nucleosome resolution using Cut&Run or Cut&Tag, instead of ChIP-seq.

>>Ref#1's thoughts: This is a similar criticism to point 2. Speaks to a deeper understanding of mechanism. I don't think it's needed for publication but authors could address the need to understand this in the discussion.

Thank you for the comments. We are actively pursuing this direction, but as mentioned above, due to the complexity and time-frame required, these mechanisms cannot all be deciphered in a single publication. We have added some additional mention of these points in the discussion section of the manuscript.

Reviewer #2 (Remarks to the Author):

We thank Reviewer #2 for their time and feedback in helping to improve our study. Please see the point-by-point responses above.

This is a revised version of a manuscript, which we found interesting, but contained several overstatements and included experimental data that did not fully support the conclusions. Now, the authors have incorporated new data to address our and other reviewers' concerns, correcting several of these deficiencies. The updated data better explain their results, and the revisions to the text more clearly highlight the significance

of this work. However, issues remain, including questionable interpretations and logical gaps that need to be addressed in a further revision.

1. Inconsistencies in Hypoxia Conditions: The authors still present experiments in which they have used varying hypoxia conditions, even within the same cell line. This raises concerns about the interpretations of the data.

2. The manuscript lacks a clear and logical explanation for how H3K4me3 modifications directly regulate TSS switching under hypoxia. The authors should explore whether specific subregions of H3K4me3 peaks drive TSS relocation and how pre-existing chromatin states influence this process.

3. The C48 inhibitor has not been well-characterized, and we have not been able to find data in the literature that has tested the specificity of the compound. Therefore, a KDM5 knockout model is needed to confirm whether H3K4me3 redistribution is necessary and sufficient for TSS switching. This would clarify whether the observed effects are due to H3K4me3 modulation itself or secondary effects from pharmacological inhibition. KDM5B another H3K4me3 specific demethylase should also be involved and discussed.

4. Despite these insights, the study acknowledges challenges in directly correlating sequence-level TSS shifts with H3K4me3 modifications, as H3K4me3 redistributions typically span hundreds to thousands of bases, whereas TSS changes occur on a smaller scale. While preliminary analyses suggest correlations between H3K4me3 shifts and 5'UTR length changes, a simple upstream/downstream model does not fully capture the complexity of the mechanism. The authors should consider mapping changes to H3K4me3 levels at nucleosome resolution using Cut&Run or Cut&Tag, instead of ChIP-seq.

Reviewer #4 (Remarks to the Author) [Reviewer#3 was unable to review the revised manuscript]:

This is a very interesting and novel manuscript, and I focused my review on the comments raised by reviewer 3. The authors have addressed all the comments and added new bioinformatics analysis and functional experiments to corroborate their original results. I agree with the authors that some of the comments raised were either not feasible to address or outside the scope of this work. I found this manuscript significant and inspiring, providing future directions in the field of gene expression, stress response, and cancer biology. The authors showed a novel interplay between epigenetic mechanisms and translation control in regulating the homeostasis of cancer cells under hypoxia. We still

do not understand the adaptive mechanisms of cancer cells to different environments, and this work provides new insights in this context that have the potential to open compelling new directions in research on gene expression, cancer development, and therapeutic opportunities. This manuscript is ready to be published.

We thank Reviewer #4 for their kind comments, and for their time and efforts in evaluating our study.